# EQUIVARIANT SHAPE-CONDITIONED GENERATION OF 3D MOLECULES FOR LIGAND-BASED DRUG DESIGN

**Keir Adams**[1] **& Connor W. Coley**[1,2]
[1]Department of Chemical Engineering, MIT
[2]Department of Electrical Engineering and Computer Science, MIT
`{keir,ccoley}@mit.edu`

## ABSTRACT

Shape-based virtual screening is widely used in ligand-based drug design to search chemical libraries for molecules with similar 3D shapes yet novel 2D graph structures compared to known ligands. 3D deep generative models can potentially automate this exploration of shape-conditioned 3D chemical space; however, no existing models can reliably generate geometrically realistic drug-like molecules in conformations with a specific shape. We introduce a new multimodal 3D generative model that enables shape-conditioned 3D molecular design by equivariantly encoding molecular shape and variationally encoding chemical identity. We ensure local geometric and chemical validity of generated molecules by using autoregressive fragment-based generation with heuristic bonding geometries, allowing the model to prioritize the scoring of rotatable bonds to best align the growing conformation to the target shape. We evaluate our 3D generative model in tasks relevant to drug design including shape-conditioned generation of chemically diverse molecular structures and shape-constrained molecular property optimization, demonstrating its utility over virtual screening of enumerated libraries.

## 1    INTRODUCTION

Generative models for *de novo* molecular generation have revolutionized computer-aided drug design (CADD) by enabling efficient exploration of chemical space, goal-directed molecular optimization (MO), and automated creation of virtual chemical libraries (Segler et al., 2018; Meyers et al., 2021; Huang et al., 2021; Wang et al., 2022; Du et al., 2022; Bilodeau et al., 2022). Recently, several 3D generative models have been proposed to directly generate low-energy or (bio)active molecular conformations using 3D convolutional networks (CNNs) (Ragoza et al., 2020), reinforcement learning (RL) (Simm et al., 2020a;b), autoregressive generators (Gebauer et al., 2022; Luo & Ji, 2022), or diffusion models (Hoogeboom et al., 2022). These methods have especially enjoyed accelerated development for structure-based drug design (SBDD), where models are trained to generate drug-like molecules in favorable binding poses inside an explicit protein pocket (Drotár et al., 2021; Luo et al., 2022; Liu et al., 2022; Ragoza et al., 2022). However, SBDD requires atomically-resolved structures of a protein target, assumes knowledge of binding sites, and often ignores dynamic pocket flexibility, rendering these methods less effective in many CADD settings.

Ligand-based drug design (LBDD) does not assume knowledge of protein structure. Instead, molecules are compared against previously identified "actives" on the basis of 3D pharmacophore or 3D shape similarity under the principle that molecules with similar structures should share similar activity (Vázquez et al., 2020; Cleves & Jain, 2020). In particular, ROCS (Rapid Overlay of Chemical Structures) is commonly used as a shape-based virtual screening tool to identify molecules with similar shapes to a reference inhibitor and has shown promising results for scaffold-hopping tasks (Rush et al., 2005; Hawkins et al., 2007; Nicholls et al., 2010). However, virtual screening relies on enumeration of chemical libraries, fundamentally restricting its ability to probe new chemical space.

Here, we consider the novel task of *generating* chemically diverse 3D molecular structures *conditioned on* a molecular shape, thereby facilitating the shape-conditioned exploration of chemical space without the limitations of virtual screening (Fig. 1). Importantly, shape-conditioned 3D molecular generation presents unique challenges not encountered in typical 2D generative models:

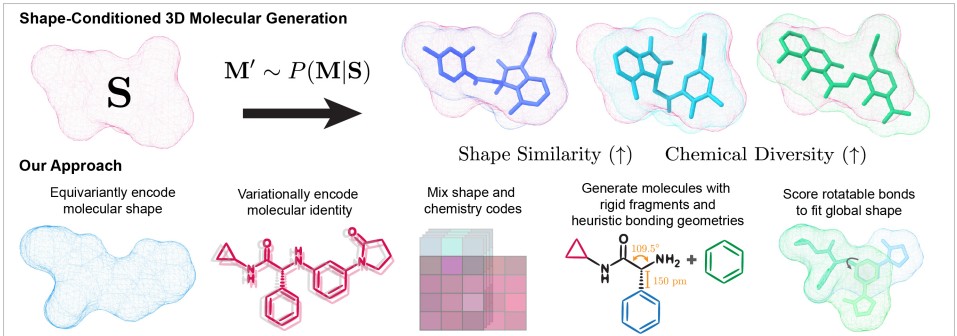

Figure 1: We explore the task of shape-conditioned 3D molecular generation to generate chemically diverse molecules in 3D conformations with high shape similarity to an encoded target shape.

**Challenge 1.** 3D shape-based LBDD involves pairwise comparisons between two arbitrary conformations of arbitrary molecules. Whereas traditional property-conditioned generative models or MO algorithms shift learned data distributions to optimize a single scalar property, a shape-conditioned generative model must generate molecules adopting any reasonable shape encoded by the model.

**Challenge 2.** Shape similarity metrics that compute volume overlaps between two molecules (e.g., ROCS) require the molecules to be aligned in 3D space. Unlike 2D similarity, the computed shape similarity between the two molecules will change if one of the structures is rotated. This subtly impacts the learning problem: if the model encodes the target 3D shape into an SE(3)-*invariant* representation, the model must learn how the generated molecule would fit the target shape under the implicit action of an SE(3)-alignment. Alternatively, if the model can natively generate an aligned structure, then the model can more easily learn to construct molecules that fit the target shape.

**Challenge 3.** A molecule's 2D graph topology and 3D shape are highly dependent; small changes in the graph can strikingly alter the shapes accessible to a molecule. It is thus unlikely that a generative model will reliably generate *chemically diverse* molecules with similar shapes to an encoded target without 1) *simultaneous* graph and coordinate generation; and 2) explicit shape-conditioning.

**Challenge 4.** The distribution of shapes a drug-like molecule can adopt is chiefly influenced by rotatable bonds, the foremost source of molecular flexibility. However, existing 3D generative models are mainly developed using tiny molecules (e.g., fewer than 10 heavy atoms), and cannot generate flexible drug-like molecules while maintaining chemical validity (satisfying valencies), geometric validity (non-distorted bond distances and angles; no steric clashes), and chemical diversity.

To surmount these challenges, we design a new generative model, SQUID[1], to enable the shape-conditioned generation of chemically diverse molecules in 3D. Our contributions are as follows:

- Given a 3D molecule with a target shape, we use equivariant point cloud networks to encode the shape into (rotationally) equivariant features. We then use graph neural networks (GNNs) to variationally encode chemical identity into invariant features. By mixing chemical features with equivariant shape features, we can generate diverse molecules in aligned poses that fit the shape.
- We develop a sequential fragment-based 3D generation procedure that fixes local bond lengths and angles to prioritize the scoring of rotatable bonds. By massively simplifying 3D coordinate generation, we generate drug-like molecules while maintaining chemical and geometric validity.
- We design a rotatable bond scoring network that learns how local bond rotations affect global shape, enabling our decoder to generate 3D conformations that best fit the target shape.

We evaluate the utility of SQUID over virtual screening in shape-conditioned 3D molecular design tasks that mimic ligand-based drug design objectives, including shape-conditioned generation of diverse 3D structures and shape-constrained molecular optimization. To inspire further research, we note that our tasks could also be approached with a hypothetical 3D generative model that *disentangles* latent variables controlling 2D chemical identity and 3D shape, thus enabling zero-shot generation of topologically distinct molecules with similar shapes to any encoded target.

---

[1]SQUID: **S**hape-Conditioned **Equi**variant Generator for **D**rug-Like Molecules

## 2 RELATED WORK

**Fragment-based molecular generation.** Seminal works in autoregressive molecular generation applied language models to generate 1D SMILES strings character-by-character (Gómez-Bombarelli et al., 2018; Segler et al., 2018), or GNNs to generate 2D molecular graphs atom-by-atom (Liu et al., 2018; Simonovsky & Komodakis, 2018; Li et al., 2018). Recent works construct molecules fragment-by-fragment to improve the chemical validity of intermediate graphs and to scale generation to larger molecules (Podda et al., 2020; Jin et al., 2019; 2020). Our fragment-based decoder is related to MoLeR (Maziarz et al., 2022), which iteratively generates molecules by selecting a new fragment (or atom) to add to the partial graph, choosing attachment sites on the new fragment, and predicting new bonds to the partial graph. Yet, MoLeR only generates 2D graphs; we generate 3D molecular structures. Beyond 2D generation, Flam-Shepherd et al. (2022) use an RL agent to generate 3D molecules by sampling and connecting molecular fragments. However, they sample from a small multiset of fragments, restricting the accessible chemical space. Powers et al. (2022) use fragments to generate 3D molecules inside a protein pocket, but only consider 7 distinct rings.

**Generation of drug-like molecules in 3D.** In this work, we generate novel drug-like 3D molecular structures in free space, e.g., *not* conformers given a known molecular graph (Ganea et al., 2021; Jing et al., 2022). Myriad models have been proposed to generate small 3D molecules such as E(3)-equivariant normalizing flows and diffusion models (Satorras et al., 2022a; Hoogeboom et al., 2022), RL agents with an SE(3)-covariant action space (Simm et al., 2020b), and autoregressive generators that build molecules atom-by-atom with SE(3)-invariant internal coordinates (Luo & Ji, 2022; Gebauer et al., 2022). However, fewer 3D generative models can generate larger drug-like molecules for realistic chemical design tasks. Of these, Hoogeboom et al. (2022) and Arcidiacono & Koes (2021) fail to generate chemically valid molecules, while Ragoza et al. (2020) rely on post-processing and geometry relaxation to extract stable molecules from their generated atom density grids. Only Roney et al. (2021) and Li et al. (2021), who develop autoregressive generators that simultaneously predict graph structure and internal coordinates, have shown to reliably generate valid drug-like molecules. We also couple graph generation with 3D coordinate prediction; however, we employ fragment-based generation with fixed local geometries to ensure local chemical and geometric validity. Futher, we focus on shape-conditioned molecular design; none of these works can natively address the aforementioned challenges posed by shape-conditioned molecular generation.

**Shape-conditioned molecular generation.** Other works partially address shape-conditioned 3D molecular generation. Skalic et al. (2019) and Imrie et al. (2021) train networks to generate 1D SMILES strings or 2D molecular graphs conditioned on CNN encodings of 3D pharmacophores. However, they do not generate 3D structures, and the CNNs do not respect Euclidean symmetries. Zheng et al. (2021) use supervised molecule-to-molecule translation on SMILES strings for scaffold hopping tasks, but do not generate 3D structures. Papadopoulos et al. (2021) use REINVENT (Olivecrona et al., 2017) on SMILES strings to propose molecules whose conformers are shape-similar to a target, but they must re-optimize the agent for each target shape. Roney et al. (2021) fine-tune a 3D generative model on the hits of a ROCS virtual screen of $> 10^{10}$ drug-like molecules to shift the learned distribution towards a target shape. Yet, this expensive screening approach must be repeated for each new target. Instead, we seek to achieve zero-shot generation of 3D molecules with similar shapes to any encoded shape, without requiring fine-tuning or *post facto* optimization.

**Equivariant geometric deep learning on point clouds.** Various equivariant networks have been designed to encode point clouds for updating coordinates in $\mathbb{R}^3$ (Satorras et al., 2022b), predicting tensorial properties (Thomas et al., 2018), or modeling 3D structures natively in Cartesian space (Fuchs et al., 2020). Especially noteworthy are architectures which lift scalar neuron features to vector features in $\mathbb{R}^3$ and employ simple operations to mix invariant and equivariant features without relying on expensive higher-order tensor products or Clebsch-Gordan coefficients (Deng et al., 2021; Jing et al., 2021). In this work, we employ Deng et al. (2021)'s Vector Neurons (VN)-based equivariant point cloud encoder VN-DGCNN to encode molecules into *equivariant* latent representations in order to generate molecules which are natively aligned to the target shape. Two recent works also employ VN operations for structure-based drug design and linker design (Peng et al., 2022; Huang et al., 2022). Huang et al. (2022) also build molecules in free space; however, they generate just a few atoms to connect existing fragments and do not condition on molecular shape.

## 3  METHODOLOGY

**Problem definition.** We model a conditional distribution $P(M|S)$ over 3D molecules $M = (G, \mathbf{G})$ with graph $G$ and atomic coordinates $\mathbf{G} = \{\mathbf{r}_a \in \mathbb{R}^3\}$ given a 3D molecular shape $S$. Specifically, we aim to sample molecules $M' \sim P(M|S)$ with high shape similarity ($\text{sim}_S(M', M_S) \approx 1$) and low graph (chemical) similarity ($\text{sim}_G(M', M_S) < 1$) to a target molecule $M_S$ with shape $S$. This scheme differs from 1) typical 3D generative models that learn $P(M)$ without modeling $P(M|S)$, and from 2) shape-conditioned 1D/2D generators that attempt to model $P(G|S)$, the distribution of molecular graphs that *could* adopt shape $S$, but do not actually generate specific 3D conformations.

We define graph (chemical) similarity $\text{sim}_G \in [0, 1]$ between two molecules as the Tanimoto similarity computed by RDKit with default settings (2048-bit fingerprints). We define shape similarity $\text{sim}_S^* \in [0, 1]$ using Gaussian descriptions of molecular shape, modeling atoms $a \in M_A$ and $b \in M_B$ from molecules $M_A$ and $M_B$ as isotropic Gaussians in $\mathbb{R}^3$ (Grant & Pickup, 1995; Grant et al., 1996). We compute $\text{sim}_S^*$ using (2-body) volume overlaps between atom-centered Gaussians:

$$\text{sim}_S^*(\mathbf{G}_A, \mathbf{G}_B) = \frac{V_{AB}}{V_{AA} + V_{BB} - V_{AB}}; \quad V_{AB} = \sum_{a \in A, b \in B} V_{ab}; \quad V_{ab} \propto \exp\left(-\frac{\alpha}{2}\|\mathbf{r}_a - \mathbf{r}_b\|^2\right), \quad (1)$$

where $\alpha$ controls the Gaussian width. Setting $\alpha = 0.81$ approximates the shape similarity function used by the ROCS program (App. A.6). $\text{sim}_S^*$ is sensitive to SE(3) transformations of molecule $M_A$ with respect to molecule $M_B$. Thus, we define $\text{sim}_S(M_A, M_B) = \max_{\mathbf{R}, \mathbf{t}} \text{sim}_S^*(\mathbf{G}_A \mathbf{R} + \mathbf{t}, \mathbf{G}_B)$ as the shape similarity when $M_A$ is optimally aligned to $M_B$. We perform such alignments with ROCS.

**Approach.** At a high level, we model $P(M|S)$ with an encoder-decoder architecture. Given a molecule $M_S = (G_S, \mathbf{G}_S)$ with shape $S$, we encode $S$ (a point cloud) into equivariant features. We then *variationally* encode $G_S$ into atomic features, conditioned on the shape features. We then mix these shape and atom features to pass global SE(3) {in,equi}variant latent codes to the decoder, which samples new molecules from $P(M|S)$. We autoregressively generate molecules by factoring $P(M|S) = P(M_0|S)P(M_1|M_0, S)...P(M|M_{n-1}, S)$, where each $M_l = (G_l, \mathbf{G}_l)$ are partial molecules defined by a BFS traversal of a tree-representation of the molecular graph (Fig. 2). Tree-nodes denote either non-ring atoms or rigid (ring-containing) fragments, and tree-links denote acyclic (rotatable, double, or triple) bonds. We generate $M_{l+1}$ by growing the graph $G_{l+1}$ around a focus atom/fragment, and then predict $\mathbf{G}_{l+1}$ by scoring a query rotatable bond to best fit shape $S$.

**Simplifying assumptions.** (1) We ignore hydrogens and only consider heavy atoms, as is common in molecular generation. (2) We only consider molecules with fragments present in our fragment library to ensure that graph generation can be expressed as tree generation. (3) Rather than generating all coordinates, we use rigid fragments, fix bond distances, and set bond angles according to hybridization heuristics (App. A.8); this lets the model focus on scoring rotatable bonds to best fit the growing conformer to the encoded shape. (4) We seed generation with $M_0$ (the root tree-node), restricted to be a small (3-6 atoms) substructure from $M_S$; hence, we only model $P(M|S, M_0)$.

### 3.1  ENCODER

**Featurization.** We construct a molecular graph $G$ using atoms as nodes and bonds as edges. We featurize each node with the atomic mass; one-hot codes of atomic number, charge, and aromaticity; and one-hot codes of the number of single, double, aromatic, and triple bonds the atom forms (including bonds to implicit hydrogens). This helps us fix bond angles during generation (App. A.8). We featurize each edge with one-hot codes of bond order. We represent a shape $S$ as a point cloud built by sampling $n_p$ points from each of $n_h$ atom-centered Gaussians with (adjustable) variance $\sigma_p^2$.

**Fragment encoder.** We also featurize each node with a learned embedding $\mathbf{f}_i \in \mathbb{R}^{d_f}$ of the atom/fragment type to which that atom belongs, making each node "fragment-aware" (similar to MoLeR). In principle, fragments could be any rigid substructure with $\geq 2$ atoms. Here, we specify fragments as ring-containing substructures without acyclic single bonds (Fig. 14). We construct a library $\mathcal{L}_f$ of atom/fragment types by extracting the top-$k$ ($k = 100$) most frequent fragments from the dataset and adding these, along with each distinct atom type, to $\mathcal{L}_f$ (App. A.13). We then encode each atom/fragment in $\mathcal{L}_f$ with a simple GNN (App. A.12) to yield the global atom/fragment embeddings: $\{\mathbf{f}_i = \sum_a \mathbf{h}_{f_i}^{(a)}, \{\mathbf{h}_{f_i}^{(a)}\} = \text{GNN}_{\mathcal{L}_f}(G_{f_i}) \ \forall f_i \in \mathcal{L}_f\}$, where $\mathbf{h}_{f_i}^{(a)}$ are per-atom features.

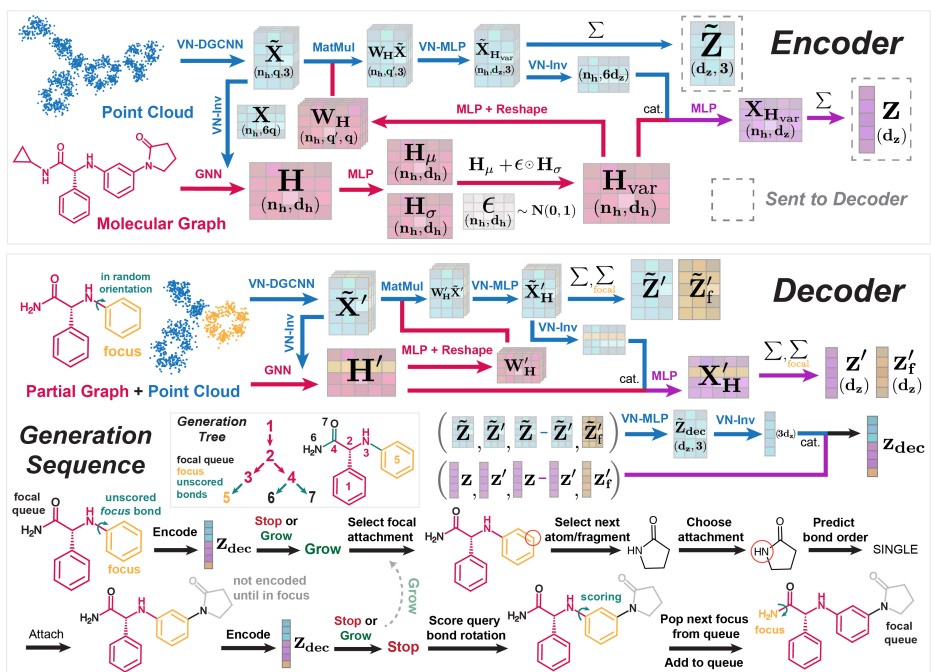

Figure 2: Encoder-decoder architecture of SQUID, which equivariantly encodes molecular shape and variationally encodes chemical identity to generate chemically diverse 3D molecules that fit the shape. SQUID generates molecules atom-by-atom and fragment-by-fragment, iteratively growing the molecular graph in a tree-expansion and generating 3D coordinates by scoring rotatable bonds.

**Shape encoder.** Given $M_S$ with $n_h$ heavy atoms, we use VN-DGCNN (App. A.11) to encode the molecular point cloud $\mathbf{P}_S \in \mathbb{R}^{(n_h n_p) \times 3}$ into a set of equivariant per-point vector features $\tilde{\mathbf{X}}_p \in \mathbb{R}^{(n_h n_p) \times q \times 3}$. We then locally mean-pool the $n_p$ equivariant features per atom:

$$\tilde{\mathbf{X}}_p = \text{VN-DGCNN}(\mathbf{P}_S); \quad \tilde{\mathbf{X}} = \text{LocalPool}(\tilde{\mathbf{X}}_p), \tag{2}$$

where $\tilde{\mathbf{X}} \in \mathbb{R}^{n_h \times q \times 3}$ are per-atom equivariant representations of the molecular shape. Because VN operations are SO(3)-equivariant, rotating the point cloud will rotate $\tilde{\mathbf{X}}$: $\tilde{\mathbf{X}}\mathbf{R} = \text{LocalPool}(\text{VN-DGCNN}(\mathbf{P}_S\mathbf{R}))$. Although VN operations are strictly SO(3)-equivariant, we subtract the molecule's centroid from the atomic coordinates prior to encoding, making $\tilde{\mathbf{X}}$ effectively SE(3)-equivariant. Throughout this work, we denote SO(3)-equivariant vector features with tildes.

**Variational graph encoder.** To model $P(M|S)$, we first use a GNN (App. A.12) to encode $G_S$ into learned atom embeddings $\mathbf{H} = \{\mathbf{h}^{(a)} \,\forall\, a \in G_S\}$. We condition the GNN on per-atom *invariant* shape features $\mathbf{X} = \{\mathbf{x}^{(a)}\} \in \mathbb{R}^{n_h \times 6q}$, which we form by passing $\tilde{\mathbf{X}}$ through a VN-Inv (App. A.11):

$$\mathbf{H} = \text{GNN}((\mathbf{H}_0, \mathbf{X}); G_S); \quad \mathbf{X} = \text{VN-Inv}(\tilde{\mathbf{X}}), \tag{3}$$

where $\mathbf{H}_0 \in \mathbb{R}^{n_h \times (d_a + d_f)}$ are the set of initial atom features concatenated with the learned fragment embeddings, $\mathbf{H} \in \mathbb{R}^{n_h \times d_h}$, and $(\cdot, \cdot)$ denotes concatenation in the feature dimension. For each atom in $M_S$, we then encode $\mathbf{h}^{(a)}_\mu, \mathbf{h}^{(a)}_{\log \sigma^2} = \text{MLP}(\mathbf{h}^{(a)})$ and sample $\mathbf{h}^{(a)}_{\text{var}} \sim N(\mathbf{h}^{(a)}_\mu, \mathbf{h}^{(a)}_\sigma)$:

$$\mathbf{H}_{\text{var}} = \left\{ \mathbf{h}^{(a)}_{\text{var}} = \mathbf{h}^{(a)}_\mu + \boldsymbol{\epsilon}^{(a)} \odot \mathbf{h}^{(a)}_\sigma; \;\; \mathbf{h}^{(a)}_\sigma = \exp(\tfrac{1}{2}\mathbf{h}^{(a)}_{\log \sigma^2}) \;\; \forall \;\; a \in G_S \right\}, \tag{4}$$

where $\boldsymbol{\epsilon}^{(a)} \sim N(\mathbf{0}, \mathbf{1}) \in \mathbb{R}^{d_h}$, $\mathbf{H}_{\text{var}} \in \mathbb{R}^{n_h \times d_h}$, and $\odot$ denotes elementwise multiplication. Here, the second argument of $N(\cdot, \cdot)$ is the standard deviation vector of the diagonal covariance matrix.

**Mixing shape and variational features.** The variational atom features $\mathbf{H}_{\text{var}}$ are insensitive to rotations of $S$. However, we desire the decoder to construct molecules in poses that are natively aligned to $S$ (Challenge 2). We achieve this by conditioning the decoder on an *equivariant* latent representation of $P(M|S)$ that mixes both shape and chemical information. Specifically, we mix $\mathbf{H}_{\text{var}}$ with

$\tilde{\mathbf{X}}$ by encoding each $\mathbf{h}_{\text{var}}^{(a)} \in \mathbf{H}_{\text{var}}$ into linear transformations, which are applied atom-wise to $\tilde{\mathbf{X}}$. We then pass the mixed equivariant features through a separate VN-MLP (App. A.11):

$$\tilde{\mathbf{X}}_{\mathbf{H}_{\text{var}}} = \left\{ \text{VN-MLP}(\mathbf{W}_{\mathbf{H}}^{(a)} \tilde{\mathbf{X}}^{(a)}, \tilde{\mathbf{X}}^{(a)}); \ \ \mathbf{W}_{\mathbf{H}}^{(a)} = \text{Reshape}(\text{MLP}(\mathbf{h}_{\text{var}}^{(a)})) \ \forall \ a \in G_S \right\}, \quad (5)$$

where $\mathbf{W}_{\mathbf{H}}^{(a)} \in \mathbb{R}^{q' \times q}$, $\tilde{\mathbf{X}}^{(a)} \in \mathbb{R}^{q \times 3}$, and $\tilde{\mathbf{X}}_{\mathbf{H}_{\text{var}}} \in \mathbb{R}^{n_h \times d_z \times 3}$. This maintains equivariance since $\mathbf{W}_{\mathbf{H}}^{(a)}$ are rotationally invariant and $\mathbf{W}_{\mathbf{H}}^{(a)}(\tilde{\mathbf{X}}^{(a)}\mathbf{R}) = (\mathbf{W}_{\mathbf{H}}^{(a)}\tilde{\mathbf{X}}^{(a)})\mathbf{R}$ for a rotation $\mathbf{R}$. Finally, we sum-pool the per-atom features in $\tilde{\mathbf{X}}_{\mathbf{H}_{\text{var}}}$ into a global equivariant representation $\tilde{\mathbf{Z}} \in \mathbb{R}^{d_z \times 3}$. We also embed a global invariant representation $\mathbf{z} \in \mathbb{R}^{d_z}$ by applying a VN-Inv to $\tilde{\mathbf{X}}_{\mathbf{H}_{\text{var}}}$, concatenating the output with $\mathbf{H}_{\text{var}}$, passing through an MLP, and sum-pooling the resultant per-atom features:

$$\tilde{\mathbf{Z}} = \sum_a \tilde{\mathbf{X}}_{\mathbf{H}_{\text{var}}}^{(a)}; \ \ \mathbf{z} = \sum_a \text{MLP}(\mathbf{x}_{\mathbf{H}_{\text{var}}}^{(a)}, \mathbf{h}_{\text{var}}^{(a)}); \ \ \mathbf{x}_{\mathbf{H}_{\text{var}}}^{(a)} = \text{VN-Inv}(\tilde{\mathbf{X}}_{\mathbf{H}_{\text{var}}}^{(a)}). \quad (6)$$

### 3.2 DECODER

Given $M_S$, we sample new molecules $M' \sim P(M|S, M_0)$ by encoding $\mathbf{P}_S$ into equivariant shape features $\tilde{\mathbf{X}}$, variationally sampling $\mathbf{h}_{\text{var}}^{(a)}$ for each atom in $M_S$, mixing $\mathbf{H}_{\text{var}}$ with $\tilde{\mathbf{X}}$, and passing the resultant $(\tilde{\mathbf{Z}}, \mathbf{z})$ to the decoder. We seed generation with a small structure $M_0$ (extracted from $M_S$), and build $M'$ by sequentially generating larger structures $M'_{l+1}$ in a tree-like manner (Fig. 2). Specifically, we grow new atoms/fragments around a "focus" atom/fragment in $M'_l$, which is popped from a BFS queue. To generate $M'_{l+1}$ from $M'_l$ (e.g., grow the tree from the focus), we factor $P(M_{l+1}|M_l, S) = P(G_{l+1}|M_l, S)P(\mathbf{G}_{l+1}|G_{l+1}, M_l, S)$. Given $(\tilde{\mathbf{Z}}, \mathbf{z})$, we sample the new graph $G'_{l+1}$ by iteratively attaching (a variable) $C$ new atoms/fragments (children tree-nodes) around the focus, yielding $G_l'^{(c)}$ for $c = 1, ..., C$, where $G_l'^{(C)} = G'_{l+1}$ and $G_l'^{(0)} = G'_l$. We then generate coordinates $\mathbf{G}'_{l+1}$ by scoring the (rotatable) bond between the focus and its parent tree-node. New bonds from the focus to its children are left unscored in $M'_{l+1}$ until the children become "in focus".

**Partial molecule encoder.** Before bonding each new atom/fragment to the focus (or scoring bonds), we encode the partial molecule $M_l'^{(c-1)}$ with the same scheme as for $M_S$ (using a parallel encoder; Fig. 2), except we do not variationally embed $\mathbf{H}'$.[2] Instead, we process $\mathbf{H}'$ analogously to $\mathbf{H}_{\text{var}}$. Further, in addition to globally pooling the per-atom embeddings to obtain $\tilde{\mathbf{Z}}' = \sum_a \tilde{\mathbf{X}}_{\mathbf{H}}'^{(a)}$ and $\mathbf{z}' = \sum_a \mathbf{x}_{\mathbf{H}}'^{(a)}$, we also selectively sum-pool the embeddings of the atom(s) in focus, yielding $\tilde{\mathbf{Z}}'_{\text{foc}} = \sum_{a \in \text{focus}} \tilde{\mathbf{X}}_{\mathbf{H}}'^{(a)}$ and $\mathbf{z}'_{\text{foc}} = \sum_{a \in \text{focus}} \mathbf{x}_{\mathbf{H}}'^{(a)}$. We then align the equivariant representations of $M_l'^{(c-1)}$ and $M_S$ by concatenating $\tilde{\mathbf{Z}}$, $\tilde{\mathbf{Z}}'$, $\tilde{\mathbf{Z}} - \tilde{\mathbf{Z}}'$, and $\tilde{\mathbf{Z}}'_{\text{foc}}$ and passing these through a VN-MLP:

$$\tilde{\mathbf{Z}}_{\text{dec}} = \text{VN-MLP}(\tilde{\mathbf{Z}}, \tilde{\mathbf{Z}}', \tilde{\mathbf{Z}} - \tilde{\mathbf{Z}}', \tilde{\mathbf{Z}}'_{\text{foc}}). \quad (7)$$

Note that $\tilde{\mathbf{Z}}_{\text{dec}} \in \mathbb{R}^{q \times 3}$ is equivariant to rotations of the overall system $(M_l'^{(c-1)}, M_S)$. Finally, we form a global invariant feature $\mathbf{z}_{\text{dec}} \in \mathbb{R}^{d_{\text{dec}}}$ to condition graph (or coordinate) generation:

$$\mathbf{z}_{\text{dec}} = (\text{VN-Inv}(\tilde{\mathbf{Z}}_{\text{dec}}), \mathbf{z}, \mathbf{z}', \mathbf{z} - \mathbf{z}', \mathbf{z}'_{\text{foc}}). \quad (8)$$

**Graph generation.** We factor $P(G_{l+1}|M_l, S)$ into a sequence of generation steps by which we iteratively connect children atoms/fragments to the focus until the network generates a (local) stop token. Fig. 2 sketches a generation sequence by which a new atom/fragment is attached to the focus, yielding $G_l'^{(c)}$ from $G_l'^{(c-1)}$. Given $\mathbf{z}_{\text{dec}}$, the model first predicts whether to stop (local) generation via $p_\emptyset = \text{sigmoid}(\text{MLP}_\emptyset(\mathbf{z}_{\text{dec}})) \in (0, 1)$. If $p_\emptyset \geq \tau_\emptyset$ (a threshold, App. A.16), we stop and proceed to bond scoring. Otherwise, we select which atom $a_{\text{foc}}$ on the focus (if multiple) to grow from:

$$\mathbf{p}_{\text{focus}} = \text{softmax}(\{\text{MLP}_{\text{focus}}(\mathbf{z}_{\text{dec}}, \mathbf{x}_{\mathbf{H}}'^{(a)}) \ \forall \ a \in \text{focus}\}). \quad (9)$$

The decoder then predicts which atom/fragment $f_{\text{next}} \in \mathcal{L}_f$ to connect to the focus next:

$$\mathbf{p}_{\text{next}} = \text{softmax}(\{\text{MLP}_{\text{next}}(\mathbf{z}_{\text{dec}}, \mathbf{x}_{\mathbf{H}}'^{(a_{\text{foc}})}, \mathbf{f}_{f_i}) \ \forall \ f_i \in \mathcal{L}_f\}). \quad (10)$$

---

[2]We have dropped the $(c)$ notation for clarity. However, each $\mathbf{Z}_{\text{dec}}$ is specific to each $(M_l'^{(c-1)}, M_S)$ system.

If the selected $f_{\text{next}}$ is a fragment, we predict the attachment site $a_{\text{site}}$ on the fragment $G_{f_{\text{next}}}$:

$$\mathbf{p}_{\text{site}} = \text{softmax}(\{\text{MLP}_{\text{site}}(\mathbf{z}_{\text{dec}}, \mathbf{x}_{\mathbf{H}}'^{(a_{\text{foc}})}, \mathbf{f}_{\text{next}}, \mathbf{h}_{f_{\text{next}}}^{(a)}) \,\forall\, a \in G_{f_{\text{next}}}\}), \tag{11}$$

where $\mathbf{h}_{f_{\text{next}}}^{(a)}$ are the encoded atom features for $G_{f_{\text{next}}}$. Lastly, we predict the bond order ($1°$, $2°$, $3°$) via $\mathbf{p}_{\text{bond}} = \text{softmax}(\text{MLP}_{\text{bond}}(\mathbf{z}_{\text{dec}}, \mathbf{x}_{\mathbf{H}}'^{(a_{\text{foc}})}, \mathbf{f}_{\text{next}}, \mathbf{h}_{f_{\text{next}}}^{(a_{\text{site}})}))$. We repeat this sequence of steps until $p_\emptyset \geq \tau_\emptyset$, yielding $G_{l+1}$. At each step, we greedily select the action after masking actions that violate known chemical valence rules. After each sequence, we bond a new atom or fragment to the focus, giving $G_l'^{(c)}$. If an atom, the atom's position relative to the focus is fixed by heuristic bonding geometries (App. A.8). If a fragment, the position of the attachment site is fixed, but the *dihedral* of the new bond is yet unknown. Thus, in subsequent generation steps we only encode the attachment site and mask the remaining atoms in the new fragment until that fragment is "in focus" (Fig. 2). This means that *prior to bond scoring*, the rotation angle of the focus is random. To account for this when training (with teacher forcing), we randomize the focal dihedral when encoding each $M_l'^{(c-1)}$.

**Scoring rotatable bonds.** After sampling $G_{l+1}' \sim P(G_{l+1}|M_l', S)$, we generate $\mathbf{G}_{l+1}'$ by scoring the rotation angle $\psi_{l+1}'$ of the bond connecting the focus to its parent node in the generation tree (Fig. 2). Since we ultimately seek to maximize $\text{sim}_S(M', M_S)$, we exploit the fact that our model generates shape-aligned structures to predict $\max\limits_{\psi_{l+2}', \psi_{l+3}', \dots} \text{sim}_S^*(\mathbf{G}'^{(\psi_{\text{foc}})}, \mathbf{G}_S)$ for various query dihedrals $\psi_{l+1}' = \psi_{\text{foc}}$ of the focus rotatable bond in a supervised regression setting. Intuitively, the scorer is trained to predict how the choice of $\psi_{\text{foc}}$ affects the maximum possible shape similarity of the final molecule $M'$ to the target $M_S$ under an optimal policy. App. A.2 details how regression targets are computed. During generation, we sweep over each query $\psi_{\text{foc}} \in [-\pi, \pi)$, encode each resultant structure $M_{l+1}'^{(\psi_{\text{foc}})}$ into $\mathbf{z}_{\text{dec, scorer}}^{(\psi_{\text{foc}})}$ [3], and select the $\psi_{\text{foc}}$ that maximizes the predicted score:

$$\psi_{l+1}' = \arg\max\limits_{\psi_{\text{foc}}} \text{sigmoid}(\text{MLP}_{\text{scorer}}(\mathbf{z}_{\text{dec, scorer}}^{(\psi_{\text{foc}})})). \tag{12}$$

At generation time, we also score chirality by enumerating stereoisomers $G_{\text{foc}}^\chi \in G_{\text{foc}}'$ of the focus and selecting the $(G_{\text{foc}}^\chi, \psi_{\text{foc}})$ that maximizes Eq. 12 (App. A.2).

**Training.** We supervise each step of graph generation with a multi-component loss function:

$$L_{\text{graph-gen}} = L_\emptyset + L_{\text{focus}} + L_{\text{next}} + L_{\text{site}} + L_{\text{bond}} + \beta_{KL} L_{KL} + \beta_{\text{next-shape}} L_{\text{next-shape}} + \beta_{\emptyset\text{-shape}} L_{\emptyset\text{-shape}}. \tag{13}$$

$L_\emptyset$, $L_{\text{focus}}$, $L_{\text{next}}$, and $L_{\text{bond}}$ are standard cross-entropy losses. $L_{\text{site}} = -\log(\sum_a p_{\text{site}}^{(a)} \mathbb{I}[c_a > 0])$ is a modified cross-entropy loss that accounts for symmetric attachment sites in the fragments $G_{f_i} \in \mathcal{L}_f$, where $p_{\text{site}}^{(a)}$ are the predicted attachment-site probabilities and $c_a$ are multi-hot class probabilities. $L_{KL}$ is the KL-divergence between the learned $\mathcal{N}(\mathbf{h}_\mu, \mathbf{h}_\sigma)$ and the prior $\mathcal{N}(\mathbf{0}, \mathbf{1})$. We also employ two auxiliary losses $L_{\text{next-shape}}$ and $L_{\emptyset\text{-shape}}$ in order to 1) help the generator distinguish between incorrect shape-similar (near-miss) vs. shape-dissimilar fragments, and 2) encourage the generator to generate structures that fill the entire target shape (App. A.10). We train the rotatable bond scorer separately from the generator with an MSE regression loss. See App. A.15 for training details.

## 4 EXPERIMENTS

**Dataset.** We train SQUID with drug-like molecules (up to $n_h = 27$) from MOSES (Polykovskiy et al., 2020) using their train/test sets. $\mathcal{L}_f$ includes 100 fragments extracted from the dataset and 24 atom types. We remove molecules that contain excluded fragments. For remaining molecules, we generate a 3D conformer with RDKit, set acyclic bond distances to their empirical means, and fix acyclic bond angles using heuristic rules. While this 3D manipulation neglects distorted bonding geometries in real molecules, the *global* shapes are marginally impacted, and we may recover refined geometries without seriously altering the shape (App. A.8). The final dataset contains 1.3M 3D molecules, partitioned into 80/20 train/validation splits. The test set contains 147K 3D molecules.

---

[3]We train the scorer independently from the graph generator, but with a parallel architecture. Hence, $\mathbf{z}_{\text{dec}} \neq \mathbf{z}_{\text{dec, scorer}}$. The main architectural difference between the two models (graph generator and scorer) is that we do not variationally encode $\mathbf{H}_{\text{scorer}}$ into $\mathbf{H}_{\text{var,scorer}}$, as we find it does not impact empirical performance.

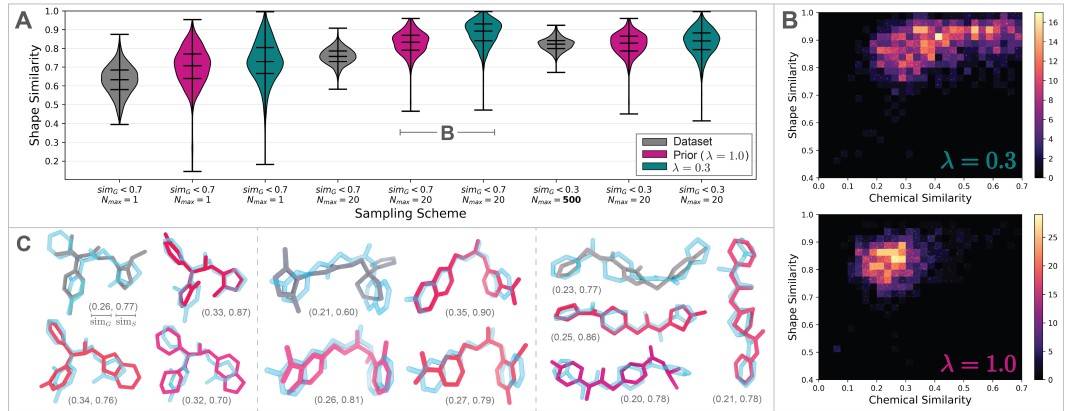

Figure 3: (A) Distributions of $\text{sim}_S(M, M_S)$ for the best of $N_{\max}$ molecules with $\text{sim}_G(M, M_S) <$ 0.7 or 0.3 when sampling from the dataset (grey) or SQUID with either $\lambda = 1.0$ (prior; red) or $\lambda = 0.3$ (green). (B) Histograms of $\text{sim}_S$ vs. $\text{sim}_G$ for two sampling schemes from (A). (C) Three sets of samples from the prior (red) and the dataset ($N_{\max} = 20$; grey), overlaid on $M_S$ (blue).

In the following experiments, we only consider molecules $M_S$ for which we can extract a small (3-6 atoms) 3D substructure $M_0$ containing a terminal atom, which we use to seed generation. In principle, $M_0$ could include larger structures from $M_S$, e.g., for scaffold-constrained tasks. Here, we use the smallest substructures to ensure that the shape-conditioned generation tasks are not trivial.

**Shape-conditioned generation of chemically diverse molecules.** "Scaffold-hopping"—designing molecules with high 3D shape similarity but novel 2D graph topology compared to known inhibitors—is pursued in LBDD to develop chemical lead series, optimize drug activity, or evade intellectual property restrictions (Hu et al., 2017). We imitate this task by evaluating SQUID's ability to generate molecules $M'$ with high $\text{sim}_S(M', M_S)$ but low $\text{sim}_G(M', M_S)$. Specifically, for 1000 molecules $M_S$ with target shapes $S$ in the test set, we use SQUID to generate 50 molecules per $M_S$. To generate chemically diverse species, we linearly interpolate between the posterior $N(\mathbf{h}_\mu, \mathbf{h}_\sigma)$ and the prior $N(\mathbf{0}, \mathbf{1})$, sampling each $\mathbf{h}_{\text{var}} \sim N((1 - \lambda)\mathbf{h}_\mu, (1 - \lambda)\mathbf{h}_\sigma + \lambda\mathbf{1})$ using either $\lambda = 0.3$ or $\lambda = 1.0$ (prior). We then filter the generated molecules to have $\text{sim}_G(M', M_S) < 0.7$, or $< 0.3$ to only evaluate molecules with substantial chemical differences compared to $M_S$. Of the filtered molecules, we randomly choose $N_{\max}$ samples and select the sample with highest $\text{sim}_S(M', M_S)$.

Figure 3A plots distributions of $\text{sim}_S(M', M_S)$ between the selected molecules and their respective target shapes, using different sampling ($N_{\max} = 1, 20$) and filtering ($\text{sim}_G(M', M_S) < 0.7, 0.3$) schemes. We compare against analogously sampling random 3D molecules from the training set. Overall, SQUID generates diverse 3D molecules that are quantitatively enriched in shape similarity compared to molecules sampled from the dataset, particularly for $N_{\max} = 20$. Qualitatively, the molecules generated by SQUID have significantly more atoms which directly overlap with the atoms of $M_S$, even in cases where the computed shape similarity is comparable between SQUID-generated molecules and molecules sampled from the dataset (Fig. 3C). We quantitatively explore this observation in App. A.7. We also find that using $\lambda = 0.3$ yields greater $\text{sim}_S(M', M_S)$ than $\lambda = 1.0$, in part because using $\lambda = 0.3$ yields less chemically diverse molecules (Fig. 3B; Challenge 3). Even so, sampling $N_{\max} = 20$ molecules from the prior with $\text{sim}_G(M', M_S) < 0.3$ still yields more shape-similar molecules than sampling $N_{\max} = 500$ molecules from the dataset. We emphasize that 99% of samples from the prior are novel, 95% are unique, and 100% are chemically valid (App. A.4). Moreover, 87% of generated structures do not have any steric clashes (App. A.4), indicating that SQUID generates realistic 3D geometries of the flexible drug-like molecules.

**Ablating equivariance.** SQUID's success in 3D shape-conditioned molecular generation is partly attributable to SQUID aligning the generated structures to the target shape in *equivariant* feature space (Eq. 7), which enables SQUID to generate 3D structures that fit the target shape without having to implicitly learn how to align two structures in $\mathbb{R}^3$ (Challenge 2). We explicitly validate this design choice by setting $\tilde{\mathbf{Z}} = \mathbf{0}$ in Eq. 7, which prevents the decoder from accessing the 3D orientation of $M_S$ during training/generation. As expected, ablating SQUID's equivariance reduces the enrichment in shape similarity (relative to the dataset baseline) by as much as 33% (App. A.9).

Table 1: Top-1 optimized objective scores across 6 objectives and 8 seed molecules $M_S$ (per objective), under the shape-similarity constraint $\text{sim}_S(M^*, M_S) \geq 0.85$. $(-)$ indicates that SQUID, limited to 31K generated samples, or virtual screening (VS) could not improve upon $M_S$.

| Objective (↑) | Method | Top-Scoring Seed Molecules | | | Top-Scoring (large) Seed Molecules | | | Random (large) Seed Molecules | |
|---|---|---|---|---|---|---|---|---|---|
| | | A | B | C | D | E | F | G | H |
| GSK3B | Seed | 0.69 | 0.69 | 0.61 | 0.48 | 0.50 | 0.42 | 0.08 | 0.02 |
| | SQUID | – | 0.70 | – | **0.52** | **0.68** | **0.66** | **0.46** | **0.53** |
| | VS | – | **0.90** | – | – | – | 0.59 | 0.10 | 0.39 |
| JNK3 | Seed | 0.86 | 0.46 | 0.46 | 0.27 | 0.31 | 0.27 | 0.00 | 0.00 |
| | SQUID | **0.95** | **0.63** | **0.63** | **0.36** | – | **0.32** | **0.23** | **0.25** |
| | VS | 0.91 | – | – | – | – | – | 0.03 | 0.21 |
| Osimertinib MPO | Seed | 0.81 | 0.80 | 0.79 | 0.80 | 0.76 | 0.75 | 0.08 | 0.09 |
| | SQUID | **0.83** | – | **0.82** | **0.83** | **0.80** | **0.80** | **0.77** | 0.78 |
| | VS | – | – | – | – | 0.79 | – | 0.46 | 0.78 |
| Sitagliptin MPO | Seed | 0.45 | 0.44 | 0.44 | 0.25 | 0.23 | 0.24 | 0.00 | 0.00 |
| | SQUID | – | **0.45** | – | **0.32** | **0.33** | **0.39** | **0.30** | 0.22 |
| | VS | – | – | – | – | 0.28 | 0.33 | 0.23 | **0.37** |
| Celecoxib Rediscovery | Seed | 0.43 | 0.42 | 0.41 | 0.33 | 0.34 | 0.33 | 0.15 | 0.21 |
| | SQUID | – | **0.43** | 0.42 | 0.35 | **0.38** | 0.36 | 0.26 | 0.24 |
| | VS | – | – | **0.45** | **0.40** | – | **0.42** | 0.26 | **0.37** |
| Thiothixene Rediscovery | Seed | 0.42 | 0.39 | 0.37 | 0.29 | 0.27 | 0.30 | 0.19 | 0.19 |
| | SQUID | **0.52** | **0.46** | 0.42 | – | **0.31** | 0.32 | **0.26** | 0.27 |
| | VS | – | 0.40 | 0.42 | – | 0.30 | 0.32 | 0.22 | **0.32** |

**Shape-constrained molecular optimization.** Scaffold-hopping is often goal-directed; e.g., aiming to reduce toxicity or improve bioactivity of a hit compound without altering its 3D shape. We mimic this shape-constrained MO setting by applying SQUID to optimize objectives from GaucaMol (Brown et al., 2019) while preserving high shape similarity ($\text{sim}_S(M, M_S) \geq 0.85$) to various "hit" 3D molecules $M_S$ from the test set. This task considerably differs from typical MO tasks, which optimize objectives *without* constraining 3D shape and *without* generating 3D structures.

To adapt SQUID to shape-constrained MO, we implement a genetic algorithm (App. A.5) that iteratively mutates the variational atom embeddings $\mathbf{H}_{\text{var}}$ of encoded seed molecules ("hits") $M_S$ in order to generate 3D molecules $M^*$ with improved objective scores, but which still fit the shape of $M_S$. Table 1 reports the optimized top-1 scores across 6 objectives and 8 seed molecules $M_S$ (per objective, sampled from the test set), constrained such that $\text{sim}_S(M^*, M_S) \geq 0.85$. We compare against the score of $M_S$, as well as the (shape-constrained) top-1 score obtained by virtual screening (VS) our training dataset (>1M 3D molecules). Of the 8 seeds $M_S$ per objective, 3 were selected from top-scoring molecules to serve as hypothetical "hits", 3 were selected from top-scoring *large* molecules ($\geq 26$ heavy atoms), and 2 were randomly selected from all large molecules.

In 40/48 tasks, SQUID improves the objective score of the seed $M_S$ while maintaining $\text{sim}_S(M^*, M_S) \geq 0.85$. Qualitatively, SQUID optimizes the objectives through chemical alterations such as adding/deleting individual atoms, switching bonding patterns, or replacing entire substructures – all while generating 3D structures that fit the target shape (App. A.5). In 29/40 of successful cases, SQUID (limited to 31K samples) surpasses the baseline of virtual screening 1M molecules, demonstrating the ability to efficiently explore new shape-constrained chemical space.

## 5 CONCLUSION

We designed a novel 3D generative model, SQUID, to enable shape-conditioned exploration of chemically diverse molecular space. SQUID generates realistic 3D geometries of larger molecules that are chemically valid, and uniquely exploits equivariant operations to construct conformations that fit a target 3D shape. We envision our model, alongside future work, will advance creative shape-based drug design tasks such as 3D scaffold hopping and shape-constrained 3D ligand design.

## REPRODUCIBILITY STATEMENT

We have taken care to facilitate the reproduciblility of this work by detailing the precise architecture of SQUID throughout the main text; we also provide extensive details on training protocols, model parameters, and further evaluations in the Appendices. Our source code can be found at `https://github.com/keiradams/SQUID`. Beyond the model implementation, our code includes links to access our datasets, as well as scripts to process the training dataset, train the model, and evaluate our trained models across the shape-conditioned generation and shape-constrained optimization tasks described in this paper.

## ETHICS STATEMENT

Advancing the shape-conditioned 3D generative modeling of drug-like molecules has the potential to accelerate pharmaceutical drug design, showing particular promise for drug discovery campaigns involving scaffold hopping, hit expansion, or the discovery of novel ligand analogues. However, such advancements could also be exploited for nefarious pharmaceutical research and harmful biological applications.

## ACKNOWLEDGMENTS

This research was supported by the Office of Naval Research under grant number N00014-21-1-2195. This material is based upon work supported by the National Science Foundation Graduate Research Fellowship under Grant No. 2141064. The authors acknowledge the MIT SuperCloud and Lincoln Laboratory Supercomputing Center for providing HPC resources that have contributed to the research results reported within this poster. The authors thank Rocío Mercado, Sam Goldman, Wenhao Gao, and Lagnajit Pattanaik for providing helpful suggestions regarding the content and presentation of this paper.

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

# A   APPENDIX

## CONTENTS

## A.1 Overview of definitions, terms, and notations

Table 2: List of definitions, terms, and notations

| Symbol | Description |
|--------|-------------|
| $M$ | a 3D molecule |
| $G$ | a 2D molecular graph |
| $\mathbf{G}$ | 3D coordinates of a molecule with graph $G$ |
| $\mathbf{r}_a$ | coordinates of atom $a$ in $\mathbb{R}^3$ |
| $S$ | a molecular shape |
| $M_S$ | an encoded 3D molecule with target shape $S$ |
| $M'$ | a generated/sampled 3D molecule |
| $M^*$ | an objective-optimized 3D molecule |
| $\mathrm{sim}_G(M', M_S)$ | graph (chemical) similarity between molecules $M'$ and $M_S$ |
| $\mathrm{sim}_S^*(\mathbf{G}_A, \mathbf{G}_B)$ | rotationally-*sensitive* shape similarity of $\mathbf{G}_A$ and $\mathbf{G}_B$ |
| $\mathrm{sim}_S(M_A, M_B)$ | shape similarity of molecules $M_A$ and $M_B$ after $M_A$ is optimally aligned to $M_B$ (with ROCS) |
| $\alpha$ | parameter controlling the width of the atom-centered Gaussians when computing $\mathrm{sim}_S(M_A, M_B)$ |
| $P(M\|S)$ | conditional distribution of 3D molecules given a molecular shape $S$ |
| $M_l$ | partial 3D molecule (e.g., during generation) |
| $G_l$ | partial molecular graph (e.g., during generation) |
| $M_0$ | small 3D substructure used to seed generation |
| $M_l'^{(c)}$ | partial 3D molecule after bonding $c$ atoms/fragments to the focus |
| $G_l'^{(c)}$ | partial molecular graph after bonding $c$ atoms/fragments to the focus |
| $n_h$ | number of heavy atoms |
| $n_p$ | number of sampled points per heavy atom |
| $\sigma_p^2$ | variance of isotropic atom-centered Gaussians in $\mathbb{R}^3$ |
| $\mathcal{L}_f$ | the atom/fragment library |
| $f_i$ | an atom/fragment in $\mathcal{L}_f$ |
| $\mathbf{f}_i$ | the learned embedding of atom/fragment $f_i \in \mathcal{L}_f$ |
| $\mathbf{h}^{(a)}$ | learned embedding of atom $a$ |
| $\mathbf{P}_S$ | point cloud representation of shape $S$ |
| $\mathbf{R}$ | an arbitrary rotation matrix in $\mathbb{R}^{3\times3}$ |
| SO(3) | the special orthogonal group (rotations) in $\mathbb{R}^3$ |
| SE(3) | the special Euclidean group (rotations and translations) in $\mathbb{R}^3$ |
| $\tilde{\mathbf{X}}_p$ | tensor containing the learned equivariant features of each point in a point cloud; $\tilde{\mathbf{X}}_p \in \mathbb{R}^{(n_h n_p)\times q\times 3}$ |
| $\tilde{\mathbf{X}}$ | tensor containing the learned equivariant features of each atom in the encoded 3D molecule; $\tilde{\mathbf{X}} \in \mathbb{R}^{n_h\times q\times 3}$ |
| $q$ | (1st) dimensionality of vector features $\tilde{\mathbf{X}}, \tilde{\mathbf{X}}_p$ |
| $\mathbf{H}$ | matrix containing the learned atom embeddings; $\mathbf{H} = \{\mathbf{h}^{(a)} \forall a \in G\}$ |
| $\mathbf{x}^{(a)}$ | invariant shape features of atom $a$; $\mathbf{x}^{(a)} \in \mathbb{R}^{6q}$ |
| $\mathbf{X}$ | matrix containing the invariant shape features of each atom in the encoded 3D molecule; $\mathbf{X} \in \mathbb{R}^{n_h\times 6q}$ |
| $d_a$ | dimension of input atom features |
| $d_f$ | dimension of learned fragment embeddings |
| $d_h$ | dimension of learned atom embeddings |
| $\mathbf{h}_\mu$ | vector of means of the posterior distribution $N(\mathbf{h}_\mu, \mathbf{h}_\sigma)$ |
| $\mathbf{h}_\sigma$ | vector of standard deviations of the posterior distribution $N(\mathbf{h}_\mu, \mathbf{h}_\sigma)$ |
| $\mathbf{h}_{\mathrm{var}}$ | sampled variational atom features |
| $\lambda$ | interpolation factor between $N(\mathbf{h}_\mu, \mathbf{h}_\sigma)$ and $N(\mathbf{0}, \mathbf{1})$ |
| $N(\mathbf{0}, \mathbf{1})$ | multidimensional standard normal distribution with $\boldsymbol{\mu} = \mathbf{0}, \boldsymbol{\sigma} = \mathbf{1}$ |

Table 3: List of definitions, terms, and notations (continued)

| Symbol | Description |
|---|---|
| $\mathbf{H}_{\text{var}}$ | matrix containing the sampled variational atom features for each atom in the encoded molecule |
| $\boldsymbol{\epsilon}^{(a)}$ | sampled Gaussian noise vector for atom $a$; $\boldsymbol{\epsilon}^{(a)} \sim N(\mathbf{0}, \mathbf{1})$ |
| $\mathbf{W}_{\mathbf{H}}^{(a)}$ | learned linear transformation for atom $a$, applied to $\tilde{\mathbf{X}}^{(a)}$ |
| $\tilde{\mathbf{X}}_{\mathbf{H}_{\text{var}}}$ | tensor containing the per-atom equivariant shape features, after mixing with $\mathbf{H}_{\text{var}}$; $\tilde{\mathbf{X}}_{\mathbf{H}_{\text{var}}} \in \mathbb{R}^{n_h \times d_z \times 3}$ |
| $\tilde{\mathbf{X}}_{\mathbf{H}_{\text{var}}}^{(a)}$ | equivariant shape features for atom $a$, after mixing with $\mathbf{h}_{\text{var}}^{(a)}$ |
| $\mathbf{x}_{\mathbf{H}_{\text{var}}}^{(a)}$ | invariant embedding of $\tilde{\mathbf{X}}_{\mathbf{H}_{\text{var}}}^{(a)}$ |
| $\tilde{\mathbf{Z}}$ | global equivariant representation of the encoded molecule |
| $\mathbf{z}$ | global invariant representation of the encoded molecule |
| $d_z$ | dimensionality of $\tilde{\mathbf{Z}} \in \mathbb{R}^{d_z \times 3}$ and $\mathbf{z} \in \mathbb{R}^{d_z}$ |
| $\mathbf{H}'$, $\mathbf{W}'_{\mathbf{H}}$, $\tilde{\mathbf{X}}'$, $\mathbf{X}'_{\mathbf{H}}$, $\tilde{\mathbf{X}}'_{\mathbf{H}}$, $\tilde{\mathbf{Z}}'$, $\mathbf{z}'$ | analogues to $\mathbf{H}, \mathbf{W}_{\mathbf{H}}, \tilde{\mathbf{X}}, \mathbf{X}_{\mathbf{H}_{\text{var}}}, \tilde{\mathbf{X}}_{\text{var}}, \tilde{\mathbf{Z}}, \mathbf{z}$, but for the encoded partially generated molecule |
| $\tilde{\mathbf{Z}}'_{\text{foc}}$ | sum-pooled $\tilde{\mathbf{X}}_{\mathbf{H}}^{\prime(a)}$ over the atoms currently in focus |
| $\mathbf{z}'_{\text{foc}}$ | sum-pooled $\mathbf{x}_{\mathbf{H}}^{\prime(a)}$ over the atoms currently in focus |
| $\tilde{\mathbf{Z}}_{\text{dec}}$ | equivariant features of the global system $(M_l^{\prime(c-1)}, M_S)$ |
| $\mathbf{z}_{\text{dec}}$ | invariant features of the global system $(M_l^{\prime(c-1)}, M_S)$ |
| $d_{\text{dec}}$ | the dimensionality of $\mathbf{z}_{\text{dec}}$ |
| $p_\emptyset$ | the predicted probability of stopping local generation |
| $\tau_\emptyset$ | probability threshold to stop local generation |
| $\mathbf{p}_{\text{focus}}$ | predicted probabilities over the attachment sites on the focus |
| $\mathbf{p}_{\text{next}}$ | predicted probabilities over the atom/fragment types in $\mathcal{L}_f$ |
| $\mathbf{p}_{\text{site}}$ | predicted probabilities over the attachment sites on the next fragment |
| $a_{\text{foc}}$ | the atom of attachment on the focus |
| $f_{\text{next}}$ | the atom/fragment to be added next |
| $G_{f_{\text{next}}}$ | the graph of the fragment to be added next |
| $a_{\text{site}}$ | the atom of attachment on the fragment to be added next |
| $\psi'_{l+1}$ | rotation angle (dihedral) of the bond connecting the focus to its parent (tree) node in the partially generated molecule |
| $\psi_{\text{foc}}$ | query dihedral angle when scoring the rotation angle of the focal rotatable bond ($\psi'_{l+1} = \psi_{\text{foc}}$) |
| $M_{l+1}^{\prime(\psi_{\text{foc}})}$ | the 3D molecule $M'_{l+1}$ after setting the query dihedral $\psi'_{l+1} = \psi_{\text{foc}}$ |
| $\mathbf{z}_{\text{dec, scorer}}^{(\psi_{\text{foc}})}$ | analogue to $\mathbf{z}_{\text{dec}}$, but for the rotatable bond scorer when encoding the system $(M_S, M_{l+1}^{\prime(\psi_{\text{foc}})})$ |
| $G_{\text{foc}}^{\chi}$ | a stereoisomer of the focus $G'_{\text{foc}}$ |
| $L_{\text{graph-gen}}$ | total loss for training graph generator |
| $L_\emptyset$ | binary cross entropy loss for predicting to stop local generation |
| $L_{\text{focus}}$ | cross entropy loss for selecting the attachment site on the focus |
| $L_{\text{next}}$ | cross entropy loss for selecting which atom/fragment to add next |
| $L_{\text{site}}$ | modified cross entropy loss for selecting the attachment site on the next fragment |
| $L_{\text{bond}}$ | cross entropy loss for predicting the bond order |
| $L_{KL}$ | KL-divergence loss between $N(\mathbf{h}_\mu, \mathbf{h}_\sigma)$ and $N(\mathbf{0}, \mathbf{1})$ |
| $L_{\text{next-shape}}$ | auxiliary loss used when predicting the atom/fragment to add next |
| $L_{\emptyset\text{-shape}}$ | auxiliary loss used when predicting whether to stop local generation |
| $\beta_{KL}$ | weighting of $L_{KL}$ in $L_{\text{graph-gen}}$ |
| $\beta_{\text{next-shape}}$ | weighting of $L_{\text{next-shape}}$ in $L_{\text{graph-gen}}$ |
| $\beta_{\emptyset\text{-shape}}$ | weighting of $L_{\emptyset\text{-shape}}$ in $L_{\text{graph-gen}}$ |

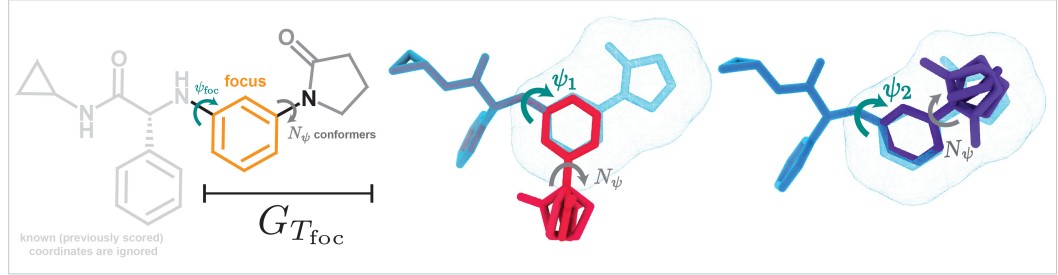

Figure 4: Demonstration of conformer sampling when scoring rotatable bonds. For each query dihedral $\psi_{\text{foc}} = \psi_1, \psi_2, ... \in [-\pi, \pi)$, we sample $N_\psi$ conformations of the subgraph $G_{T_{\text{foc}}}$ (keeping $\psi_{\text{foc}}$ held fixed) induced by the subtree $T_{\text{foc}}$ whose root node is the current focus. For each sampled (subgraph) conformation $\mathbf{G}_{T_{\text{foc}}}$, we compute the shape similarity of between the sampled conformation and the ground truth (subgraph) conformation $\mathbf{G}_{S_{T_{\text{foc}}}}$. We select the maximum computed (non-aligned) shape similarity $\text{sim}_S^*$ amongst the $N_\psi$ sampled conformations as the regression target for that $\psi_{\text{foc}}$.

## A.2 SCORING ROTATABLE BONDS AND STEREOCHEMISTRY

Recall that our goal is to train the scorer to predict $\max\limits_{\psi'_{l+2}, \psi'_{l+3}, ...} \text{sim}_S^*(\mathbf{G}'^{(\psi_{\text{foc}})}, \mathbf{G}_S)$ for various query dihedrals $\psi'_{l+1} = \psi_{\text{foc}}$. That is, we wish to predict the maximum possible shape similarity of the final molecule $M'$ to $M_S$ when fixing $\psi'_{l+1} = \psi_{\text{foc}}$ and optimally rotating all the yet-to-be-scored (or generated) rotatable bond dihedrals $\psi'_{l+2}, \psi'_{l+3}, ...$ so as to maximize $\text{sim}_S^*(\mathbf{G}'^{(\psi_{\text{foc}})}, \mathbf{G}_S)$.

**Training.** We train the scorer independently from the graph generator (with a parallel architecture) using a mean squared error loss between the predicted scores $\hat{s}_{\text{dec, scorer}}^{(\psi_{\text{foc}})} = \text{sigmoid}(\text{MLP}(\mathbf{z}_{\text{dec, scorer}}^{(\psi_{\text{foc}})}))$ and the regression targets $s^{(\psi_{\text{foc}})}$ for $N_s$ different query dihedrals $\psi_{\text{foc}} \in [-\pi, \pi)$:

$$L_{\text{scorer}} = \frac{1}{N_s} \sum_{i=1}^{N_s} (s^{(\psi_{\text{foc}}^{(i)})} - \hat{s}_{\text{dec, scorer}}^{(\psi_{\text{foc}}^{(i)})})^2 \tag{14}$$

**Computing regression targets.** When training with teacher forcing ($M'_l = M_{S_l}$, $G' = G_S$), we compute regression targets $s^{\psi_{\text{foc}}} \approx \max\limits_{\psi_{l+2}, \psi_{l+3}, ...} \text{sim}_S^*(\mathbf{G}'^{(\psi_{\text{foc}})}, \mathbf{G}_S)$ by setting the focal dihedral $\psi_{l+1} = \psi_{\text{foc}}$, sampling $N_\psi$ conformations of the "future" graph $G_{T_{\text{foc}}}$ induced by the *subtree* $T_{\text{foc}}$ whose root (sub)tree-node is the focus, and computing $s_{\psi_{\text{foc}}} = \max\limits_{i=0, ..., N_\psi} \text{sim}_S^*(\mathbf{G}_{T_{\text{foc}}}^{(i)}, \mathbf{G}_{S_{T_{\text{foc}}}}; \alpha = 2.0)$. Since we fix bonding geometries, we need only sample $N_\psi$ sets of dihedrals of the rotatable bonds in $G_{S_{T_{\text{foc}}}}$ to sample $N_\psi$ conformers, making this conformer enumeration very fast. Note that rather than using $\alpha = 0.81$ in these regression targets, we use $\alpha = 2.0$ to make the scorer more sensitive to shape differences (App. A.7). When computing regression targets, we use $N_\psi < 1800$ and select 36 (evenly spaced) $\psi_{\text{focus}} \in [-\pi, \pi)$ per rotatable bond. Figure 4 visualizes how regression targets are computed. App. A.15 contains further training specifics.

**Scoring stereochemistry.** At generation time, we also enumerate all possible stereoisomers of the focus (except cis/trans bonds) and score each stereoisomer separately, ultimately selecting the (stereoisomer, $\psi_{\text{foc}}$) pair that maximizes the predicted score. Figure 5 illustrates how we enumerate stereoisomers. Note that although we *use* the learned scoring function to score stereoisomerism at generation time, we do not explicitly *train* the scorer to score different stereoisomers.

**Masking severe steric clashes.** At generation time, we do not score any query dihedral $\psi_{\text{foc}}$ that causes a severe steric clash ($< 1\text{Å}$) with the existing partially generated structure (unless all query dihedrals cause a severe clash).

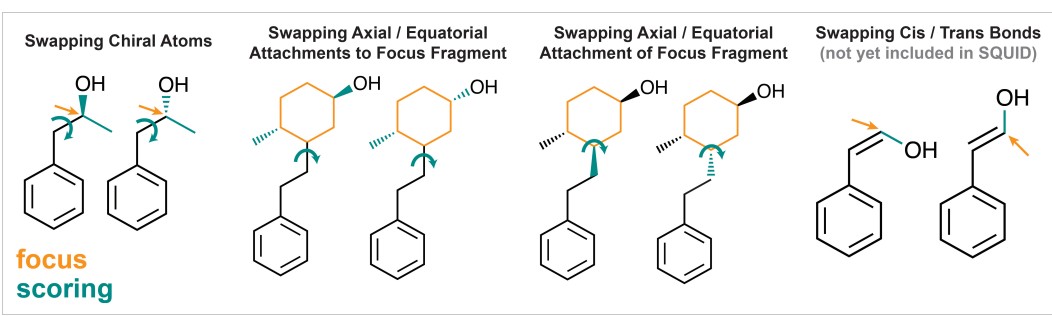

Figure 5: In addition to scoring the dihedral of the focal bond, we also enumerate all stereoisomers around the focus and score each stereoisomer's focal dihedral separately. We then select the best scoring (steroisomer, dihedral) pair. We enumerate stereoisomers by combinatorially swapping chiral atoms, swapping axial/equatorial attachments to the focus (if relevant), and swapping the axial/equatorial attachment of the focus to the rest of the partial molecule (if relevant). Although swapping cis/trans bonds is feasible within SQUID's framework, we *currently* do not enumerate cis/trans isomers because of their minor presence in the dataset. Hence, any generated double bonds will have random stereochemistry.

## A.3 RANDOM EXAMPLES OF GENERATED 3D MOLECULES

Figures 6 and 7 show additional random examples of molecules generated by SQUID when sampling $N_{\max} = 1, 20$ molecules with $\text{sim}_G(M', M_S) < 0.7$ from the prior ($\lambda = 1.0$) or $\lambda = 0.3$ and selecting the sample with the highest $\text{sim}_S(M', M_S)$. Note that the visualized poses of the generated conformers are those which are directly generated by SQUID; the generated conformers have not been explicitly aligned to $M_S$ (e.g., using ROCS). Even so, the conformers are (for the most part) aligned to $M_S$ since SQUID's equivariance enables the model to generate natively aligned structures.

It is apparent in these examples that using larger $N_{\max}$ yields molecules with significantly improved shape similarity to $M_S$, both qualitatively and quantitatively. This is in part caused by: 1) stochasticity in the variationally sampled atom embeddings $\mathbf{H}_{\text{var}}$; 2) stochasticity in the input molecular point clouds, which are sampled from atom-centered isotropic Gaussians in $\mathbb{R}^3$; 3) sampling sets of variational atom embeddings that may not be entirely self-consistent (e.g., for instance, if we sample only 1 atom embedding that implicitly encodes a ring structure); and 4) the choice of $\tau_\emptyset$, the threshold for stopping local generation. While a small $\tau_\emptyset$ (we use $\tau_\emptyset = 0.01$) helps prevent the model from adding too many atoms or fragments around a single focus, a small $\tau_\emptyset$ can also lead to early (local) stoppage, yielding molecules that do not completely fill the target shape. By sampling more molecules (using larger $N_{\max}$), we have more chances to avoid these adverse random effects. Further work will attempt to improve the robustness of the encoding scheme and generation procedure in order to increase SQUID's overall sample efficiency.

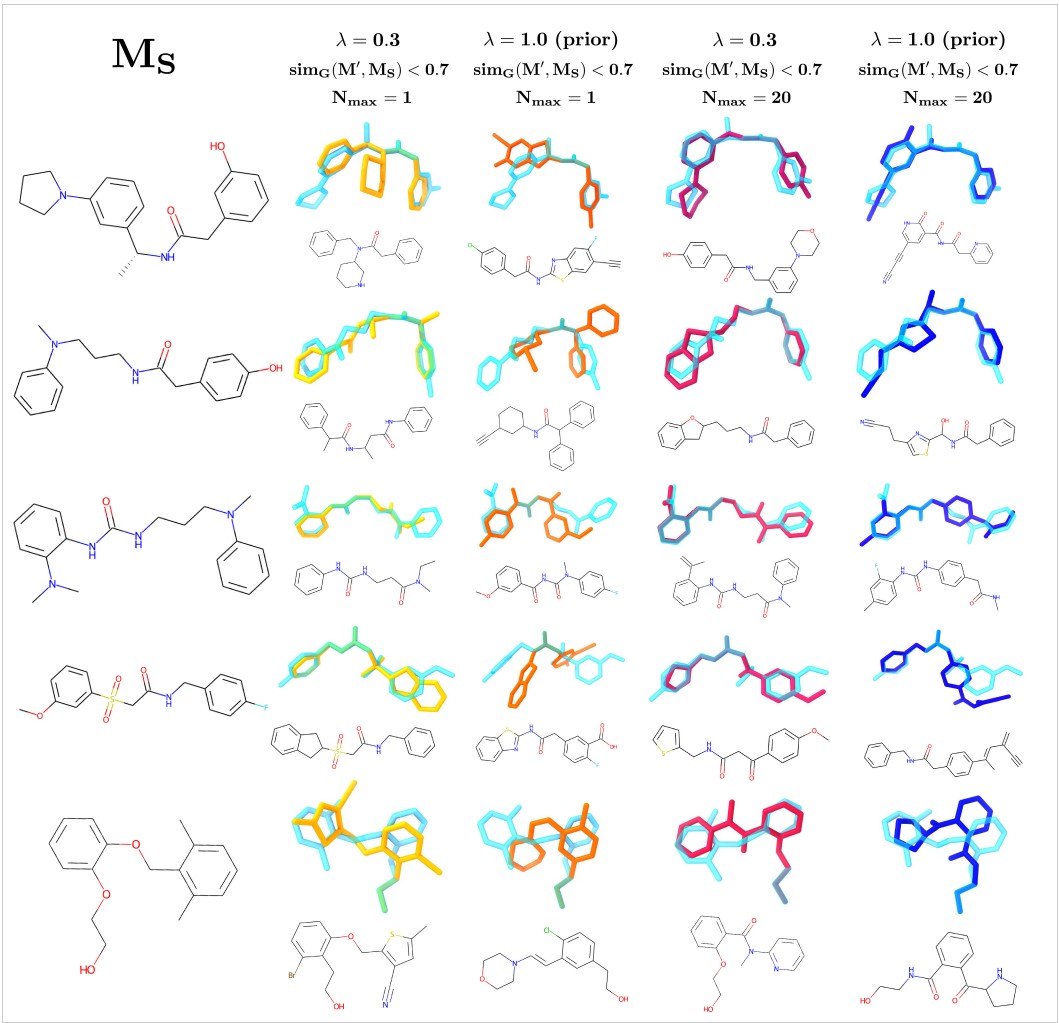

Figure 6: Random examples of molecules generated by SQUID using different sampling strategies. The generated 3D molecules are shown overlaid on the target molecule $M_S$ (blue). In these examples, the generated molecules have *not* been explicitly aligned to $M_S$, and their displayed poses are those directly generated by SQUID.

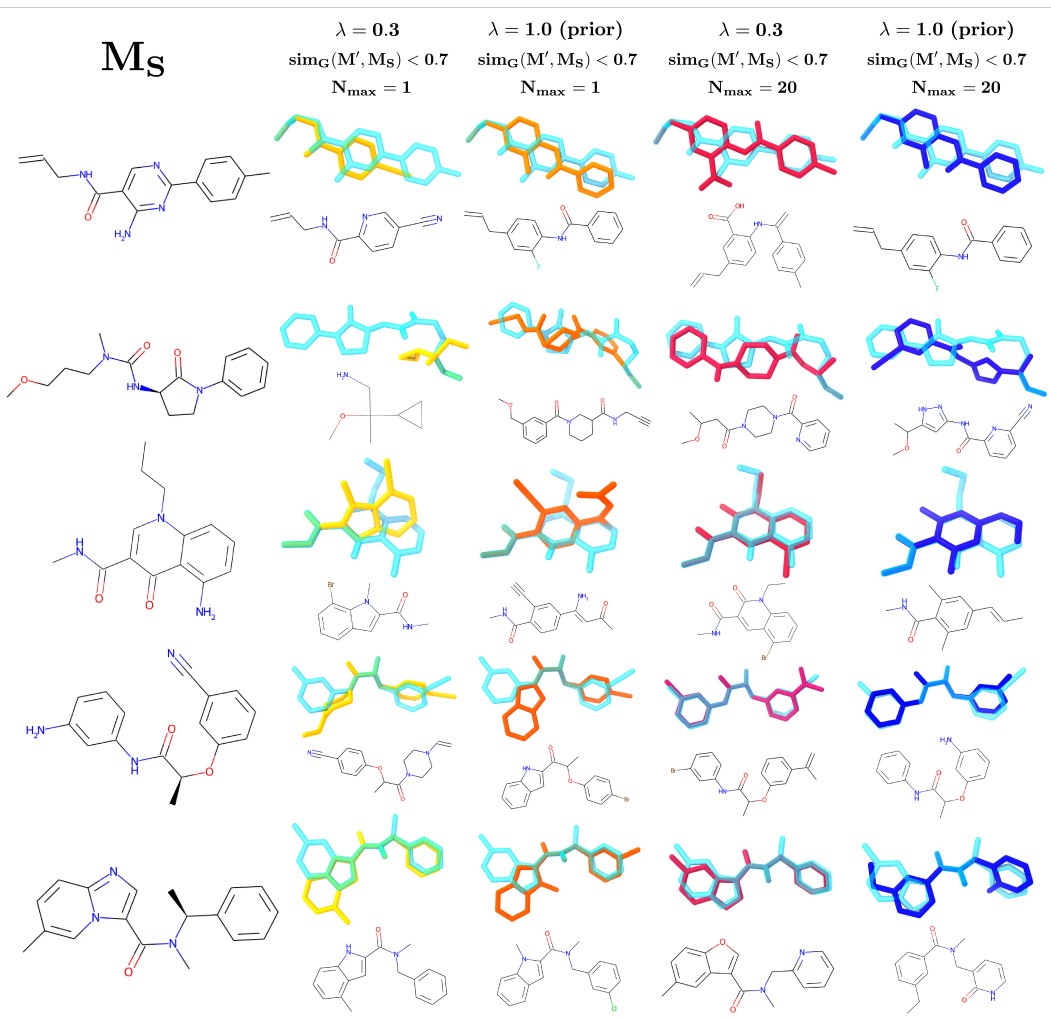

Figure 7: Additional random examples of molecules generated by SQUID using different sampling strategies. The generated 3D molecules are shown overlaid on the target molecule $M_S$ (blue). In these examples, the generated molecules have *not* been explicitly aligned to $M_S$, and their displayed poses are those directly generated by SQUID.

A.4   GENERATION STATISTICS

Table 4 reports the percentage of molecules that are chemically valid, novel, and unique when sampling 50 molecules from the prior ($\lambda = 1.0$) for 1000 encoded molecules $M_S$ (e.g., target shapes) from the test set, yielding a total of 50K generated molecules. We define chemical validity as passing RDKit sanitization. Since we directly generate the molecular graph and mask actions which violate chemical valency, 100% of generated molecules are valid. We define novelty as the percentage of generated molecules whose molecular graphs are not present in the training data. We define uniqueness as the percentage of generated molecular graphs (of the 50K total) that are only generated once. For novelty and uniqueness calculations, we consider different stereoisomers to have the same molecular graph. We also report the percentage of generated 3D structures that have an apparent steric clash, defined to be a non-bonded interatomic distance below 2Å.

When sampling from the prior ($\lambda = 1.0$), the average internal chemical similarity of the generated molecules is $0.26 \pm 0.04$. When sampling with $\lambda = 0.3$, the average internal chemical similarity is $0.32 \pm 0.07$. We define internal chemical similarity to be the average pairwise chemical similarity (Tanimoto fingerprint similarity) between molecules that are generated for the same target shape.

Table 5 reports the graph reconstruction accuracy when sampling 3D molecules from the posterior ($\lambda = 0.0$), for 1000 target molecules $M_S$ from the test set. We report the top-$k$ graph reconstruction accuracy (ignoring stereochemical differences) when sampling $k = 1$ molecule per encoded $M_S$, and when sampling $k = 20$ molecules per encoded $M_S$. Since we have intentionally trained SQUID inside a shape-conditioned variational autoencoder framework in order to generate chemically *diverse* molecules with similar *3D shapes*, the significance of graph reconstruction accuracy is debatable in our setting. However, it is worth noting that the top-1 reconstruction accuracy is 16.3%, while the top-20 reconstruction accuracy is much higher (57.2%). This large difference is likely attributable to both stochasticity in the variational atom embeddings *and* stochasticity in the input 3D point clouds.

Table 4: Generation statistics computed by sampling 50 molecules from the prior ($\lambda = 1.0$) per encoded target molecule $M_S$ for 1000 random targets from the test set.

| Statistic | % |
|---|---|
| Chemical Validity ($\uparrow$) | 100% |
| Novelty ($\uparrow$) | 98.8% |
| Uniqueness ($\uparrow$) | 94.7% |
| Steric Clash ($<$2Å) ($\downarrow$) | 13.1% |

Table 5: *Graph* reconstruction accuracy when sampling 3D molecules from the posterior ($\lambda = 0.0$) for 1000 targets $M_S$ from the test set. We report the "top-k" reconstruction accuracy when sampling $k = 1$ molecule per target, as well as $k = 20$ molecules per target.

| Graph Reconstruction | % |
|---|---|
| $k = 1$ | 16.3% |
| $k = 20$ | 57.2% |

### A.5  SHAPE-CONSTRAINED MOLECULAR OPTIMIZATION

#### A.5.1  GENETIC ALGORITHM

We adapt SQUID to shape-constrained molecular optimization by implementing a genetic algorithm on the variational atom embeddings $\mathbf{H}_{\mathrm{var}}$. Algorithm 1 details the exact optimization procedure. In summary, given the seed molecule $M_S$ with a target 3D shape and an initial substructure $M_0$ (which is contained by all generated molecules for a given $M_S$), we first generate an initial population of generated molecules $M'$ by repeatedly sampling $\mathbf{H}_{\mathrm{var}}$ for various interpolation factors $\lambda$, mixing these $\mathbf{H}_{\mathrm{var}}$ with the encoded shape features of $M_S$, and decoding new 3D molecules. We only add a generated molecule to the population if $\mathrm{sim}_S(M', M_S) \geq \tau_S$ (we use $\tau_S = 0.75$), so that the GA does not overly explore regions of chemical space that have no chance of satisfying the ultimate constraint $\mathrm{sim}_S(M', M_S) \geq 0.85$. After generating the initial population, we iteratively 1) select the top-scoring samples in the population, 2) cross the top-scoring $\mathbf{H}_{\mathrm{var}}$ in crossover events, 3) mutate the top and crossed $\mathbf{H}_{\mathrm{var}}$ via adding random noise, and 4) generate new molecules $M'$ for each mutated $\mathbf{H}_{\mathrm{var}}$. The final optimized molecule $M^*$ is the top-scoring generated molecule that satisfies the shape-similarity constraint $\mathrm{sim}_S(M', M_S) \geq 0.85$.

#### A.5.2  VISUALIZATION OF OPTIMIZED MOLECULES

Figure 8 visualizes the structures of the SQUID-optimized molecules $M^*$ and their respective seed molecules $M_S$ (e.g., the starting "hit" molecules with target shapes) for each of the optimization tasks which led to an improvement in the objective score. We also overlay the generated 3D conformations of $M^*$ on those of $M_S$, and report the objective scores for each $M^*$ and $M_S$.

---

**Algorithm 1** Genetic algorithm for shape-constrained optimization with SQUID

---

**Given:** $M_S$ with $n_H$ heavy atoms, $M_0$, objective oracle $O$
**Params:** $\tau_S, \tau_G, N_e, N_T, N_c$ ▷ Defaults: $\tau_S = 0.75, \tau_G = 0.95, N_e = 20, N_T = 20, N_c = 10$

$\mathbf{H}_\mu, \mathbf{H}_\sigma = \text{Encode}(M_S)$ ▷ Encode target molecule
Initialize population $P = \{(M_S, \mathbf{H}_\mu)\}$
**for** $\lambda \in [0.0, 0.2, 0.4, 0.6, 0.8, 1.0]$ **do** ▷ Create initial population of $(M', \mathbf{H}_{\text{var}})$
    **for** $i = 1, ..., 100$ **do**
        Sample noise $\boldsymbol{\epsilon} \in \mathbb{R}^{(n_H \times d_h)} \sim N(\mathbf{0}, \mathbf{1})$
        $\mathbf{H}_{\text{var}} = (1 - \lambda)\mathbf{H}_\mu + \boldsymbol{\epsilon} \odot ((1 - \lambda)\mathbf{H}_\sigma + \lambda\mathbf{1})$ ▷ Mutate variational atom embeddings
        $\mathbf{z}, \tilde{\mathbf{Z}} = \text{Encode}(M_S; \mathbf{H}_{\text{var}})$ ▷ Mix mutated chemical and shape information
        $M' = \text{Decode}(M_0, \mathbf{z}, \tilde{\mathbf{Z}})$ ▷ Generate mutated molecule $M'$
        Compute $\text{sim}_S(M', M_S)$
        **if** $\text{sim}_S(M', M_S) >= \tau_S$ **then** ▷ Add $M'$ to population only if $\text{sim}_S(M', M_S)$ is high
            Add $(M', \mathbf{H}_{\text{var}})$ to $P$
        **end if**
    **end for**
**end for**

**for** $e = 1, ..., N_e$ **do** ▷ For each evolution
    Construct $P_{\text{sorted}}$ by sorting $P$ by $O(M)$ ▷ Sort population by objective score, high to low.
    Initialize $T_M = \{\}, T_{H_{\text{var}}} = \{\}$
    **for** $(M, \mathbf{H}_{\text{var}}) \in P_{\text{sorted}}$ **do** ▷ Collect top-$N_T$ scoring $(M', \mathbf{H}_{\text{var}})$
        **if** $(\text{sim}_G(M, M_T) < \tau_G \;\; \forall \;\; M_T \in T_M)$ **and** $(|T| < N_T)$ **then**
            Add $M$ to $T_M$
            Add $\mathbf{H}_{\text{var}}$ to $T_{H_{\text{var}}}$
        **end if**
    **end for**

    Initialize $T_C = \{\}$
    **for** $c = 1, ..., N_c$ **do** ▷ Add crossovers to set of top-scoring $\mathbf{H}_{\text{var}}$
        Sample $\mathbf{H}_i \in T_{H_{\text{var}}}, \mathbf{H}_{j \neq i} \in T_{H_{\text{var}}}$
        $\mathbf{H}_c = CROSS(\mathbf{H}_i, \mathbf{H}_j)$ ▷ Cross by randomly swapping half of the atom embeddings
        Add $\mathbf{H}_c$ to $T_C$
    **end for**
    $T_{H_{\text{var}}} = T_{H_{\text{var}}} \cup T_C$

    **for** $\mathbf{H}_{\text{var}} \in T_{H_{\text{var}}}$ **do** ▷ Adding to population
        **for** $\lambda \in [0.0, 0.2, 0.4, 0.6, 0.8, 1.0]$ **do**
            **for** $i = 1, ..., 10$ **do**
                Sample noise $\boldsymbol{\epsilon} \in \mathbb{R}^{(n_H \times d_h)} \sim N(\mathbf{0}, \mathbf{1})$
                $\mathbf{H}_{\text{var}} = (1 - \lambda)\mathbf{H}_{\text{var}} + \boldsymbol{\epsilon}$ ▷ Mutate variational atom embeddings
                $\mathbf{z}, \tilde{\mathbf{Z}} = \text{Encode}(M_S; \mathbf{H}_{\text{var}})$ ▷ Mix mutated chemical and shape information
                $M' = \text{Decode}(M_0, \mathbf{z}, \tilde{\mathbf{Z}})$ ▷ Generate mutated molecule $M'$
                Compute $\text{sim}_S(M', M_S)$
                **if** $\text{sim}_S(M', M_S) >= \tau_S$ **then** ▷ Add $M'$ to $P$ only if $\text{sim}_S(M', M_S)$ is high
                    Add $(M', \mathbf{H}_{\text{var}})$ to $P$
                **end if**
            **end for**
        **end for**
    **end for**
**end for**
    **return** $M^* = \underset{M' \in P}{\arg\max} \, O(M')$ subject to $\text{sim}_S(M', M_S) >= 0.85$

---

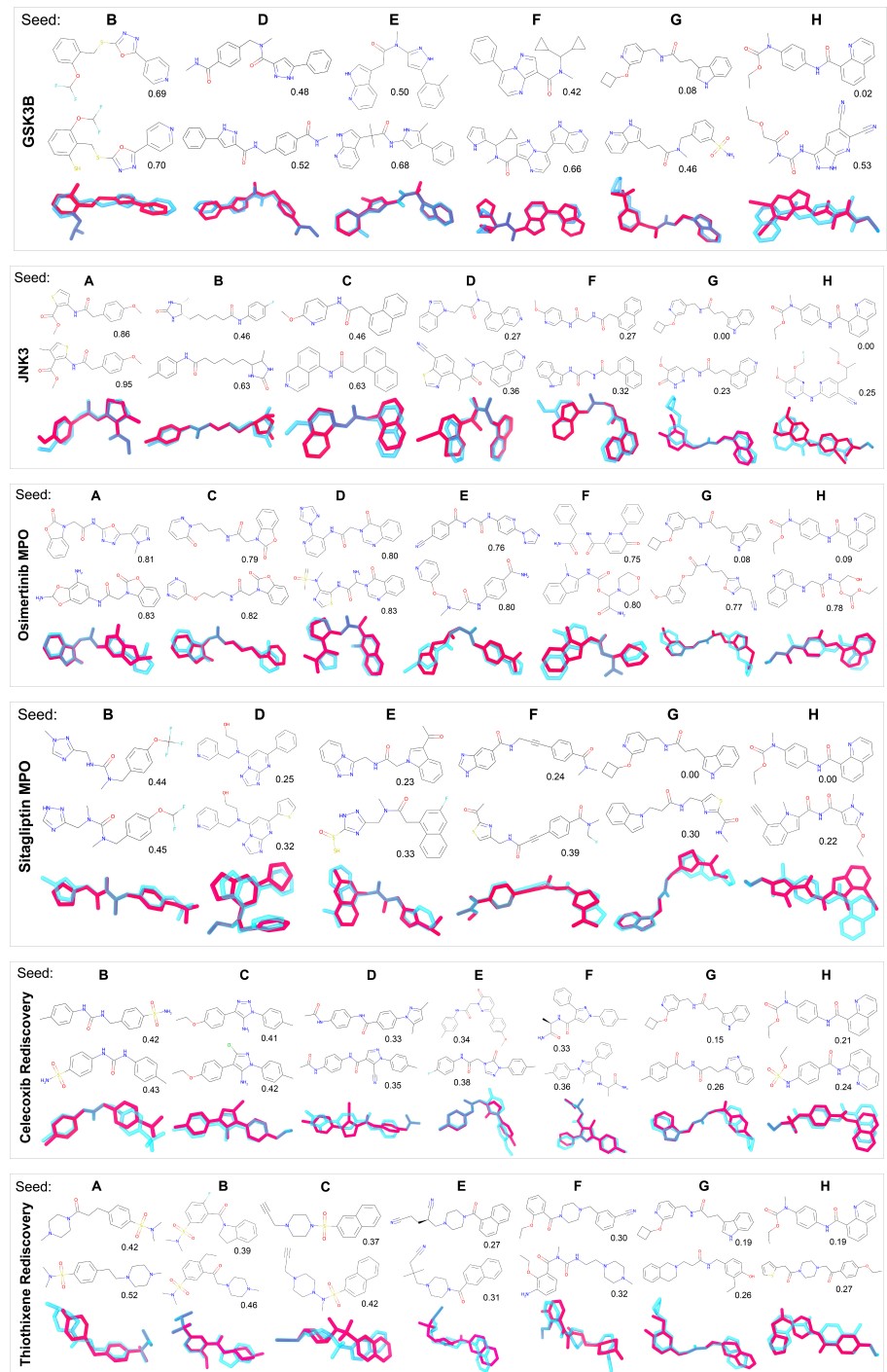

Figure 8: Results of shape-constrained MO for the 40/48 tasks which improved the objective score. Subpanels depict the target molecules $M_S$ (top row), the SQUID-optimized molecules $M^*$ (middle row), the overlaid 3D structures (bottom row; blue is $M_S$, red is $M^*$), and their respective objective scores for the (from top to bottom) GSK3B, JNK3, Osimertinib MPO, Sitagliptin MPO, Celecoxib Rediscovery, and Thiothixene Rediscovery tasks. In the overlaid structures, the generated molecules have *not* been explicitly aligned to $M_S$; the poses shown are those directly generated by SQUID.

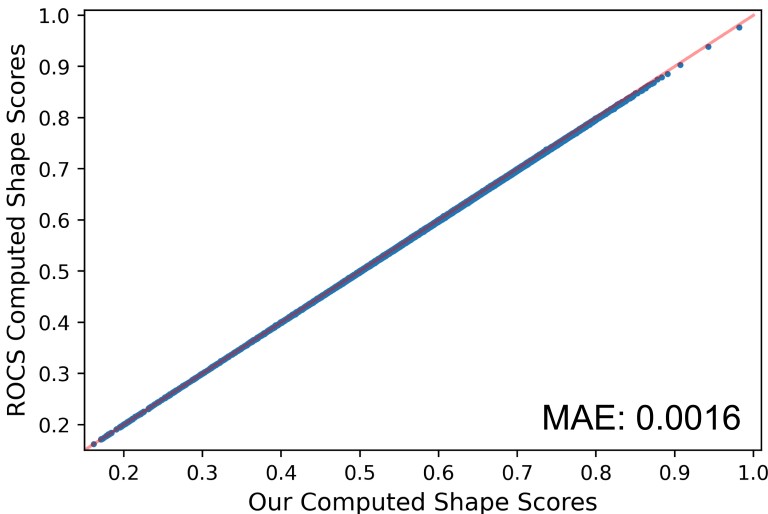

Figure 9: Parity plot empirically showing the close approximation of Eq. 1 ($\alpha = 0.81$) to the shape-similarity function used by the commercial ROCS program, for 50000 shape comparisons.

## A.6    COMPARING SIM$_S$ TO ROCS SCORING FUNCTION

Our shape similarity function described in Equation 1 closely approximates the shape (only) scoring function employed by ROCS, when $\alpha = 0.81$. Figure 9 demonstrates the near-perfect correlation between our computed shape scores and those computed by ROCS for 50,000 shape comparisons, with a mean absolute error of 0.0016. Note that Equation 1 computes non-aligned shape similarity. We still employ ROCS to align the generated molecules $M'$ to the target molecule $M_S$ before computing their (aligned) shape similarity in our experiments. However, we do not require explicit alignment when training SQUID; we do not use the commercial ROCS program during training.

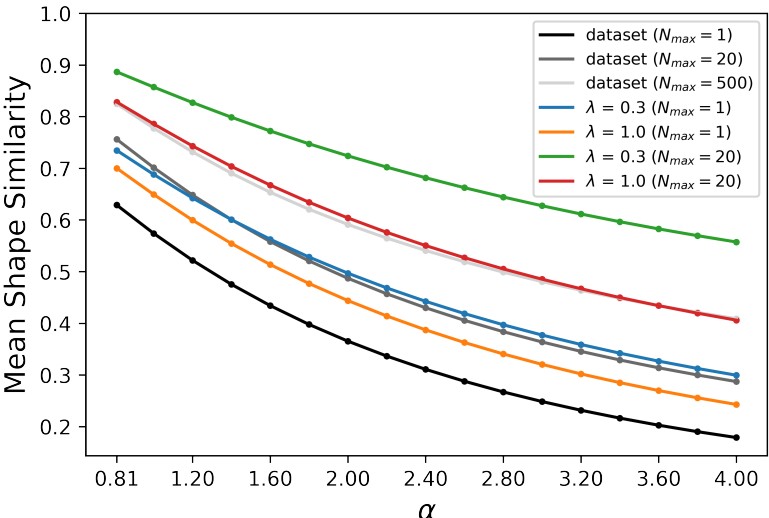

Figure 10: Mean shape similarity $\text{sim}_S(M, M_S; \alpha)$ for increasing values of $\alpha$ (Equation 1) for 1000 target molecules $M_S$ from the test set. When sampling either from the dataset or from SQUID, we compute the mean shape similarity (across the 1000 targets) after selecting the best sample $M$ which maximizes $\text{sim}_S(M', M_S; \alpha)$ amongst $N_{\max}$ samples $M'$ for which $\text{sim}_G(M', M_S) < 0.7$.

### A.7 EXPLORING DIFFERENT VALUES OF $\alpha$ IN $\text{SIM}_S$

Our analysis of shape similarity thus far has used Equation 1 with $\alpha = 0.81$ in order to recapitulate the shape similarity function used by ROCS, which is widely used in drug discovery. However, compared to randomly sampled molecules in the dataset, the molecules generated by SQUID *qualitatively* appear to do a significantly better job at fitting the target shape $S$ on an atom-by-atom basis, even if the computed shape similarities (with $\alpha = 0.81$) are comparable (see examples in Figure 3). We quantify this observation by increasing the value of $\alpha$ when computing $\text{sim}_S(M', M_S; \alpha)$ for generated molecules $M'$, as $\alpha$ is inversely related to the width of the isotropic 3D Gaussians used in the volume overlap calculations in Equation 1. Intuitively, increasing $\alpha$ will greater penalize $\text{sim}_S$ if the atoms of $M'$ and $M_S$ do not perfectly align.

Figure 10 plots the mean $\text{sim}_S(M, M_S; \alpha)$ for the most shape-similar molecule $M$ of $N_{\max}$ sampled molecules $M'$ for increasing values of $\alpha$. Averages are calculated over 1000 target molecules $M_S$ from the test set, and we only consider generated molecules for which $\text{sim}_G(M', M_S) < 0.7$. Crucially, the gap between the mean $\text{sim}_S(M, M_S; \alpha)$ obtained by generating molecules with SQUID vs. randomly sampling molecules from the dataset significantly widens with increasing $\alpha$. This effect is especially apparent when using SQUID with $\lambda = 0.3$ and $N_{\max} = 20$, although can be observed with other generation strategies as well. Hence, SQUID does a much better job at generating (still chemically diverse) molecules that have significant atom-to-atom overlap with $M_S$.

## A.8 HEURISTIC BONDING GEOMETRIES AND THEIR IMPACT ON GLOBAL SHAPE

In all molecules (dataset and generated) considered in this work, we fix acyclic bond distances to their empirical averages and set acyclic bond angles to heuristic values based on hybridization rules in order to reduce the degrees of freedom in 3D coordinate generation. Here, we describe how we fix these bonding geometries and explore whether this local 3D structure manipulation significantly alters the *global* molecular shape.

**Fixing bonding geometries.** We fix acyclic bond distances by computing the mean bond distance between pairs of atom types across all the RDKit-generated conformers in our training set. After collecting these empirical mean values, we manually set each acyclic bond distance to its respective mean value for each conformer in our datasets.

We set acyclic bond angles using simple hybridization rules. Specifically, sp3-hybridized atoms will have bond angles of $109.5°$, sp2-hybridized atoms will have bond angles of $120°$, and sp-hybridized atoms will have bond angles of $180°$. We manually fix the acyclic bond angles to these heuristic values for all conformers in our datasets. We use RDKit to determine the hybridization states of each atom. During generation, occasionally the hybridization of certain atoms (N, O) may change once they are bonded to new neighbors. For instance, an sp3 nitrogen can become sp2 once bonded to an aromatic ring. We adjust bond angles on-the-fly in these edge cases.

**Impact on global shape.** Figure 11 plots the histogram of $\mathrm{sim}_S(M_{\mathrm{fixed}}, M_{\mathrm{relaxed}})$ for 1000 test set conformers $M_{\mathrm{fixed}}$ whose bonding geometries have been fixed, and the original RDKit-generated conformers $M_{\mathrm{relaxed}}$ with relaxed (true) bonding geometries. In the vast majority of cases, fixing the bonding geometries negligibly impacts the global shape of the 3D molecule ($\mathrm{sim}_S(M_{\mathrm{fixed}}, M_{\mathrm{relaxed}}) \approx 1$). This is because the main factor influencing global molecular shape is rotatable bonds (e.g., flexible dihedrals), which are *not* altered by fixing bond distances and angles.

**Recovering refined bonding geometries.** Even though fixing bond distances and angles only marginally impacts molecular shape, we still may wish to recover refined bonding geometries of the generated 3D molecules without altering the generated 3D shape. We can accomplish this (to a first approximation) for generated molecules by creating a geometrically relaxed conformation of the generated molecular graph with RDKit, and then manually setting the dihedrals of the rotatable bonds in the relaxed conformer to match the corresponding dihedrals in the generated conformers. Importantly, if we perform this relaxation procedure for both the dataset molecules and the SQUID-generated molecules, the (relaxed) generated molecules still have significantly enriched shape-similarity to the (relaxed) target shape compared to (relaxed) random molecules from the dataset (Fig. 12).

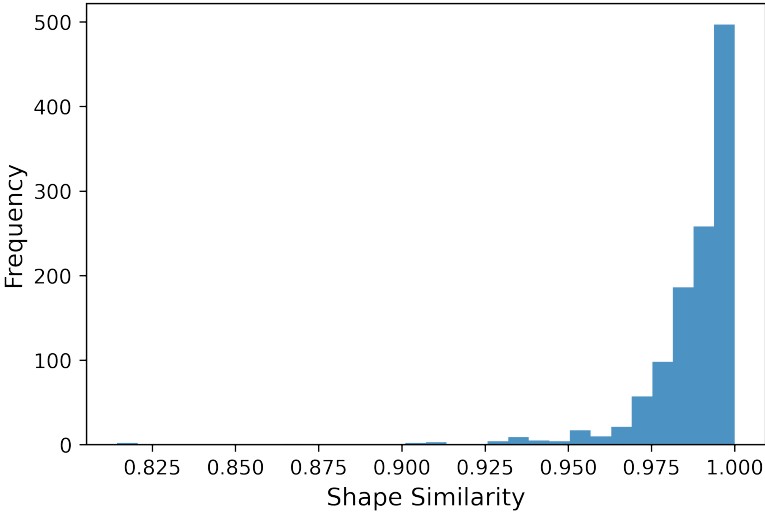

Figure 11: Histogram of shape similarity $\text{sim}_S(M_{\text{fixed}}, M_{\text{relaxed}})$ between 1000 test molecules whose acyclic bonding geometries are fixed by heuristics and their geometrically relaxed counterparts.

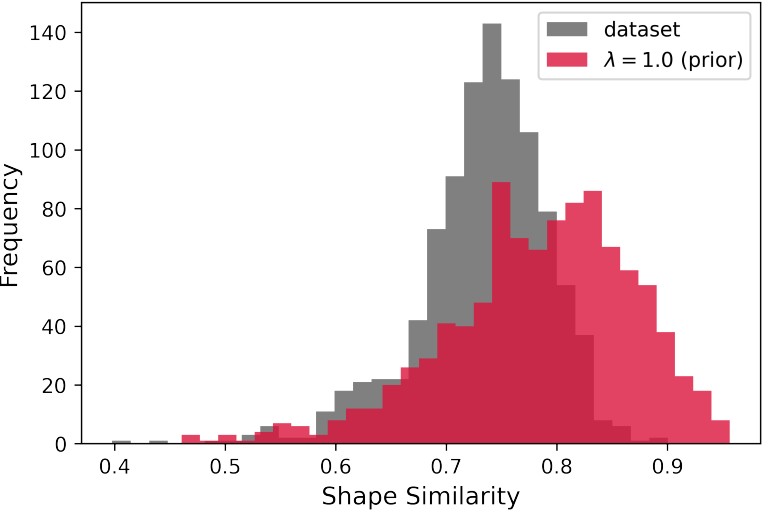

Figure 12: Distribution of $\text{sim}_S(M', M_S)$ when sampling ($N_{\text{max}} = 20$, $\text{sim}_G(M', M_S) < 0.7$) molecules $M'$ from SQUID ($\lambda = 1.0$) and from the dataset, when relaxing the bonding geometries of each $M'$ and $M_S$. Importantly, SQUID generates 3D molecules that are significantly enriched in shape similarity even after relaxing the (fixed) bonding geometries of the generated structures.

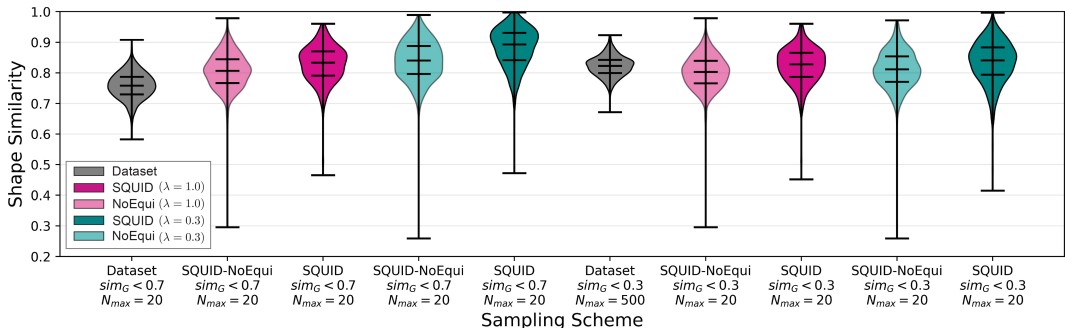

Figure 13: Distributions of $\text{sim}_S(M', M_S)$ for the best of $N_{\max}$ generated molecules for which $\text{sim}_G(M', M_S) < 0.7$ or $0.3$, when using SQUID or SQUID-NoEqui. Generated distributions are compared to those obtained by similarly sampling molecules from the training dataset. Overall, ablating SQUID's equivariance signficantly reduces the enrichment in shape similarity of generated molecules to the target shape.

## A.9 ABLATING EQUIVARIANCE

SQUID aligns the equivariant representations of the encoded target shape and the partially generated structures in order to generate 3D conformations that natively fit the target shape, without having to implicitly learn SE(3)-alignments (Challenge 2). We achieve this in Equation 7, where we mix the equivariant representations of $M_S$ and the partially generated structure $M_l'^{(c-1)}$. To empirically motivate this design choice, we ablate the equivariant alignment by setting $\tilde{\mathbf{Z}} = \mathbf{0}$ in Eq. 7. We denote this ablated model as SQUID-NoEqui. Note that because we still pass the unablated *invariant* features $\mathbf{z}$ to the decoder (Eq. 8), SQUID-NoEqui is still conditioned on the shape of $M_S$ — the model simply no longer has access to any explicit information about the relative spatial orientation of $M_l'^{(c-1)}$ to $M_S$ (and thus must learn this spatial relationship from scratch). As expected, ablating SQUID's equivariance significantly reduces SQUID's ability to generate chemically diverse molecules that fit the target shape.

Figure 13 plots the distributions of $\text{sim}_S(M', M_S)$ for the best of $N_{\max}$ generated molecules with $\text{sim}_G(M', M_S) < 0.7$ or $0.3$ when using SQUID or SQUID-NoEqui. Crucially, the mean shape similarity when sampling with ($\lambda = 1.0, N_{\max} = 20, \text{sim}_G(M', M_S) < 0.7$) decreases from 0.828 (SQUID) to 0.805 (SQUID-NoEqui). When sampling with ($\lambda = 0.3, N_{\max} = 20, \text{sim}_G(M', M_S) < 0.7$), the mean shape similarity also decreases from 0.879 (SQUID) to 0.839 (SQUID-NoEqui). Relative to the mean shape similarity of 0.758 achieved by sampling random molecules from the dataset ($N_{\max} = 20, \text{sim}_G(M', M_S) < 0.7$), this corresponds to a substantial 33% reduction in the shape-enrichment of SQUID-generated molecules.

Interestingly, sampling ($\lambda = 1.0, N_{\max} = 20, \text{sim}_G(M', M_S) < 0.7$) with SQUID-NoEqui still yields shape-enriched molecules compared to analogously sampling random molecules from the dataset (mean shape similarity of 0.805 vs. 0.758). This is because even without the equivariant feature alignment, SQUID-NoEqui still conditions molecular generation on the (invariant) encoding of the target shape $S$, and hence biases generation towards molecules which better fit the target shape (after alignment with ROCS).

### A.10 Auxiliary training losses

We employ two auxiliary losses when training the graph generator in order to encourage the generated graphs to better fit the encoded target shape.

The first auxiliary loss penalizes the graph generator if it adds an incorrect atom/fragment to the focus that is of significantly different size than the correct (ground truth) atom/fragment. We first compute a matrix $\Delta \mathbf{V}_f \in \mathbb{R}_+^{(|\mathcal{L}_f| \times |\mathcal{L}_f|)}$ containing the (pairwise) volume difference between all atoms/fragments in the library $\mathcal{L}_f$

$$\Delta \mathbf{V}_f^{(i,j)} = |v_{f_i} - v_{f_j}| \tag{15}$$

where $v_{f_i}$ is the volume of atom/fragment $f_i \in \mathcal{L}_f$ (computed with RDKit).

We then compute the auxiliary loss $L_{\text{next-shape}}$ as:

$$L_{\text{next-shape}} = \frac{1}{|\mathcal{L}_f|}(\mathbf{p}_{\text{next}} \cdot \Delta \mathbf{V}_f^{(g)}) \tag{16}$$

where $g$ is the index of the correct (ground truth) next atom/fragment $f_{\text{next, true}}$, $\Delta \mathbf{V}_f^{(g)}$ is the $g$th row of $\Delta \mathbf{V}_f$, and $\mathbf{p}_{\text{next}}$ are the predicted probabilities over the next atom/fragment types to be connected to the focus (see Eq. 10).

The second auxiliary loss penalizes the graph generator if it prematurely stops (local) generation, with larger penalties if the premature stop would result in larger portions of the (ground truth) graph not being generated. When predicting (local) stop tokens during graph generation (with teacher forcing), we compute the number of atoms in the subgraph induced by the subtree whose root tree-node is the next atom/fragment to be added to the focus (in the current generation sequence). We then multiply the predicted probability for the local stop token by this number of "future" atoms that would *not* be generated if a premature stop token were generated. Hence, if the correct action is to indeed stop generation around the focus, the penalty will be zero. However, if the correct action is to add a large fragment to the current focus but the generator predicts a stop token, the penalty will be large. Formally, we compute:

$$L_{\emptyset\text{-shape}} = p_\emptyset |G_{S_{T_{\text{next}}}}| \quad \text{if } p_{\emptyset, \text{ true}} = 0 \quad \text{otherwise } 0 \tag{17}$$

where $p_{\emptyset, \text{ true}}$ is the ground truth action for local stopping ($p_{\emptyset, \text{ true}} = 0$ indicates that the correct action is to *not* stop local generation), and $G_{S_{T_{\text{next}}}}$ is the subgraph induced by the subtree whose root node is the next atom/fragment (to be generated) in the ground-truth molecular graph.

## A.11 OVERVIEW OF VECTOR NEURONS (VN) OPERATIONS

In this work, we use Deng et al. (2021)'s VN-DGCNN to encode molecular point clouds into equivariant shape features. We also employ their general VN operations (VN-MLP, VN-Inv) during shape and chemical feature mixing. We refer readers to Deng et al. (2021) for a detailed description of these equivariant operations and models. Here, we briefly summarize some relevant VN-operations for the reader's convenience.

**VN-MLP.** Vector neurons (VN) lift scalar neuron features to vector features in $\mathbb{R}^3$. Hence, instead of having features $\mathbf{x} \in \mathbb{R}^q$, we have vector features $\tilde{\mathbf{X}} \in \mathbb{R}^{q \times 3}$. While linear transformations are naturally equivariant to global rotations $\mathbf{R}$ since $\mathbf{W}(\tilde{\mathbf{X}}\mathbf{R}) = (\mathbf{W}\tilde{\mathbf{X}})\mathbf{R}$ for some rotation matrix $\mathbf{R} \in \mathbb{R}^{3 \times 3}$, Deng et al. (2021) construct a set of non-linear equivariant operations $\tilde{f}$ such that $\tilde{f}(\tilde{\mathbf{X}}\mathbf{R}) = \tilde{f}(\tilde{\mathbf{X}})\mathbf{R}$, thereby enabling natively equivariant network design.

VN-MLPs combine linear transformations with equivariant activations. In this work, we use VN-LeakyReLU, which Deng et al. (2021) define as:

$$\text{VN-LeakyReLU}(\tilde{\mathbf{X}}; \alpha) = \alpha\tilde{\mathbf{X}} + (1 - \alpha)\text{VN-ReLU}(\tilde{\mathbf{X}}) \tag{18}$$

where

$$\text{VN-ReLU}(\tilde{\mathbf{X}}) = \begin{cases} \tilde{\mathbf{x}}, & \text{if } \tilde{\mathbf{x}} \cdot \frac{\tilde{\mathbf{k}}}{||\tilde{\mathbf{k}}||} \geq 0 \\ \tilde{\mathbf{x}} - (\tilde{\mathbf{x}} \cdot \frac{\tilde{\mathbf{k}}}{||\tilde{\mathbf{k}}||})\frac{\tilde{\mathbf{k}}}{||\tilde{\mathbf{k}}||} & \text{otherwise} \end{cases} \quad \forall \, \tilde{\mathbf{x}} \in \tilde{\mathbf{X}} \tag{19}$$

where $\tilde{\mathbf{k}} = \mathbf{U}\tilde{\mathbf{X}}$ for a learnable weight matrix $\mathbf{U} \in \mathbb{R}^{1 \times q}$, and where $\tilde{\mathbf{x}} \in \mathbb{R}^3$.

By composing series of linear transformations and equivariant activations, VN-MLPs map $\tilde{\mathbf{X}} \in \mathbb{R}^{q \times 3}$ to $\tilde{\mathbf{X}}' \in \mathbb{R}^{q' \times 3}$ such that $\tilde{\mathbf{X}}'\mathbf{R} = \text{VN-MLP}(\tilde{\mathbf{X}}\mathbf{R})$.

**VN-Inv**. Deng et al. (2021) also define learnable operations that map *equivariant* features $\tilde{\mathbf{X}} \in \mathbb{R}^{q \times 3}$ to *invariant* features $\mathbf{x} \in \mathbb{R}^{3q}$. In general, VN-Inv constructs invariant features by multiplying equivariant features $\tilde{\mathbf{X}}$ with other equivariant features $\tilde{\mathbf{Y}} \in \mathbb{R}^{3 \times 3}$ :

$$\hat{\mathbf{X}} = \tilde{\mathbf{X}}\tilde{\mathbf{Y}}^\top \tag{20}$$

The invariant features $\hat{\mathbf{X}} \in \mathbb{R}^{q \times 3}$ can then be reshaped into standard invariant features $\mathbf{x} \in \mathbb{R}^{3q}$. In our work, we slightly modify Deng et al. (2021)'s original formulation. Given a set of equivariant features $\tilde{\mathbf{X}} = \{\tilde{\mathbf{X}}^{(i)}\} \in \mathbb{R}^{n \times q \times 3}$, we define a VN-Inv as:

$$\text{VN-Inv}(\tilde{\mathbf{X}}) = \mathbf{X} \tag{21}$$

where $\mathbf{X} = \{\mathbf{x}^{(i)}\} \in \mathbb{R}^{n \times 6q}$ and:

$$\mathbf{x}^{(i)} = \text{Flatten}(\tilde{\mathbf{V}}^{(i)}\tilde{\mathbf{T}}_i^\top) \tag{22}$$

$$\tilde{\mathbf{V}}^{(i)} = (\tilde{\mathbf{X}}^{(i)}, \sum_i \tilde{\mathbf{X}}^{(i)}) \text{ if } n > 1 \text{ otherwise } \tilde{\mathbf{X}}^{(i)} \tag{23}$$

$$\tilde{\mathbf{T}}_i = \text{VN-MLP}(\tilde{\mathbf{V}}^{(i)}) \tag{24}$$

where $\tilde{\mathbf{T}}_i \in \mathbb{R}^{3 \times 3}$, and $\tilde{\mathbf{V}}^{(i)} \in \mathbb{R}^{2q \times 3}$ ($n > 1$) or $\tilde{\mathbf{V}}^{(i)} \in \mathbb{R}^{q \times 3}$ ($n = 1$).

**VN-DGCNN.** Deng et al. (2021) introduce VN-DGCNN as an SO(3)-equivariant version of the Dynamic Graph Convolutional Neural Network (Wang et al., 2019). Given a point cloud $\mathbf{P} \in \mathbb{R}^{n \times 3}$, VN-DGCNN uses (dynamic) equivariant edge convolutions to update equivariant per-point features:

$$\tilde{\mathbf{E}}_{nm}^{(t+1)} = \text{VN-LeakyReLU}^{(t)}(\mathbf{\Theta}^{(t)}(\tilde{\mathbf{X}}_m^{(t)} - \tilde{\mathbf{X}}_n^{(t)}) + \mathbf{\Phi}^{(t)}\tilde{\mathbf{X}}_n^{(t)}) \tag{25}$$

$$\tilde{\mathbf{X}}_n^{(t+1)} = \sum_{m \in \text{KNN}_f(n)} \tilde{\mathbf{E}}_{nm}^{(t+1)} \tag{26}$$

where $\text{KNN}_f(n)$ are the k-nearest neighbors of point $m$ in feature space, $\mathbf{\Phi}^{(t)}$ and $\mathbf{\Theta}^{(t)}$ are weight matrices, and $\tilde{\mathbf{X}}_n^{(t)} \in \mathbb{R}^{q \times 3}$ are the per-point equivariant features.

### A.12 GRAPH NEURAL NETWORKS

In this work, we employ graph neural networks (GNNs) to encode:

- each atom/fragment in the library $\mathcal{L}_f$
- the target molecule $M_S$
- each partial molecular structure $M_l^{\prime(c)}$ during sequential graph generation
- the query structures $M_{l+1}^{\prime(\psi_{\text{foc}})}$ when scoring rotatable bonds

Our GNNs are loosely based upon a simple version of the EGNN (Satorras et al., 2022b). Given a molecular graph $G$ with atoms as nodes and bonds as edges, we use graph convolutional layers defined by the following:

$$\mathbf{m}_{ij}^{t+1} = \phi_m^t \left( \mathbf{h}_i^t, \mathbf{h}_j^t, ||\mathbf{r}_i - \mathbf{r}_j||^2, \mathbf{m}_{ij}^t \right) \tag{27}$$

$$\mathbf{m}_i^{t+1} = \sum_{j \in N(i)} \mathbf{m}_{ij} \tag{28}$$

$$\mathbf{h}_i^{(t=1)} = \phi_h^{(0)}(\mathbf{h}_i^0, \mathbf{m}_i^{(t=1)}) \tag{29}$$
$$\mathbf{h}_i^{t+1} = \phi_h^t(\mathbf{h}_i^t, \mathbf{m}_i^{t+1}) + \mathbf{h}_i^t \quad (t > 0) \tag{30}$$

where $\mathbf{h}_i^t$ are the learned atom embeddings at each GNN-layer, $\mathbf{m}_{ij}^t$ are learned (directed) messages, $\mathbf{r}_i \in \mathbb{R}^3$ are the coordinates of atom $i$, $N(i)$ is the set of 1-hop *bonded* neighbors of atom $i$, and each $\phi_m^t, \phi_h^t$ are MLPs. Note that $\mathbf{h}_i^0$ are the initial atom features, and $\mathbf{m}_{ij}^0$ are the initial bond features for the bond between atoms $i$ and $j$. In general, $\mathbf{m}_{ij}^t \neq \mathbf{m}_{ji}^t$ for $t > 0$, but here $\mathbf{m}_{ij}^0 = \mathbf{m}_{ji}^0$. Note that since we only aggregate messages from directly bonded neighbors, $||\mathbf{r}_i - \mathbf{r}_j||$ only encodes bond distances, and does not encode any information about specific 3D conformations. Hence, our GNNs effectively only encode 2D chemical identity, as opposed to 3D shape.

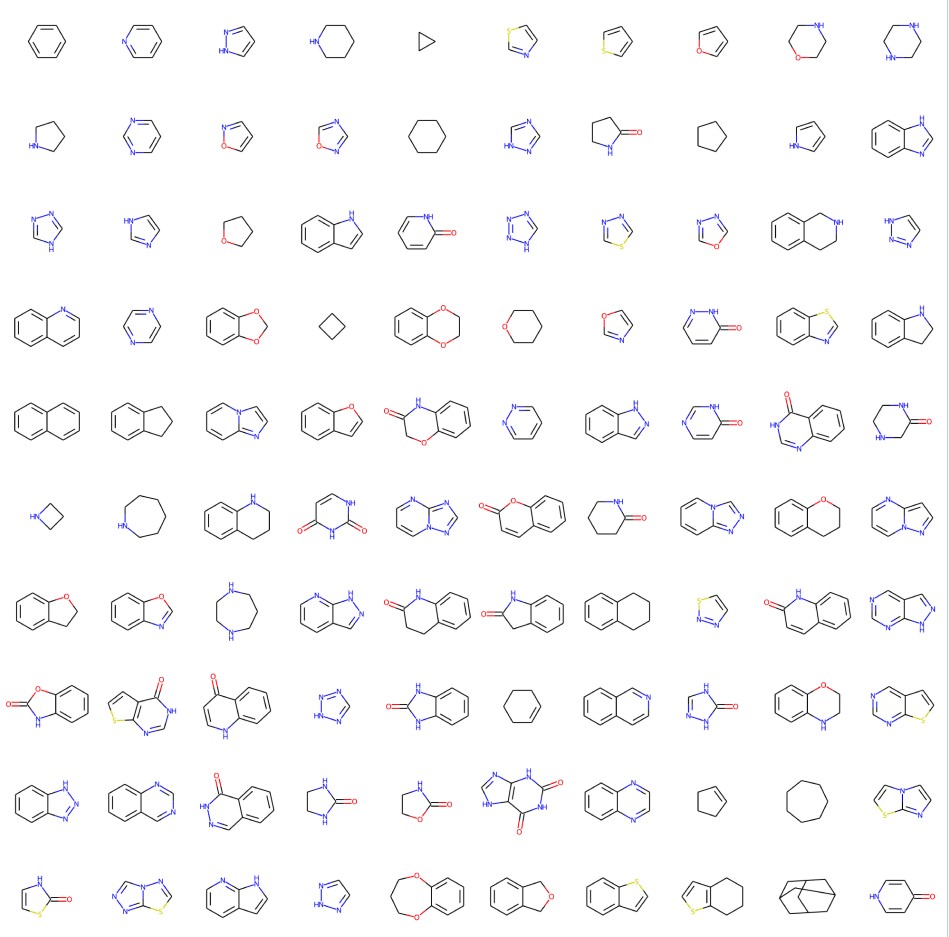

Figure 14: All 100 distinct fragments in our atom/fragment library $\mathcal{L}_f$.

## A.13 FRAGMENT LIBRARY.

Our atom/fragment library $\mathcal{L}_f$ includes 100 distinct fragments (Fig. 14) and 24 unique atom types. The 100 fragments were selected based on the top-100 most frequently occurring fragments in our training set. In this work, we specify fragments as ring-containing substructures that do not contain any acyclic single bonds. However, in principle fragments could be any (valid) chemical substructure. Note that we only use 1 (geometrically optimized) conformation per fragment, which is assumed to be rigid. Hence, in its current implementation, SQUID does not consider different ring conformations (e.g., boat vs. chair conformations of cyclohexane).

## A.14 MODEL PARAMETERS

**Parameter sharing.** For both the graph generator and the rotatable bond score, the (variational) molecule encoder (in the Encoder, Fig. 2) and the partial molecule encoder (in the Decoder, Fig. 2) share the same fragment encoder ($\mathcal{L}_f$-GNN), which is trained end-to-end with the rest of the model. Apart from $\mathcal{L}_f$-GNN, these encoders do not share any learnable parameters, despite having parallel architectures. The graph generator and the rotatable bond scorer are completely independent, and are trained separately.

**Hyperparameters.** Tables 6 and 7 tabulate the set of hyperparameters used for SQUID across all the experiments conducted in this paper. Table 8 summarizes training and generation parameters, but we refer the reader to App. A.15 and A.16 for more detailed discussion of training and generation protocols.

Because of the large hyperparameter search space and long training times, we did not perform extensive hyperparameter optimizations. We manually tuned the learning rates and schedulers to maintain training stability, and we maxed-out batch sizes given memory constraints. We set $\beta_{\emptyset\text{-shape}} = 10$ and $\beta_{\text{next-shape}} = 10$ to make the magnitudes of $L_{\emptyset\text{-shape}}$ and $L_{\text{next-shape}}$ comparable to the other loss components for graph-generation. We slowly increase $\beta_{KL}$ over the course of training from $10^{-5}$ to a maximum of $10^{-1}$, which we found to provide a reasonable balance between $L_{KL}$ and graph reconstruction.

Table 6: Architecture hyperparameters for SQUID's graph generator. Many of the GNN, VN-DGCNN, Vn-Inv, VN-MLP, and MLP modules use the same set of hyperparameters (*not* learnable parameters), but some do not. Where it appears, ($\rightarrow$) indicates the outputs of the module(s) to which we refer (see Fig. 2). The graph generator has a total of 1468103 learnable parameters.

| Symbol | Parameter Description | Value |
|---|---|---|
| $\sigma_p^2$ | variance of atom-centered Gaussians | 0.049 Å$^2$ |
| $n_p$ | number of sampled points per atom | 5 |
| $d_f$ | dimensionality of atom/fragment embeddings | 64 |
| $q$ | $\tilde{\mathbf{X}} \in \mathbb{R}^{q \times 3}$ | 64 |
| $d_a$ | dimensionality of input atom/node features | 45 |
| $d_b$ | dimensionality of input bond/edge features | 5 |
| $d_h$ | dimensionality of atom embeddings | 64 |
| $q'$ | $\mathbf{W}_H^{(a)} \in \mathbb{R}^{q' \times q}$ | 32 |
| $d_z$ | dimensionality of $\mathbf{z}$ | 64 |
| $d_{\text{dec}}$ | dimensionality of $\mathbf{z}_{\text{dec}}$ | 448 |
| | $\phi_m^{(t)}$ (GNN$_{\mathcal{L}_f}$) [hidden, (output)] layer sizes | [64, (64)] |
| | $\phi_h^{(t)}$ (GNN$_{\mathcal{L}_f}$) [hidden, (output)] layer sizes | [64, (64)] |
| | Number of GNN$_{\mathcal{L}_f}$ layers | 3 |
| | $\phi_m^{(t=0)}$ (Enc., Dec. GNNs) [hidden, (output)] layer sizes | [128, (64)] |
| | $\phi_h^{(t=0)}$ (Enc., Dec. GNNs) [hidden, (output)] layer sizes | [128, (64)] |
| | $\phi_m^{(t>0)}$ (Enc., Dec. GNNs) [hidden, (output)] layer sizes | [64, (64)] |
| | $\phi_h^{(t>0)}$ (Enc., Dec. GNNs) [hidden, (output)] layer sizes | [64, (64)] |
| | Number of (Enc., Dec.)-GNN layers | 3 |
| | Conv. dimensions for VN-DGCNN ($\rightarrow \tilde{\mathbf{X}}, \tilde{\mathbf{X}}'$) | [32, 32, 64, 128] |
| | Conv. pooling type for VN-DGCNN ($\rightarrow \tilde{\mathbf{X}}, \tilde{\mathbf{X}}'$) | mean |
| | Number of k-NN for VN-DGCNN ($\rightarrow \tilde{\mathbf{X}}, \tilde{\mathbf{X}}'$) | 5 |
| | VN-Inv ($\rightarrow \mathbf{X}, \mathbf{X}', \mathbf{X}_{\mathbf{H}_{\text{var}}}, \mathbf{X}'_{\mathbf{H}}$) [hidden, (output)] layer sizes | [64, 32, (3)] |
| | VN-Inv ($\rightarrow \mathbf{z}_{\text{dec}}$) [hidden, (output)] layer sizes | [(3)] |
| | VN-MLP ($\rightarrow \tilde{\mathbf{X}}_{\mathbf{H}_{\text{var}}}, \tilde{\mathbf{X}}'_{\mathbf{H}}$) [hidden, (output)] layer sizes | [(64)] |
| | VN-MLP ($\rightarrow \tilde{\mathbf{Z}}_{\text{dec}}$) [hidden, (output)] layer sizes | [128, 64, (64)] |
| | MLP ($\rightarrow (\mathbf{H}_\mu, \mathbf{H}_\sigma)$) [hidden, (output)] layer sizes | [64, (128)] |
| | MLP ($\rightarrow \mathbf{W}_{\mathbf{H}}, \mathbf{W}'_{\mathbf{H}}$) [hidden, (output)] layer sizes | [64, (2048)] |
| | MLP ($\rightarrow \mathbf{X}_{\mathbf{H}_{\text{var}}}, \mathbf{X}'_{\mathbf{H}}$) [hidden, (output)] layer sizes | [128, 64, (64)] |
| | MLP ($\rightarrow p_\emptyset, \mathbf{p}_{\text{focus}}, \mathbf{p}_{\text{next}}, \mathbf{p}_{\text{site}}, \mathbf{p}_{\text{bond}}$) [hidden] layer sizes | [64, 64, 64] |
| | MLP (all) hidden layer activation function | LeakyReLU(0.2) |

Table 7: Architecture hyperparameters for SQUID's rotatable bond scorer. Many of the GNN, VN-DGCNN, Vn-Inv, VN-MLP, and MLP modules use the same set of hyperparameters (*not* learnable parameters), but some do not. Where it appears, ($\rightarrow$) indicates the outputs of the module(s) to which we refer (see Fig. 2). The rotatable bond scorer has a total of 1270080 learnable parameters.

| Symbol | Parameter description | Value |
|---|---|---|
| $\sigma_p^2$ | variance of atom-centered Gaussians | 0.049 Å$^2$ |
| $n_p$ | number of sampled points per atom | 5 |
| $d_f$ | dimensionality of atom/fragment embeddings | 64 |
| $q$ | $\tilde{\mathbf{X}} \in \mathbb{R}^{q \times 3}$ | 64 |
| $d_a$ | dimensionality of input atom/node features | 45 |
| $d_b$ | dimensionality of input bond/edge features | 5 |
| $d_h$ | dimensionality of atom embeddings | 64 |
| $q'$ | $\mathbf{W}_H^{(a)} \in \mathbb{R}^{q' \times q}$ | 32 |
| $d_z$ | dimensionality of $\mathbf{z}$ | 64 |
| $d_{\text{dec}}$ | dimensionality of $\mathbf{z}_{\text{dec, scorer}}$ | 448 |
| | $\phi_m^{(t)}$ (GNN$_{\mathcal{L}_f}$) [hidden, (output)] layer sizes | [64, (64)] |
| | $\phi_h^{(t)}$ (GNN$_{\mathcal{L}_f}$) [hidden, (output)] layer sizes | [64, (64)] |
| | Number of GNN$_{\mathcal{L}_f}$ layers | 3 |
| | $\phi_m^{(t=0)}$ (Enc./Dec. GNNs) [hidden, (output)] layer sizes | [128, (64)] |
| | $\phi_h^{(t=0)}$ (Enc./Dec. GNNs) [hidden, (output)] layer sizes | [128, (64)] |
| | $\phi_m^{(t>0)}$ (Enc./Dec. GNNs) [hidden, (output)] layer sizes | [64, (64)] |
| | $\phi_h^{(t>0)}$ (Enc./Dec. GNNs) [hidden, (output)] layer sizes | [64, (64)] |
| | Number of (Enc./Dec.)-GNN layers | 3 |
| | Conv. dimensions for VN-DGCNN ($\rightarrow \tilde{\mathbf{X}}, \tilde{\mathbf{X}}'$) | [32, 32, 64, 128] |
| | Conv. pooling type for VN-DGCNN ($\rightarrow \tilde{\mathbf{X}}, \tilde{\mathbf{X}}'$) | mean |
| | Number of k-NN for VN-DGCNN ($\rightarrow \tilde{\mathbf{X}}, \tilde{\mathbf{X}}'$) | 10 |
| | VN-Inv ($\rightarrow \mathbf{X}, \mathbf{X}', \mathbf{X_H}, \mathbf{X'_H}$) [hidden, (output)] layer sizes | [64, 32, (3)] |
| | VN-Inv ($\rightarrow \mathbf{z}_{\text{dec}}$) [hidden, (output)] layer sizes | [(3)] |
| | VN-MLP ($\rightarrow \tilde{\mathbf{X}}_{\mathbf{H}_{\text{var}}}, \tilde{\mathbf{X}}'_{\mathbf{H}}$) [hidden, (output)] layer sizes | [(64)] |
| | VN-MLP ($\rightarrow \tilde{\mathbf{Z}}_{\text{dec}}$) [hidden, (output)] layer sizes | [128, 64, (64)] |
| | MLP ($\rightarrow \mathbf{W_H}, \mathbf{W'_H}$) [hidden, (output)] layer sizes | [64, (2048)] |
| | MLP ($\rightarrow \mathbf{X_H}, \mathbf{X'_H}$) [hidden, (output)] layer sizes | [128, 64, (64)] |
| | MLP (scoring) [hidden, (output)] layer sizes | [64, 64, 64, (1)] |
| | MLP (all) hidden layer activation function | LeakyReLU(0.2) |

Table 8: Summary of training and generation parameters. See App. A.15, A.16 for further descriptions on training and generation protocols.

| Symbol | Parameter description | Value |
|---|---|---|
| **Training parameters for graph generator** | | |
| $\beta_{\emptyset\text{-shape}}$ | weighting of $L_{\emptyset\text{-shape}}$ in loss | 10.0 |
| $\beta_{\text{next-shape}}$ | weighting of $L_{\text{next-shape}}$ in loss | 10.0 |
| $\beta_{KL}$ | weighting of $L_{KL}$ in loss | log-linear from $10^{-5}$ to $10^{-1}$ over 1M batches |
| | max. learning rate | $2.5 \times 10^{-4}$ |
| | min. learning rate | $5 \times 10^{-6}$ |
| | learning rate scheduler | exp. decay ($\gamma = 0.9$) every 50K batches |
| | max. batch size (generation sequences) | 400 |
| | # training iterations (batches) | 2M |
| **Training parameters for rotatable bond scorer** | | |
| | max. learning rate | $5 \times 10^{-4}$ |
| | min. learning rate | $1 \times 10^{-5}$ |
| | learning rate scheduler | exp. decay ($\gamma = 0.9$) every 50K batches |
| | max. # of focus bonds per batch | 32 |
| | # of query dihedrals $\psi_{\text{foc}}$ per focus bond, per batch | 10 |
| | # training iterations (batches) | 2M |
| **Generation parameters** | | |
| $\tau_{\emptyset}$ | threshold for stopping local generation | 0.01 |
| | Number of scored dihedrals $\psi_{\text{foc}}$ per focus | 36 |

39

### A.15 ADDITIONAL TRAINING DETAILS

**Dataset.** We use molecules from MOSES (Polykovskiy et al., 2020) to train, validate, and test SQUID. Starting from the train/test sets provided by MOSES, we first generate an RDKit conformer for each molecule, and remove any molecules for which we cannot generate a conformer. Conformers are initially created with the ETKDG algorithm in RDKit, and then separately optimized for 200 iterations with the MMFF force field. We then fix the acyclic bond distances and bond angles for each conformer (App. A.8). Using the molecules from MOSES's train set, we then create the fragment library by extracting the top-100 most frequently occurring fragments (ring-containing substructures without acylic bonds). We separately generate a 3D conformer for each distinct fragment, optimizing the fragment structures with MMFF for 1000 steps. Given these 100 fragments, we then remove all molecules from the train and test sets containing non-included fragments. From the filtered training set, we then extract 24 unique atom types, which we add to the atom/fragment library $\mathcal{L}_f$. We remove any molecule in the test set that contains an atom type not included in these 24. Finally, we randomly split the (filtered) training set into separate training/validation splits. The training split contains 1058352 molecules, the validation split contains 264589 molecules, and the test set contains 146883 molecules. Each molecule has one conformer.

**Collecting training data for graph generation and scoring**. We individually supervise each step of autoregressive graph generation and use teacher forcing. We collect the ground-truth generation actions by representing each molecular graph as a tree whose root tree-node is either a terminal atom *or* a terminal fragment in the graph. A "terminal" atom is only bonded to one neighboring atom. A "terminal" fragment has only one acyclic (rotatable) bond to a neighboring atom/fragment. Starting from this terminal atom/fragment, we construct the molecule according to a breadth-first-search traversal of the generation tree (see Fig. 2); we break ties using RDKit's canonical atom ordering. We augment the data by enumerating all generation trees starting from each possible terminal atom/fragment in the molecule. For each rotatable bond in the generation trees, we collect regression targets for training the scorer by following the procedure outlined in App. A.2.

**Batching**. When training the graph generator, we batch together graph-generative actions which are part of the same generation sequence (e.g., generating $G_l'^{(c)}$ from $G_l'^{(c-1)}$). Otherwise, generation sequences are treated independently. When training the rotatable bond scorer, we batch together different query dihedrals $\psi_{\text{foc}}$ of the same focal bond. Rather than scoring all 36 rotation angles in the same batch, we include the ground-truth rotation angle and randomly sample 9/35 others to include in the batch. Within each batch (for both graph-generation and scoring), all the encoded molecules $M_S$ are constrained to have the same number of atoms, and all the partial molecular structures $G_l'^{(c)}$ are constrained to have the same number of atoms. This restriction on batch composition is purely for convenience: the public implementation of VN-DGCNN from Deng et al. (2021) is designed to train on point clouds with the same number of points, and we construct point clouds by sampling a (fixed) $n_p$ points for each atom.

**Training setup.** We train the graph generator and the rotatable bond scorer separately.

For the graph generator, we train for 2M iterations (batches), with a maximum batch size of 400 (generation sequences). We use the Adam optimizer with default parameters. We use an initial learning rate of $2.5 \times 10^{-4}$, which we exponentially decay by a factor of 0.9 every 50K iterations to a minimum of $5 \times 10^{-6}$. We weight the auxiliary losses by $\beta_{\text{next-shape}} = 10.0$ and $\beta_{\emptyset\text{-shape}} = 10.0$. We log-linearly increase $\beta_{KL}$ from $10^{-5}$ to $10^{-1}$ over the first 1M iterations, after which it remains constant at $10^{-1}$. For each generation sequence, we randomize the rotation angle of the bond connecting the focus to the rest of the partial graph (e.g., the focal dihedral), as this dihedral has yet to be scored. In order to make the graph generator more robust to imperfect rotatable bond scoring at generation time, during training, we perturb the dihedrals of each rotatable bond in the partially generated structure $M_l'$ by $\delta\psi \sim N(\mu = 0°, \sigma = 15°)$ while fixing the coordinates of the focus.

For the rotatable bond scorer, we train for 2M iterations (baches), with a maximum batch size of 32 (focal bonds). Since we sample 10 query dihedrals per focal bond, the effective (maximum) batch size is 320. We use the Adam optimizer with default parameters. We use an initial learning rate of $5 \times 10^{-4}$, which we exponentially decay by a factor of 0.9 every 50K iterations to a minimum of $1 \times 10^{-5}$. In order to make the scorer more robust to imperfect bond scoring earlier in generation

sequence, during training, we perturb the dihedrals of each rotatable bond in the partially generated structure $M_l'$ by $\delta\psi \sim N(\mu = 0°, \sigma = 5°)$ while fixing the coordinates of the focus.

**Training times.** Training the graph generator on 1 GPU takes $\sim$ 5-7 days. Training the rotatable bond scorer on 1 GPU takes $\sim$ 5 days.

**Dataset processing times.** Pre-computing the regression targets for the rotatable bond scorer is the most compute-intensive part of dataset pre-processing. We required $\sim$ 2-3 days when parallelizing across 384 cpu cores.

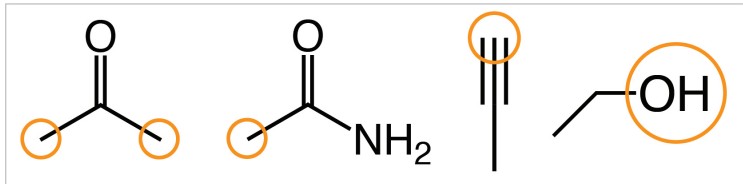

Figure 15: Examples of starting seed structures $M_0$ highlighting where new atoms/fragments might be attached (orange).

### A.16 ADDITIONAL GENERATION DETAILS

**Seeding generation.** When generating new molecules $M'$ from $M_S$, we start by extracting a small structure $M_0$ from $M_S$ to seed generation. In our experiments, $M_0$ is determined by the following criteria:

- $M_0$ has $\leq 6$ heavy atoms
- $M_0$ contains at least 1 terminal atom (an atom with only 1 bonded neighbor)
- Any new atom/fragment that is bonded to $M_0$ forms a valid dihedral. For this to be satisfied, $M_0$ must contain at least 3 atoms.
- $M_0$ does *not* contain a sequence of 4 atoms that forms a valid dihedral of a flexible rotatable bond. Hence, all rotatable bonds are scored during generation.

Figure 15 shows some examples of common seed structures. If there are multiple suitable $M_0$ for a given $M_S$, we arbitrarily select one of them. We only consider one $M_0$ per $M_S$ in our experiments. Once we extract $M_0$, we *fix* the coordinates of $M_0$ to their ground truth positions in $M_S$.

We note that the above seeding procedure could be replaced by a suitable model that builds and (equivariantly) predicts the coordinates of the starting structure $M_0$. In this work, we extract seeds for simplicity, but use the above criteria to ensure that the downstream shape-conditioned 3D generation and design tasks are non-trivial. We also emphasize that our choice of seeding procedure is fairly arbitrary, and does not necessarily restrict how the model is used in other generation tasks. For instance, larger seed structures could be used.

**Masking invalid actions.** When generating the molecular graph, we mask actions which would violate chemical valency. Because we define atom types in part by the number of single, double, and triple bonds the atom forms (including bonds to implicit hydrogens), we create valency masks by ensuring that each atom forms the correct number and types of bonds. When scoring rotatable bonds, we also mask query dihedrals that would lead to a severe steric clash ($<1\text{Å}$) with the existing partially generated molecular structure (App. A.2).

**Bond scoring**. When scoring a rotable bond at generation time, we sample 36 dihedral angles separated by $10°$. In principle, one could use finer discretization schemes to generate more refined 3D conformations, at the cost of speed and/or memory.

**Threshold for stopping local generation**. We manually tuned $\tau_\emptyset$, the threshold for local stopping, to $\tau_\emptyset = 0.01$ in order to balance tendencies of the model to 1) prematurely stop local generation (thus leading to molecules that do not completely fill the target shape), and 2) generate extra atoms/fragments around the focus (leading to steric crowding).

**Generation time.** Typically, generating (in serial, on a cpu) one 3D molecule with 20-30 atoms takes 2-3 seconds. Overall generation time (on a cpu) and memory cost significantly increase if many stereoisomers are enumerated when scoring any particular rotatable bond (App. A.2). We thus cap the number of enumerated stereoisomers to 32 (per focus).

## A.17 RELAXATION OF GENERATED GEOMETRIES

This work focuses on generating flexible drug-like molecules in specific 3D conformations that fit a target shape. Notably, SQUID generates 3D conformations using heuristic bond distances and bond angles. Consequently, the generated 3D conformations are explicitly constrained to have reasonable bonding geometries (e.g., local structures). Empirically, the generated molecules also have few steric clashes, with just 13% of the generated molecules having a steric clash under 2 Å. Nevertheless, the generated molecules are not necessarily in their minimum energy conformations. Note that this is by design, as SQUID is intentionally designed to generate molecules that fit *arbitrary* 3D molecular shapes. In this section, we relax the generated geometries with force field optimization and consider how this geometry relaxation affects the shape similarity to the target shape.

We use the following procedure to (locally) optimize a generated 3D conformer. Given a SQUID-generated 3D molecule, we first extract the molecule's 2D graph, and use RDKit to generate a new (unrelated) conformation of the molecule, with explicit hydrogens included. Note that explicit hydrogens (which are not natively generated by SQUID) are needed for accurate force-field geometry optimization. We then manually rotate each rotatable bond in the RDKit-generated conformer to the exact rotation angle that was natively generated by SQUID, thereby (approximately) yielding the same conformation that SQUID generated. The major differences between these new conformations and the original conformations are that the new conformations (1) include hydrogens, (2) have relaxed local bonding geometries, and (3) may have slightly different fragment geometries. Note that this is similar to the procedure that is performed in Appendix A.8, except now explicit hydrogens are included in the 3D structure. We then use the MMFF force field in RDKit to optimize the new 3D conformation, using either 20 or 50 optimization steps, and finally remove the hydrogens from the optimized conformation.

Figure 16 shows the distribution of the shape similarity between original (unrelaxed) SQUID-generated ($\lambda = 1.0$) conformations and their geometry-optimized counterparts. After 20 steps of MMFF optimization, approximately 68% of generated conformers have a shape similarity to their geometry-relaxed counterparts of at least 0.9. After 50 steps, this proportion only drops to 62%. Hence, the majority of generated conformations do not undergo large shape changes upon (local) geometry optimization. This indicates that the generated conformations, whilst not necessarily the minimum energy conformations, are for the most part geometrically reasonable. The largest changes in 3D shape upon geometry relaxation either occur due to steric clashes in the generated conformations ($\sim 13\%$ of generated molecules have a steric clash), or due to the model approximating all acyclic rotatable bonds as fully rotatable. In reality, some "rotatable" bonds actually have high barriers to rotation due to favorable orbital interactions. For instance, some amide bonds (such as those in peptides) prefer to be planar due to pi-orbital interactions and electron delocalization. In its current formulation, SQUID does not attempt to account for these kinds of preferred orientations of otherwise rotatable bonds.

We are especially interested in how relaxing the generated geometries affects the shape similarity to the target 3D shape. Figure 17 shows the distribution in $\text{sim}_S(M', M_S)$ for the best of $N_{\max} = 20$ SQUID-generated molecules $M'$ (sampled from the prior, $\lambda = 1.0$) with $\text{sim}_G(M', M_S) < 0.7$, before and after the generated molecules undergo geometry optimization. Importantly, relaxing the geometries of the generated 3D molecules *does not* have a large impact on the distribution of shape similarity to the target 3D shape. In particular, even after 50 steps of geometry optimization, the generated molecules are still significantly enriched in shape similarity to the target 3D shape compared to molecules randomly sampled from the dataset.

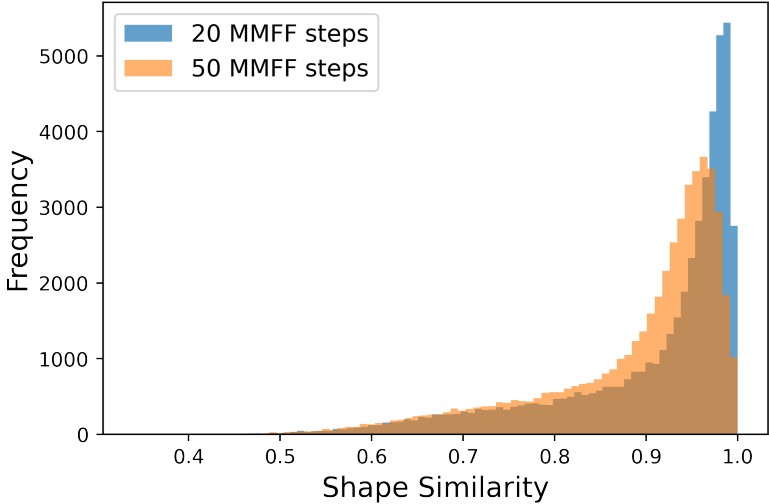

Figure 16: Distribution of shape similarity between SQUID-generated molecules sampled from the prior ($\lambda = 1.0$), and the same molecules after 20 (blue) or 50 (orange) steps of geometry relaxation with MMFF.

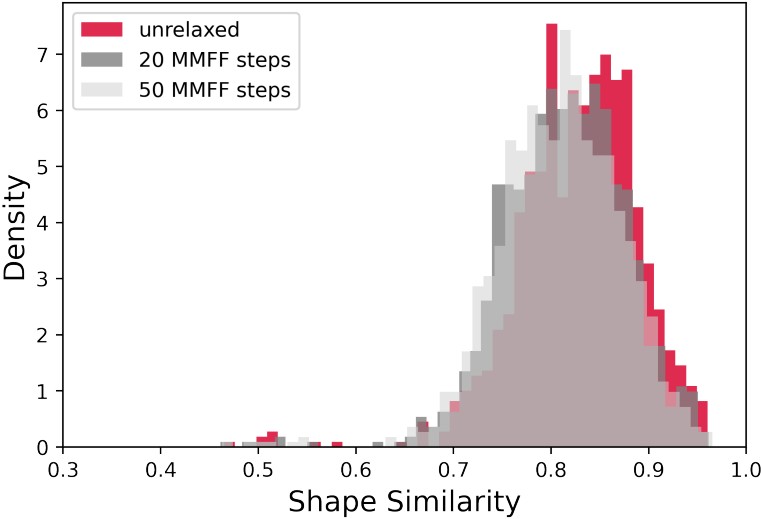

Figure 17: Distributions of $\mathrm{sim}_S(M', M_S)$ for the best of $N_{\max} = 20$ SQUID-generated ($\lambda = 1.0$) molecules $M'$ with $\mathrm{sim}_G(M', M_S) < 0.7$, before (red) or after each $M'$ has undergone 20 (dark grey) or 50 (light grey) steps of MMFF geometry relaxation.

A.18 Comparison to LigDream (Skalic et al., 2019)

In this section, we compare SQUID to LigDream (Skalic et al., 2019), which is a shape-captioning network that generates 1D SMILES strings of molecules conditioned on a 3D CNN encoding of a target 3D molecular shape and pharmacophores. In contrast to SQUID, LigDream (1) does not generate 3D conformers end-to-end, and hence requires *post hoc* conformer generation/enumeration; (2) is not robust to rotations, as LigDream's 3D CNN encoder is not SO(3)-invariant; and (3) does not separately encode molecular shape and chemical identity. Nevertheless, we still compare to LigDream to demonstrate SQUID's significant advantages for shape-conditioned molecular generation and design.

Skalic et al. (2019) train LigDream to convergence using $> 25$ million molecules from a drug-like subset of ZINC15. Rather than re-training LigDream on our significantly smaller dataset of $\sim 1$ million molecules, we directly apply Skalic et al. (2019)'s pre-trained LigDream on molecules from our test set. Note that our dataset is derived from MOSES, itself a subset of ZINC15, and hence the two models are trained on very similar molecules. We find this sufficient for the purposes of this comparison, with the caveat that LigDream is still trained on $\sim$20 times more data than SQUID.

To fairly and directly compare SQUID (a 3D model) to LigDream (a 1D/2D model requiring *post hoc* conformer generation), we perform the following procedure. Given a reference molecule $M_S$ with a target 3D shape, we encode $M_S$ using both SQUID ($\lambda = 1.0$) and LigDream. LigDream's encoder also employs a VAE; LigDream samples diverse molecules by controlling a "variability factor" $\lambda_{\text{LigDream}}$, which scales the variance of LigDream's posterior distribution. Following Skalic et al. (2019), we consider both $\lambda_{\text{LigDream}} = 1.0$ and $5.0$. After encoding $M_S$, we use SQUID and LigDream to generate a pool of 50 molecules, which we then filter to only include those with $\text{sim}_G(M', M_S) < 0.7$. Of the filtered molecules, we then sample $N_{\max} = 20$ molecules. For SQUID, we directly select the generated molecule from these $N_{\max}$ molecules that maximizes $\text{sim}_S(M', M_S)$. For LigDream, we use RDKit to generate $N_C$ conformations for each of the $N_{\max}$ generated molecules, and select the conformer amongst these $N_{\max} N_C$ total conformations that maximizes $\text{sim}_S(M', M_S)$. Since SQUID directly generates just 1 conformation per generated molecule, we use $N_C = 1$ for a head-to-head comparison.

Using this procedure, Figure 18 plots the distribution in $\text{sim}_S(M', M_S)$ for 1000 distinct $M_S$, using SQUID (prior, $\lambda = 1.0$) and LigDream ($\lambda_{\text{LigDream}} = 1.0, 5.0$). Overall, SQUID generates 3D molecules which are significantly more shape-similar to $M_S$ versus using LigDream with $\lambda_{\text{LigDream}} = 1.0$ or $5.0$. This is due to SQUID being able to directly generate 3D conformers that fit the target shape, and not relying on a non-shape-conditioned conformer generator (such as RDKit).

Importantly, the SQUID-generated molecules have an average *chemical* similarity to the encoded molecule ($\text{sim}_G(M', M_S)$) of 0.26, whereas the LigDream-generated molecules (using $\lambda_{\text{LigDream}} = 1.0$) have a higher average chemical similarity of 0.36. This is a substantial result, as we are interested in generating *chemically diverse* molecules that fit the target shape – not generating molecules that look like $M_S$. Note that SQUID can always generate more chemically similar molecules by decreasing $\lambda$, if desired. Increasing $\lambda_{\text{LigDream}}$ to $5.0$ causes LigDream to generate more chemically diverse molecules, with the average chemical similarity now on par with that obtained with SQUID. However, setting $\lambda_{\text{LigDream}} = 5.0$ causes the LigDream-generated molecules to be substantially less shape-similar to $M_S$. This emphasizes that 2D shape-conditioned generative models will struggle to generate *chemically diverse* molecules that still fit the target 3D shape, presumably because they do not simultaneously generate both the molecular graph and 3D coordinates (see Challenge 3). On the other hand, because SQUID (1) *separately* encodes 3D shape and 2D chemical identity, and (2) *simultaneously* generates both the molecular graph and the molecular coordinates, SQUID is able to generate more diverse molecules while still fitting the target 3D shape.

To test whether increasing $N_C$ can close the gap in the shape similarity distributions between SQUID and LigDream (with $\lambda_{\text{LigDream}} = 5.0$), we increase $N_C$ to 20. This essentially permits RDKit to sample more conformational space in order to search for conformations of the LigDream-generated molecules that best fit the target shape. Note that this is no longer a head-to-head comparison, as we are now comparing the best-of-**20** SQUID-generated conformations to the best-of-**400** LigDream(+RDKit)-generated conformations. Nevertheless, Figure 19 shows that even when using $N_C = 20$, SQUID still significantly outperforms LigDream in generating chemically diverse molecules with high shape similarity to the target 3D shape.

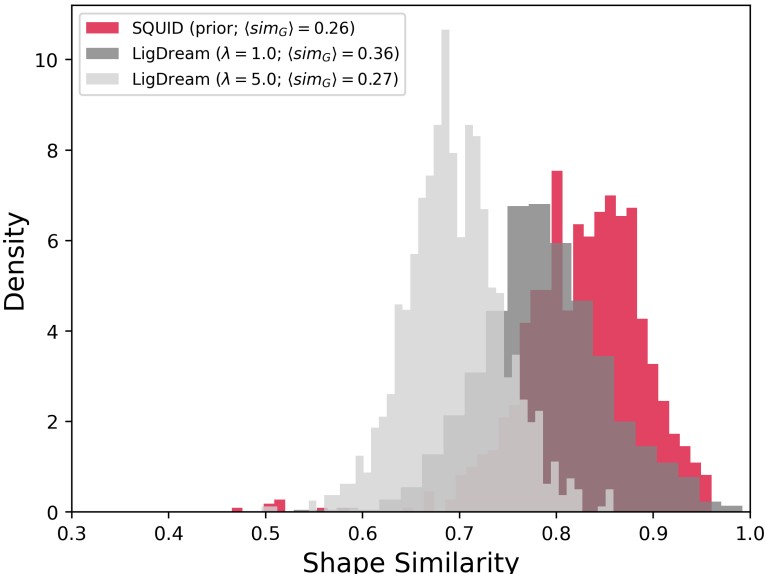

Figure 18: Distributions of $\text{sim}_S(M', M_S)$ for the best of $N_{\max} = 20$ SQUID-generated (prior, $\lambda = 1.0$) or LigDream-generated ($N_C = 1, \lambda = 1.0$ or $5.0$) molecules with $\text{sim}_G(M', M_S) < 0.7$. SQUID ($\lambda = 1.0$) generates more *chemically diverse* molecules compared to LigDream ($\lambda = 1.0$), and SQUID generates more *shape similar* molecules compared to LigDream ($\lambda = 5.0$).

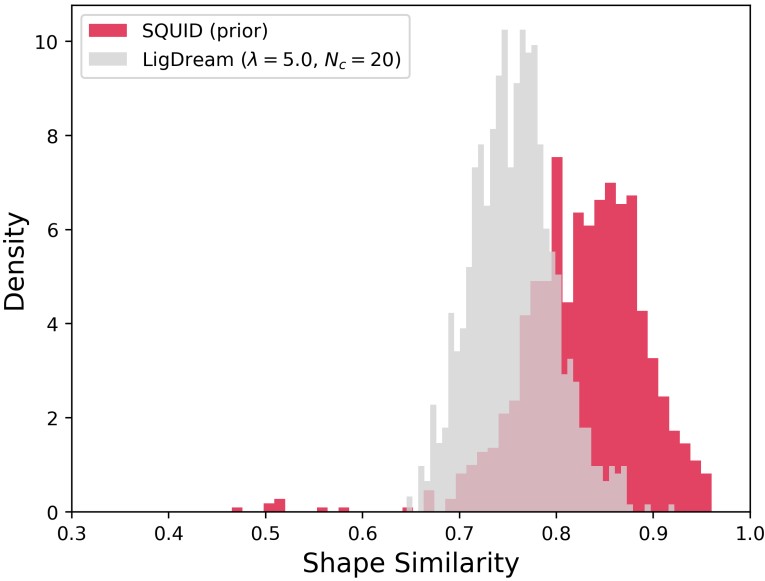

Figure 19: Distributions of $\text{sim}_S(M', M_S)$ for the best of $N_{\max} = 20$ SQUID-generated (prior, $\lambda = 1.0$) or LigDream-generated ($N_C = 20, \lambda = 5.0$) molecules with $\text{sim}_G(M', M_S) < 0.7$.

