# OpenReview forum: "Equivariant Shape-Conditioned Generation of 3D Molecules for Ligand-Based Drug Design"
_ICLR.cc/2023/Conference — ICLR 2023 poster_

### Official Review · Reviewer_TAEy · 2022-10-20

**Confidence:** 3
**Correctness:** 3
**Technical Novelty And Significance:** 4
**Empirical Novelty And Significance:** 2
**Recommendation:** 6

**Clarity, Quality, Novelty And Reproducibility:**

Clarity: Overall, the manuscript is well-written and to a sufficient level of detail so that it can be replicated. The Appendix is also quite thorough in regards to providing sufficient context for sections that for page limitations could not be extended appropriately.

Quality and novelty: As mentioned in the strength & weaknesses section, to my belief, this is the first method of its kind for the task of equivariantly generating molecules end-to-end conditioned on a specific pharmacophore and therefore deserves publication. In terms of quality, I believe that the manuscript could benefit from additional evaluations, particularly considering that there are other more basic techniques (e.g. 3D CNNs captioning networks that have been used for this same task in the past)

Reproducibility: The authors have provided supplementary code to reproduce most of the results provided in the manuscript.

**Strength And Weaknesses:**

Strengths:

* Manuscript very well written
* First approach of its kind, where the goal is to directly generate a 3D dimensional molecule end-to-end conditioned on a specific pharmacophore.
* Methodology shows potential for scaffold-hopping.
* Accompanying code to facilitate method testing


Weaknesses:
* Evaluation limited to the proposed method itself. While the proposed method is currently unique in the sense that it's fully end-to-end, the authors could have compared to other "multi-step" approaches such as the ones proposed by Skalic et al. (2019) or Imrie et al. (2021), even if the latter require additional conformation generation / alignment. From a technical point of view the model is novel but its applicability advantage compared to previous approaches remains unclear.


**Summary Of The Paper:**

In the proposed work, the authors present a method for de novo drug design conditioned on specific pharmacophores. They do so by using an autoregressive model similar to MoLeR (Maziarz et al 2021) and cleverly extending it to the 3D dimensional case with equivariant/invariant networks.

**Summary Of The Review:**

Overall, this is a good paper and believe that it is technically novel. I think, however, that it would benefit from an extended experiments section where the practical applicability of the method is compared to that of previous approaches.

Additional comments:
* One of the main selling points of shape-conditional generation is to generate molecules similar to a specific co-crystallized structure (i.e., in structure=based drug design). The authors could potentially use structure data to generate molecules for a given pharmacophore inside a protein pocket and compare how docking scores (without redocking, only scoring) compare to those of the reference compound.
* How is the p_{site} MLP computed? Since different fragments will have a variable number of attachment points, does that mean that each fragment type has its own separate MLP?
* Similarly are the MLPs of p_{focus}, p_{next} and p_{site} separate? If so, do they need a subscript?
* (On Table 5): Could the authors elaborate on why the graph reconstruction accuracy is so low? Is that a consequence of the autoregressive nature of the model? Similarly, could they comment on the amount of steric clashes (reported 13.1%)?
* (On the shape-constrained molecular optimization section) Do the authors use an existing genetic algorithm implementation or do they implement their own?
* Equations are lacking commas and periods when appropriate
* Figure 2 caption has a typo. ("encoder-decod er")

---

> ### Author Response · Authors · 2022-11-13
> **Authors' Response to Reviewer TAEy (Part 1)**
>
> We thank the reviewer for their comments and suggestions, and we provide detailed responses to each specific critique below. We will also provide an overall response to all the reviewers in a separate post. We kindly ask the reviewer to increase their recommendation if we address their concerns.
>
> > Evaluation limited to the proposed method itself. While the proposed method is currently unique in the sense that it's fully end-to-end, the authors could have compared to other "multi-step" approaches such as the ones proposed by Skalic et al. (2019) or Imrie et al. (2021), even if the latter require additional conformation generation / alignment. From a technical point of view the model is novel but its applicability advantage compared to previous approaches remains unclear…. In terms of quality, I believe that the manuscript could benefit from additional evaluations, particularly considering that there are other more basic techniques (e.g. 3D CNNs captioning networks that have been used for this same task in the past).
>
> Thank you for this suggestion. We compare to Skalic et al. (2019)’s model, denoted LigDream, in the new Appendix A.19. For convenience, we give a summary of the results here. Note that Imrie et al. (2021)’s model is designed to link existing fragments or elaborate upon an existing molecular scaffold, and thus is not as comparable to our model (which does *de novo* 3D generation, starting from a small seed).
>
> Skalic et al. (2019) train their model (LigDream) to convergence for ~10 days on 2 GPUs, using >25 million molecules. Because of the tight rebuttal deadline, we do not attempt to re-train and re-optimize LigDream on our much smaller dataset of only ~1 million molecules. However, their publicly available model is trained on ZINC15. Since our dataset (from MOSES) is itself a subset of ZINC, we believe it is a fair comparison to directly evaluate their pre-trained LigDream on molecules from our test set – with the caveat that LigDream is trained on 20X more molecules than SQUID.
>
> It can be difficult to establish a completely fair comparison between a natively 3D model (ours) and a disconnected workflow consisting of a 1D/2D model (LigDream) plus *post hoc* conformer generation/alignment. For a direct head-to-head comparison, we follow the procedure that we outlined in our Experiments section. Specifically, we use LigDream (with variability factor  = 1.0) to generate up to 50 molecules from an encoding of $M_S$ (using the same 1000 $M_S$ used to evaluate SQUID). From the 50 generated molecules per $M_S$, we then sample $N_\text{max}$ molecules that have $\text{sim}_G(M, M_S) < 0.7$. We then use RDKit to generate $N_C$ conformers for each of the $N_\text{max}$ molecules, and select the best overall conformer that maximizes $\text{sim}_S(M, M_S)$. Because our model generates just 1 conformer per molecule, the fairest comparison uses $N_C=1$. With ($N_\text{max} = 20$, $N_C = 1$), the mean best shape similarity of their generated conformers is ~0.78, compared to our model’s 0.83 (when using $N_\text{max} = 20$, $\lambda = 1.0$). This highlights the benefits of directly generating 3D molecules: our model can efficiently explore **3D** chemical space to directly find conformers that fit the shape.
>
> We also emphasize a key limitation of Skalic et al. (2019)’s model: it has difficulty generating *chemically diverse* molecules (low $\text{sim}_G$) whose conformers have high $\text{sim}_S$. The central motivation of our work is to develop a 3D shape-conditioned generative model that can generate **chemically diverse** molecules that fit a target shape – we are not interested in generating look-alike compounds with only small changes in chemical structure. Using the procedure above, the average chemical similarity of the LigDream’s best-of-$N_\text{max}$ molecules to $M_S$ is $\sim$0.36. For our model, the average chemical similarity to $M_S$ is significantly lower ($\sim$0.26), which is desired. If we increase LigDream’s variability factor to 5.0 in order to generate more chemically diverse molecules, then we can obtain a lower average chemical similarity of $\text{sim}_G \sim 0.27$. **However, this substantially reduces the average shape similarity (still using $N_\text{max} = 20$, $N_C=1$) to just 0.69.** Even if we increase $N_C$ to $N_C=20$ (with variability factor 5.0), the shape similarity of LigDream’s molecules are significantly lower compared to those generated by SQUID. This emphasizes that 1D/2D models struggle to explore *chemically diverse* shape-constrained molecular space, as they do not simultaneously generate both graph structure and 3D geometries. This is an important, novel, and practically useful achievement of our 3D model.
>
> We also note that LigDream uses standard 3D CNNs to encode shape, which means LigDream *is not* invariant/equivariant to rotations. In contrast, our model provides geometric symmetry guarantees, all while actually generating specific 3D conformers end-to-end.

---

> > ### Author Response · Authors · 2022-11-13
> > **Authors' Response to Reviewer TAEy (Part 2)**
> >
> > > One of the main selling points of shape-conditional generation is to generate molecules similar to a specific co-crystallized structure (i.e., in structure-based drug design). The authors could potentially use structure data to generate molecules for a given pharmacophore inside a protein pocket and compare how docking scores (without redocking, only scoring) compare to those of the reference compound.
> >
> > While one could certainly imagine applying our model in this setting (e.g., for structure-based drug design), we are primarily concerned with ligand-based drug design, where we may not have access to the protein structure. We note that shape-based virtual screens (e.g., with ROCS) are often used *instead* of docking-based screens, and can even lead to better results ([1]). We leave these sorts of structure-based studies to future protein-specific applications, which are outside the scope of this submission. We also note that this proposed study would primarily evaluate the well-tested hypothesis that shape similarity can be a good proxy for relating bioactivity. This hypothesis has been evaluated in numerous studies in drug discovery, and is one of the core reasons why ROCS and other shape-based similarity tools continue to be widely used in drug discovery.
> >
> > We also note that shape-similarity is useful in other chemical applications beyond drug discovery. Hence, our model is potentially useful for a broad range of applications in shape-based molecular design.
> >
> > [1] Hawkins et al., Comparison of Shape-Matching and Docking as Virtual Screening Tools, J. Med. Chem., 50(1), 74-82, 2007.
> >
> > > How is the $p_{site}$ MLP computed? Since different fragments will have a variable number of attachment points, does that mean that each fragment type has its own separate MLP?
> >
> > There is only 1 $p_{site}$ MLP. We use the MLP to predict a scalar (unnormalized) score for each attachment site on the fragment, and then softmax these scores for all attachment sites on the fragment in order to compute a cross-entropy loss from the fragment-normalized scores. Note that we feed the learned atom-features of each attachment site into the MLP when predicting that attachment site’s score. Also, note that this procedure is analogous to contrastive learning, where there can be a variable number of negative examples. Empirically, we find this method to work just fine.
> >
> > > Similarly are the MLPs of $p_{focus}$, $p_{next}$ and $p_{site}$ separate? If so, do they need a subscript?
> >
> > The MLPs of $p_{focus}$, $p_{next}$ and $p_{site}$ are separate. We have added subscripts to make this more clear.
> >
> > > (On Table 5): Could the authors elaborate on why the graph reconstruction accuracy is so low? Is that a consequence of the autoregressive nature of the model?
> >
> > The graph reconstruction accuracy is (relatively) low (15-60%) for a few reasons:
> >
> > 1. Our graph-decoder is inspired by MoLeR, which itself has a low reconstruction rate (<20%, based on private conversations with the authors of the MoLeR paper; the MoLeR paper does not report their graph reconstruction rate).
> >
> > 2. As the reviewer mentioned, the model is autoregressive, which can reduce reconstruction rate.
> >
> > 3. We report the reconstruction rate when sampling from the posterior for each atom’s chemical features, which leads to a significant source of stochasticity even when sampling chemical features from the encoded posterior. Note that the reconstruction rate increases significantly as we increase $N_\text{max}$, suggesting that stochasticity is a significant factor that reduces the graph reconstruction rate.
> >
> > 4. Our model has the additional complexity in that it generates 3D structure alongside the 2D graph. Imperfect 3D conformer generation can adversely influence subsequent graph generation steps. This also means that the model’s overall learning task is harder; reconstructing the 2D graph structure is not the only goal during training.
> >
> > 5. The shape encoder takes as input a sampled point cloud. The stochastic sampling of this point cloud introduces another source of randomness that can degrade reconstruction accuracy.
> >
> > 6. Non-optimal hyperparameters. We did not perform an extensive hyperparameter search to squeeze out all possible performance gains.
> >
> > 7. Most importantly, reconstructing graph identity was never our end-goal: we developed our model specifically to generate chemically diverse 3D molecules with high 3D shape similarity to a target.

---

> > > ### Author Response · Authors · 2022-11-13
> > > **Authors' Response to Reviewer TAEy (Part 3)**
> > >
> > > > Similarly, could they comment on the amount of steric clashes (reported 13.1%)?
> > >
> > > As a reminder, the reported amount of steric clashes (13%) indicates that approximately 13% of the final generated molecules have a steric clash, defined to be a non-bonded interatomic distance below 2 Å. Considering the state-of-the-art in 3D molecular generation, we consider this to be a very good result–especially since all the local geometries of our generated molecules are explicitly constrained to be geometrically reasonable. Also, consider that we did not train the model to explicitly avoid steric clashes; rather, we trained the rotatable bond scorer to optimize downstream shape similarity (which only implicitly penalizes steric clashes).
> > >
> > > While it would be ideal to have a model that does not generate any steric clashes, this is an unrealistic expectation given the current performance of prior 3D generative models for molecules – which by and large cannot reliably generate reasonable geometries for *de novo* flexible drug-like molecules (in free space).
> > >
> > > > (On the shape-constrained molecular optimization section) Do the authors use an existing genetic algorithm implementation or do they implement their own?
> > >
> > > We implement our own genetic algorithm that operates on the per-atom chemical features. We detail the exact algorithm in Appendix 5.
> > >
> > > > Equations are lacking commas and periods when appropriate. Figure 2 caption has a typo. ("encoder-decod er")
> > >
> > > These typos have been fixed. Thank you for pointing them out.
> > >
> > > > Empirical Novelty And Significance: 2: The contributions are only marginally significant or novel.
> > >
> > > It is worth reiterating that we have introduced a *novel* and uniquely *challenging* task for 3D generative models: generating **chemically diverse** molecules in 3D conformations that fit a target 3D shape, while respecting key geometric symmetries (rotational equivariance). In essence, our task probes the ability of 3D generative models to *separately* encode and explore (2D) chemical and (3D) conformational space. One could argue that this is the *primary* motivation for using (end-to-end) 3D models as opposed to 2D models for *de novo* molecular generation tasks. Yet, no studies have explicitly tried to explore shape-constrained chemical space with 3D generative models in a symmetry-robust manner.
> > >
> > > Besides the important empirical novelty and significance of this task, we also emphasize that our model mostly generates *realistic* 3D geometries (albeit not necessarily minimum energy geometries, by design) for very flexible drug-like molecules in an end-to-end fashion. Given notable limitations of prior work in 3D molecular generation, we do believe our work to have high empirical novelty and significance.

---

### Official Review · Reviewer_UfVq · 2022-10-23

**Confidence:** 4
**Correctness:** 3
**Technical Novelty And Significance:** 2
**Empirical Novelty And Significance:** 2
**Recommendation:** 6

**Clarity, Quality, Novelty And Reproducibility:**

The content can be understood with effort, but clarity can be still significantly improved. See weakness.1 for details.

The quality is pretty good. The author provides extensive content for both methods and experiments.

The tackled task is new but methodology novelty is limited from ML perspective. See Weaknesses 2&3 for details.

The author provides experimental setups in paper, and also submits their code. Reproducibility is pretty good.

**Strength And Weaknesses:**

Strength:

1. The tackled problem is new for the "ML for molecule" community, which hasn't been explored in previous literature.
2. The chemical background and challenges of this task are greatly explained, making the content friendly to the general ML audience.
3. The authors conduct comprehensive experiments and ablations to justify many different

Weakness:

1. Overall, the paper is a little hard to follow. As in Fig.2 and Sec.3, the author introduced too many intermediate variables within the neural network parameterizations, with just minor differences in subscripts. This is not very informative and makes the content not coherent enough. I suggest for the methodology part, try to just simply summarize the parameterization of the individual encoders & decoders; besides, make key parts such as the objective function and generation tree more clear and self-contained in the paper.
2. As shown in Sec.3 problem definition paper, the author claims to generate molecules with "similar shape" and "low similarity molecular representation". However, I think overall the proposed model just learns to generate existing molecules, without new objectives or RL  methods to explicitly encourage the exploration. The major difference is taking shape as additional inputs. However, as shown in Appendix A.9, actually the performance improvement by taking shapes as inputs are not significant.
3. Furthermore, the model actually is a combination of "molecular graph generative models" and "torsion scoring models". This makes me feel like you should also take existing molecular graph generation VAE models as baselines, such as [1]. For example, input $M_S$ into JT-VAE's encoder, and test whether decoder-generated molecules can maybe have competitive results with your model.

[1] Jin, Wengong, Regina Barzilay, and Tommi Jaakkola. "Junction tree variational autoencoder for molecular graph generation." In International conference on machine learning, pp. 2323-2332. PMLR, 2018.

**Summary Of The Paper:**

The paper studied shape-conditioned 3D molecule generation, which aims to generate molecules with a desirable shape. The author proposed an encoder-decoder architecture, where the encoder can encode both molecular graph representation and molecular shape representation by point clouds. Specifically, the main difference between the proposed model and existing molecular generative models is additionally taking shape (point clouds) as model inputs. Experiments show that the model can enable the shape-constrained generation and optimization of molecules.

**Summary Of The Review:**

The paper solves a new task of shape-conditioned molecule generation. I suggest significantly refining the content to make the paper easier to follow. Besides, comparing recent state-of-the-art VAE-based molecular generative models to justify the empirical improvements (see weakness 3).

---

> ### Author Response · Authors · 2022-11-13
> **Authors' Response to Reviewer UfVq (Part 1)**
>
> We thank the reviewer for their comments and suggestions, and we provide detailed responses to each specific critique below. We will also provide an overall response to all the reviewers in a separate post. We kindly ask the reviewer to increase their recommendation given our important clarifications.
>
> > Overall, the paper is a little hard to follow. As in Fig.2 and Sec.3, the author introduced too many intermediate variables within the neural network parameterizations, with just minor differences in subscripts. This is not very informative and makes the content not coherent enough. I suggest for the methodology part, try to just simply summarize the parameterization of the individual encoders & decoders; besides, make key parts such as the objective function and generation tree more clear and self-contained in the paper.
>
> While we respect the reviewer’s viewpoint, we believe that the detailed description of our model architecture is a core strength of our paper that ensures reproducibility, reduces misconceptions upon careful reading, and provides a detailed understanding of the complex equivariances of our model. When describing a complex model operating in both 2D and 3D, we view it to be imperative that the network operations are explicitly laid out without muddling – especially given the novelty of this task. The “minor differences in subscripts” are intended to allow the reader to relate information flow in the model (visualized in Figure 2), while still distinguishing distinct latent codes.
>
> Of course, conceptual understanding is also important. Figure 1 is intended to give a qualitative overview of the model, and we summarize the key conceptual motivations and design choices at the start of each section in the Methodology section. The other reviewers have stated that our work has **“very high clarity, quality and reproducibility”**, that the **“technical aspects of the method are well thought out and described in a clear fashion”**, and that **“the manuscript is well-written and to a sufficient level of detail so that it can be replicated”**.
>
> > As shown in Sec.3 problem definition paper, the author claims to generate molecules with "similar shape" and "low similarity molecular representation". However, I think overall the proposed model just learns to generate existing molecules, without new objectives or RL methods to explicitly encourage the exploration.
>
> By definition, generative models learn to sample from a distribution that is representative of the training set. Generating molecules which are highly unlike the support set is neither desirable nor practically useful in our setting, as that would lead to molecules that are not shape-similar to the target. **However, we emphasize that our model *does not* just generate existing molecules.** As clearly reported in the paper, the generated molecules have a 95% novelty rate (do not appear in the training set), a 99% uniqueness rate, and are *chemically dissimilar* compared to the encoded molecules (while still fitting the 3D shape, as is desired).
>
> Contrary to the reviewer’s statements, we do in fact explicitly encourage exploration with a genetic algorithm, which enables shape-constrained molecular optimization. We urge the reviewer to re-read our Experiments section where this is explained.
>
> > The major difference is taking shape as additional inputs. However, as shown in Appendix A.9, actually the performance improvement by taking shapes as inputs are not significant.
>
> This comment indicates an incorrect understanding of the motivations and results of the ablations performed in Appendix 9. The ablations performed in Appendix 9 do not ablate the shape encoding. Appendix 9 ablates the *equivariant alignment* between the encoded molecule (e.g., the target shape) and the partially generated molecule. *The ablated model still has access to the conditional shape encoding*; this is why there is still a notable improvement in the shape similarity distributions relative to the dataset baseline. In short, this ablation addresses Challenge 2, not Challenge 1 (see our introduction).
>
> ***However, just because a model encodes shape, does not mean that the model can use that shape information effectively.*** We have intentionally designed our equivariant network to use 3D shape information effectively – a primary technical novelty in our overall model design.
>
> In particular, Appendix 9 demonstrates that ablating the model’s equivariant alignment between the encoded/decoded molecules in representation space leads to a 33% reduction in shape similarity on average (relative to the dataset baseline, which is the lower bound for a well-trained conditional generative model). This reduction is indeed quite significant, and demonstrates the necessity of considering geometric equivariances and 3D representation alignment when “taking shape as inputs”.
>
> We request the reviewer to consider amending their review / score, given these important clarifications.

---

> > ### Author Response · Authors · 2022-11-13
> > **Authors' Response to Reviewer UfVq (Part 2)**
> >
> > > Furthermore, the model actually is a combination of "molecular graph generative models" and "torsion scoring models". This makes me feel like you should also take existing molecular graph generation VAE models as baselines, such as [1]. For example, input M_S into JT-VAE's encoder, and test whether decoder-generated molecules can maybe have competitive results with your model.
> >
> > While this is an understandable request, a non-shape-conditioned 2D model is effectively equivalent to our dataset baseline. Recall that our dataset baseline draws random samples from the training set and compares the shape similarity of the sampled molecules to the target shape. Any non-shape-conditioned 2D generative model, including JT-VAE, will learn to generate molecules that are representative of the training set (the support set). When sampling from the prior of these VAEs, the sampled molecules won’t be especially shape-similar to the target shape. Hence, our dataset baseline can be considered to be the *upper bound* on the performance of non-shape-conditioned 2D generative models such as VAEs, when sampling from the prior of these models.
> >
> > Of course, you could sample the posterior of these 2D VAEs to get molecules that are *chemically similar* to the encoded molecule $M_S$ (and which will also likely have high shape similarity upon conformer generation). However, our goal is *not* to generate molecules which are chemically similar to the encoded molecule. We seek to generate chemically diverse molecules (chemically dissimilar to the encoded molecule) while maintaining high shape-similarity. Hence, off-the-shelf non-conditioned 2D models are not a very relevant baseline.
> >
> > Besides, we are fundamentally interested in the task of shape-conditioned molecular generation in **3D space**. 3D generative models for molecules are a nascent field of study, even without considering shape-conditioning (a novel task).
> >
> > **As suggested by Reviewer TAEy, we have added a new comparison to a 2D generative model that *is conditioned on 3D shape*, and which is therefore a more relevant baseline than non-shape-conditioned 2D generative models.** These new comparisons are in Appendix A.19, and we urge the reviewer to read our detailed response to Reviewer TAEy. We emphasize that this shape-conditioned 2D model **does not** generate 3D conformers end-to-end, **cannot** efficiently explore 3D conformational space, and **is not** rotationally invariant/equivariant with respect to its shape encoding. Moreover, these comparisons empirically demonstrate that our 3D model has immense practical utility over 2D generative models.
> >
> > > The tackled task is new but methodology novelty is limited from ML perspective…The major difference is taking shape as additional inputs.
> >
> > We emphasize that our model is not merely a simple application of existing components to a new task, and does not simply take shape as inputs. As mentioned previously, to use shape information effectively, we develop a substantially novel equivariant/invariant architecture design that explicitly disentangles the encodings of 3D molecular shape and 2D molecular identity to enable the generation of *chemically diverse* molecules in specific conformations that fit a target 3D shape. *No work* has previously attempted to separate 3D shape embeddings from 2D chemical (graph) identity embeddings in a symmetry-robust manner. Moreover, our novel model design uniquely enables us to separately optimize 2D molecular properties while preserving 3D shape similarity to the target shape, all while actually generating 3D conformations end-to-end. *This has never been attempted before using machine learning.* This allows us to apply our model (without re-training) for shape-constrained molecular optimization, an important task in scaffold-hopping for drug design.
> >
> > Our model’s overall performance, robustness, and practical utility would be significantly harmed without our novel design choices, as Appendix 9 demonstrates.

---

> > ### Comment · Reviewer_UfVq · 2022-12-07
> > **Thanks for the clarifications**
> >
> > The authors have successfully addressed my major concerns, especially regarding the technical details. I'm glad to raise my rating.

---

### Official Review · Reviewer_vKuJ · 2022-10-25

**Confidence:** 4
**Correctness:** 3
**Technical Novelty And Significance:** 3
**Empirical Novelty And Significance:** 3
**Recommendation:** 6

**Clarity, Quality, Novelty And Reproducibility:**

The quality, clarity, and originality of this work is high.  The authors have provided code and links to the data to ensure reproducibility (though I haven't tested that assumption myself.)


**Strength And Weaknesses:**

The paper is a solid contribution to a reasonably recent area of exploration, the conditional generation of small molecules with 3D coordinates.  The technical aspects of the method are well thought out and described in a clear fashion, though I list a couple of remaining questions and omissions below. The authors demonstrate their method on a reasonable number of synthetic examples, and document its level of success.

One weakness in the presentation of this paper is the lack of a clear explanation of the rationale and implications from the restriction of chemical space to the combinatorial chemical landscape defined by the chosen set of 100 fragments and their non-ring linkers. I understand the need to demonstrate a method quickly, and perhaps this was the main motivation, however, while reading this work I kept wondering if perhaps the cost of a larger search tree perhaps becomes prohibitive and renders the method impractical. If that's not the case, would it be possible to simply generate molecules atom-by-atom, similar to previous 3D generation models, and let the model learn to enter rings and keep generating ring atoms until they close and so on? Alternatively, could one increase the size of fragments dramatically to include all possible single, double, and triple rings in any of the currently available patents?  Would such a change degrade the reconstruction quality?  Approximately how would the time for generation of a molecule (currently listed as 2--3 seconds of walltime) scale if one increased the size of the fragment library by a factor of 2, 5, 10, 100?

In terms of coordinates, the model is only learning to generate the rotatable dihedrals, which sounds like a rational way to evaluate a shape-matching model.  However, I wonder how much strain these dihedrals undergo in the final proposed conformation.  Instead of using the procedure to generate Figure 12, could the authors find the local minimum of the MMFF (the force field they used for generating the input conformations) and show the statistics of the shape similarity for these relaxed configurations?

Could the model drop the dependence on the initial fragment by learning the distribution of such initial fragments in the training set and learning to condition their initial placements on the shape?  I'd imagine that this problem is much easier than learning how to construct a reasonable search tree, and if it is harder than I imagine, then perhaps the authors could use an existing conditional pose generator, deep (or shallow) docking tool, or something else that is efficient as a starting point.  In that way, there will be no need to disclose any information about the test molecule that generated the shape for the final shape-matching task.

A confusing mistake in the paper is the improper description of the work in the pocket2mol paper, which also conditions on 3D inputs (the pocket) and generates complete molecules (not only fragments).  I strongly recommend that the authors improve their otherwise clear review of the related literature at the time of their submission, and amend the relevant clause in the second sentence of the abstract.

Finally, a question that doesn't need to be addressed in the paper, however, I would appreciate any thoughts that the authors might have on the subject below, and perhaps their answer could improve the paper after all.  The method that the authors use to align the molecules, ROCS, also has a GPU implementation that is reasonably scalable and efficient, and could thus render other, less specialized algorithms competitive (e.g. a traditional genetic algorithm that mixes graphs to iteratively optimize the shape scores in an ensemble, or a naive iterative trainable 3D molecule generator that scores and filters partial molecule graphs.)  Additionally, ROCS also can align and score molecules using both shape and color (chemical features)---I expect that a generalization that addresses color in addition to shape would be relevant in a practical drug discovery setting.  It would be trivial to modify an ensemble optimization algorithm to score via shape+color ROCS, however, typical ML methods would require training from scratch.  Is there a missing flexibility in the current wave of neural-network-based acceleration of chemoinformatics tasks and are there any generic ways to go around them to help build flexible building blocks (similar to what VNs did for building SO3 equivariant networks, but now for small molecule drug discovery)?


**Summary Of The Paper:**

This paper describes a method that can generate a sets of molecules that approximately match an input 3D "shape".  The method employs an autoregressive encoder-decoder architecture that learns a representation of the input shape (VN-DGCNN) and partial graph and decodes an updated molecule graph with an additional atom or fragment at each step (inspired by MoLeR). The method use a number of useful heuristics for bonds and angles to limit learning to the rotatable dihedral angles of the small molecule graphs. The authors demonstrate their model on a number of shape-conditioned tasks.


**Summary Of The Review:**

This paper is a useful, original contribution that can become stronger with minor modifications.

---

> ### Author Response · Authors · 2022-11-13
> **Authors' Response to Reviewer vKuJ (Part 1)**
>
>
> We thank the reviewer for their comments and suggestions, and we provide detailed responses to each specific critique below. We will also provide an overall response to the reviewers in a separate post. We kindly ask the reviewer to increase their recommendation if we address their concerns.
>
> > One weakness in the presentation of this paper is the lack of a clear explanation of the rationale and implications from the restriction of chemical space to the combinatorial chemical landscape defined by the chosen set of 100 fragments and their non-ring linkers. I understand the need to demonstrate a method quickly, and perhaps this was the main motivation, however, while reading this work I kept wondering if perhaps the cost of a larger search tree perhaps becomes prohibitive and renders the method impractical. If that's not the case, would it be possible to simply generate molecules atom-by-atom, similar to previous 3D generation models, and let the model learn to enter rings and keep generating ring atoms until they close and so on?
>
> There are multiple motivations for generating molecules (in-part) fragment-by-fragment rather than fully atom-by-atom. Specifically, atom-by-atom generative models are less efficient, prone to mistakes (e.g., not closing rings or generating distorted ring geometries), and struggle to generate large, flexible drug-like molecules with realistic 3D geometries. Consider the molecule O=C(C1=C(CC2CC2)C=CC1)CC3=CC(C4=CC=CC4)=CC(CC)=C3, which has 26 non-hydrogen atoms. To generate this 3D molecule using internal coordinates (predicting bond distances, bond angles, and dihedrals), one must score *at least* 23 dihedrals (e.g., the Z-matrix has 23 *independent* dihedrals). If any of the dihedrals associated with a ring isn’t precisely generated, the molecule will have a distorted 3D structure. In contrast, our model would only need to score 7 rotatable bonds to generate a very realistic 3D geometry. This makes 2D graph prediction **and** 3D geometry prediction efficient, and completely avoids the geometric pitfalls of ring completion. Since cyclic fragments relevant to drug discovery are relatively rigid, using fragments also does not significantly restrict the accessible 3D conformational space. Finally, atom-by-atom generators often generate complex fused cyclic systems that are not semantically sensible (e.g., could never be synthesized). By constraining generation to known fragments, you automatically gain a boost in synthetic reasonability. A number of works (for 2D generation) use fragment-based generation for similar reasons.
>
> While it is true that using a predefined set of fragments theoretically limits the combinatorial chemical space, in practice this isn’t particularly limiting. The chemical space one can enumerate with 100 common fragments is immense. Furthermore, prior 3D generative models that use fragment-based generation often use more restricted fragment libraries; for instance, [1] employs a fragment library that contains just 7 ring types.
>
> [1] Powers et al., Fragment-Based Ligand Generation Guided by Geometric Deep Learning on Protein-Ligand Structure, 2022.
>
> > Alternatively, could one increase the size of fragments dramatically to include all possible single, double, and triple rings in any of the currently available patents? Would such a change degrade the reconstruction quality? Approximately how would the time for generation of a molecule (currently listed as 2--3 seconds of walltime) scale if one increased the size of the fragment library by a factor of 2, 5, 10, 100?
>
> Yes, we could in principle increase our fragment library to cover more chemical space if desired – the choice of 100 was largely arbitrary. This would have a negligible impact on walltime, as encoding the 3D shape (which is unrelated to fragment prediction) is the most compute-intensive component of the model. At generation time, the fragment library needs to only be encoded once (a fixed cost). Beyond this, increasing the size of the library would only increase the output dimensions of the MLP that predicts which atom/fragment to add next, leading to only a marginal increase in model parameters and computational cost.
>
> Increasing the fragment library size may slightly degrade reconstruction quality, but only if the employed graph neural networks weren’t sufficiently expressive. The expressivity of graph neural networks is not the focus of our work. We do question the practical utility of using an extremely large fragment library (e.g., >1000). If such grand chemical spaces are required (unnecessary for most tasks in drug design), there are more specialized solutions that, for instance, might use hierarchical fragment generation.
>
> On the balance, the benefits of using fragment-based generation for **3D** molecular generation seems to vastly outweigh the downsides of limiting the accessible chemical space.

---

> > ### Author Response · Authors · 2022-11-13
> > **Authors' Response to Reviewer vKuJ (Part 2)**
> >
> > > In terms of coordinates, the model is only learning to generate the rotatable dihedrals, which sounds like a rational way to evaluate a shape-matching model. However, I wonder how much strain these dihedrals undergo in the final proposed conformation. Instead of using the procedure to generate Figure 12, could the authors find the local minimum of the MMFF (the force field they used for generating the input conformations) and show the statistics of the shape similarity for these relaxed configurations?
> >
> > We thank the reviewer for this suggestion. We have performed these additional experiments in Appendix A.18. Specifically, we perform 20-50 steps of MMFF optimization on the generated 3D geometries, following a very similar procedure as reported in Appendix 8 . Overall, (locally) relaxing the geometries of the generated 3D structures only slightly decreases the enrichment in shape similarity to the target 3D shape. Importantly, the shape similarity distributions for the relaxed geometries are still significantly enriched compared to our baselines.
> > Also, we note that our model is not explicitly trained to generate ground-state conformations. Rather, it is trained to generate specific conformations (of flexible molecules) that fit the target shape. Even so, because we use domain-informed heuristic bond angles and distances, the generated molecules predominantly have **realistic** 3D geometries. The generated molecules also have limited steric clashes (87% of molecules have no steric clashes under 2 Å), which is a primary contributor of torsional strain.
> >
> > Qualitatively, large changes in the 3D shape upon geometry relaxation are a result of our approximation that all acyclic single bonds are fully rotatable. In reality, some types of acyclic single bonds have preferred orientations. For instance, amide bonds (such as those in peptides) prefer to be planar, due to the pi-orbital delocalization (e.g., resonance effects). We did not explicitly try to account for these types of preferred orbital interactions in our model. In principle, our rotatable-bond scorer could be easily adapted to learn these preferred orientations, with or without explicit constraints, in order to generate more precise structures.
> >
> > > Could the model drop the dependence on the initial fragment by learning the distribution of such initial fragments in the training set and learning to condition their initial placements on the shape? I'd imagine that this problem is much easier than learning how to construct a reasonable search tree, and if it is harder than I imagine, then perhaps the authors could use an existing conditional pose generator, deep (or shallow) docking tool, or something else that is efficient as a starting point. In that way, there will be no need to disclose any information about the test molecule that generated the shape for the final shape-matching task.
> >
> > Yes, in principle the model could drop this dependence. As suggested by the reviewer, there are multiple available options if practitioners need to generate fully *de novo* compounds, without any sort of seed. In addition to those great suggestions, one could always add three atoms (e.g., a propyl group) to the tail-end of the encoded molecules, use these atoms as the seed, and then remove the seed after generation. Because of the many suitable options – which are primarily engineering-limited, not research-limited – we leave the exact choice to depend on the specific application one may be interested in.
> >
> > For our (highly functional) proof-of-concept model, seeding generation is by far the least complex option that does not significantly simplify the main task. As indicated by the reviewer, the primary research question is how to generate the vast majority of large, flexible, and chemically diverse molecules that fit a 3D shape. For this task, SQUID performs quite well.

---

> > > ### Author Response · Authors · 2022-11-13
> > > **Authors' Response to Reviewer vKuJ (Part 3)**
> > >
> > > > A confusing mistake in the paper is the improper description of the work in the pocket2mol paper, which also conditions on 3D inputs (the pocket) and generates complete molecules (not only fragments). I strongly recommend that the authors improve their otherwise clear review of the related literature at the time of their submission, and amend the relevant clause in the second sentence of the abstract.
> > >
> > > We cite Pocket2Mol in our Related Work section, as Pocket2Mol also uses Vector-Neuron operations. We do not ever mean to claim that Pocket2Mol only generates fragments; the last line of our Related Work section refers specifically to the 3DLinker paper (Huang et al., 2022).
> > >
> > > If we understand correctly, the reviewer may be taking issue with our comment in our abstract: “No existing models can reliably generate valid drug-like molecules in conformations that adopt a specific shape such as a known binding pose.” This comment was *intended* to emphasize that (1) existing generative models for 3D generation **in free space** struggle to generate **geometrically reasonable** and **chemically valid** drug-like molecules; and (2) no work (at the time of submission) explicitly enables the exploration of shape-conditioned (as opposed to protein-pocket conditioned) 3D chemical space.
> > >
> > > As far as *3D geometric validity* is concerned, even Pocket2Mol is not exempt from our observation. We commend the authors of Pocket2Mol for reporting KL divergences between bond distance/angle distributions for generated molecules vs. dataset molecules, as well as RMSD values between generated conformers and optimized conformers. *However*, the reported metrics for Pocket2Mol’s generated molecules, although outperforming its ML baselines, *do not* match those of test set molecules. Moreover, these metrics only indirectly quantify geometric sensibility. Distributional metrics such as the KL divergence tell us little about the global geometric validity of individual conformations, as a single bad bond distance/angle can completely ruin the sensibility of a 3D structure (and without significantly affecting the RMSD).
> > >
> > > In contrast, since our work uses heuristic bond distances/angles and rigid fragments, our generated molecules have proper local 3D geometries by design. But, our goal is not to compare to Pocket2Mol, or other pocket-conditioned generative models. Since we focus on ligand-based drug design rather than structure-based drug design, we will remove the clause “such as a known binding pose”, which may be confusing and lead to misinterpretation. In our updated submission, the sentence now reads: “No existing models can reliably generate geometrically realistic drug-like molecules in conformations that adopt a specific shape.” Please let us know if there is still an issue with this statement, but we find it to be a fair assessment of prior 3D generative models for molecules.

---

> > > > ### Author Response · Authors · 2022-11-13
> > > > **Authors' Response to Reviewer vKuJ (Part 4)**
> > > >
> > > > > Finally, a question that doesn't need to be addressed in the paper, however, I would appreciate any thoughts that the authors might have on the subject below, and perhaps their answer could improve the paper after all. The method that the authors use to align the molecules, ROCS, also has a GPU implementation that is reasonably scalable and efficient, and could thus render other, less specialized algorithms competitive (e.g. a traditional genetic algorithm that mixes graphs to iteratively optimize the shape scores in an ensemble, or a naive iterative trainable 3D molecule generator that scores and filters partial molecule graphs.)
> > > >
> > > > One could feasibly combine ROCS with other 2D or 3D graph-generative models for shape optimization or *post hoc* shape-filtering. We would argue that this strategy is not efficient, and does not guarantee desired results. If one only generates a 2D graph, then one must use a conformer generator to actually generate a 3D structure that can be scored with ROCS. For large molecules, this conformer generation can be quite expensive. Moreover, one may require an extensive conformer search to actually find the specific 3D conformation that best fits the target shape.
> > > >
> > > > If one uses a naive (not shape-conditioned) 3D generative model with *post hoc* shape-filtering, then you may avoid the conformer enumeration. But, you would likely have to generate many 3D structures in order to get a suitable structure that is shape-similar – this is akin to our “dataset” baseline in Figure 3. Since 3D generators are typically slow (relative to ROCS, at least), this is not an optimal strategy, especially since existing 3D generators struggle to generate *geometrically* and *chemically* valid molecules.
> > > > In contrast, our model directly predicts 3D structures and doesn’t require conformer enumeration – SQUID automatically finds conformations of (flexible) molecules that fit the shape.
> > > >
> > > > These naive 2D/3D methods will also likely be ineffective if you want molecules with optimized 2D properties, but which still fit the target shape. Our model enables this hybrid 2D/3D multi-objective optimization by encoding molecular shape and subsequently optimizing chemical features with a genetic algorithm, affording a much more efficient way of finding molecules that have a target 3D shape *and* have optimized chemical properties.
> > > >
> > > > > Additionally, ROCS also can align and score molecules using both shape and color (chemical features)---I expect that a generalization that addresses color in addition to shape would be relevant in a practical drug discovery setting. It would be trivial to modify an ensemble optimization algorithm to score via shape+color ROCS, however, typical ML methods would require training from scratch.
> > > >
> > > > Extending our model to colored ROCS scores would indeed be an interesting extension, though is out of the scope of this submission, which primarily focuses on exploring shape-conditioned chemical space with a model that explicitly separates 3D shape and 2D graph encodings. Nevertheless, there are multiple ways to include color features inside SQUID’s current implementation, if desired. For instance, one could simply use the genetic algorithm to optimize the color score, without requiring the model to be re-trained. Alternatively, one could adapt the shape encoder to also encode color features in addition to shape features, and condition 3D molecular generation on both.
> > > >
> > > > > Is there a missing flexibility in the current wave of neural-network-based acceleration of chemoinformatics tasks and are there any generic ways to go around them to help build flexible building blocks (similar to what VNs did for building SO3 equivariant networks, but now for small molecule drug discovery)?
> > > >
> > > > There is a tradeoff between developing highly generalizable neural-network-based tools for a broad range of cheminformatic tasks in drug discovery, and actually having those tools be useful and effective for each subtask. Cheminformatics and drug discovery is a long-standing field, with strong non-neural baselines. While broad flexibility is a good long-term goal, we believe the most impact in the near-term will occur with domain-specific models that properly account for the geometric symmetries and chemical priors that are relevant for each task.

---

> > ### Comment · Reviewer_vKuJ · 2022-12-01
> > **limitations of the fragment-based demonstration.**
> >
> > One of the practical limitations when using a model that cannot express some possible chemistries at all is that the fine-tuning or training on a new subset of molecules will inevitably hit a wall and require re-evaluating the full system from scratch.  The example molecule that the authors provided in their rebuttal indeed has a large number of dihedral angles, however, almost all of them would be flat or trivial and any reasonable DL model should quickly learn to not waste capacity on them.  If one modified a couple atoms to nitrogens, or closed this molecule to make a macrocycle, however, suddenly the authors model may no longer be able to deal with it as an input during training, which is a bigger concern than the inability to write out / generate such a modified molecule.  I do not clearly see the balance that the authors mention, and I would guess that even 1K or 10K fragments would be super limited in terms of the complexity of chemical space seen in practical applications.  I'd have liked to see a practical demonstration of the performance of the method, however, because I would have liked to know if such fragmentation perhaps works well on contrived examples and starts to fail in practice, which the authors have the ability to test.  In any case, I still think that this paper is at the borderline for this conference, and I will retain my previous score.

---

> > > ### Author Response · Authors · 2022-12-09
> > > **Authors' Response to "limitations of the fragment-based demonstration" (Part 1/2)**
> > >
> > > Although we understand the reviewer’s concerns about the expressivity of fragment-based molecular generation, we emphasize that this is not a major focus of our work. Our primary contribution is on how to generate chemically-diverse, large, flexible molecules in realistic conformational poses that fit a desired 3D shape -- essentially, how to encode, explore, and decode a shape-conditioned 3D chemical space. Previous atom-by-atom 3D molecular decoders are not currently amenable for this task, in large part because they are not able to generate realistic 3D geometries for flexible drug-like molecules. Fragment-based generation is thus a tool that allows us to investigate and achieve our primary goal.
> > >
> > > To alleviate the reviewer’s concerns, we respond to their specific comments below. But, we again emphasize that remedying the limitations of fragment-based generation, which is an exceedingly common framework within the community, is not our focus.
> > >
> > > > One of the practical limitations when using a model that cannot express some possible chemistries at all is that the fine-tuning or training on a new subset of molecules will inevitably hit a wall and require re-evaluating the full system from scratch.
> > >
> > > In practice, all models will have failure modes when applied in settings outside their training distribution, or when re-training on a new dataset/task meaningfully different from what it was developed on. For instance, most previous 3D generative models developed on tiny molecules (e.g., from QM9) will perform poorly when re-trained and applied to larger drug like molecules, even if they can technically "encode" the new chemistries. In our setting, we argue it is more important to design a 3D generative model that can actually generate realistic geometries, potentially at the cost of chemical space restrictions. But, we also note that our model may not have the limitations the reviewer indicates. We use a GNN-based fragment encoder to encode each fragment in our library; we do not one-hot encode the fragment library. As a result, our fragment-encoder could feasibly be applied to encode (and subsequently decode) other unique fragments that the model hasn’t seen during training. As long as the new fragments are somewhat close in chemical space to the original fragment set, our model will likely have good performance (with appropriate regularization) and hence not require full re-training from scratch.
> > >
> > > > The example molecule that the authors provided in their rebuttal indeed has a large number of dihedral angles, however, almost all of them would be flat or trivial and any reasonable DL model should quickly learn to not waste capacity on them.
> > >
> > > We agree that a reasonable DL model should be able to accurately predict dihedrals that are implicitly constrained by the generated graph structure (e.g., ring dihedrals). However, in practice, existing 3D molecular decoders do not (consistently) achieve this, especially when generating larger molecules. Because we focus on generating 3D molecules conditioned on a highly geometry-dependent property (e.g., molecular shape), it is important that our model does not “cheat” by generating unrealistic geometries that technically fit the shape. Fragment-based generation allows us to ensure that the model generates realistic geometries.
> > >
> > > > If one modified a couple atoms to nitrogens, or closed this molecule to make a macrocycle, however, suddenly the authors model may no longer be able to deal with it as an input during training, which is a bigger concern than the inability to write out / generate such a modified molecule.
> > >
> > > As mentioned above, we use a graph neural network to encode each fragment (we don’t use a one-hot embedding over fragment types). If we “modified a couple atoms to nitrogens”, then the fragment encoder could simply encode this different fragment structure. With suitable regularization, our GNN-based fragment encoder will encode a fragment representation space. Hence, our model does not necessarily suffer from the reviewer’s stated limitation.
> > >
> > > Encoding and decoding large macrocycles is its own representation learning challenge that, to our knowledge, no 3D molecular generative model tackles well. Often, non-fragment-based 3D generative models will generate highly constrained macrocycles that are not chemically stable or synthesizable, particularly if they rely on post-processing algorithms to extract the graph structure from a generated point cloud. While being able to generate semantically-sensible macrocycles would be nice feature of a generative model, it is outside the focus of our work.

---

> > > > ### Author Response · Authors · 2022-12-09
> > > > **Authors' Response to "limitations of the fragment-based demonstration" (Part 2/2)**
> > > >
> > > > > I do not clearly see the balance that the authors mention, and I would guess that even 1K or 10K fragments would be super limited in terms of the complexity of chemical space seen in practical applications.
> > > >
> > > > For a 3D generative model, there is a balance between developing a model that 1) can in theory capture all of 2D chemical space, and 2) is actually capable of generating proper 3D geometries for each molecule (without having access to the graph-structure beforehand). Since we focus on shape-conditioned molecular generation in 3D, our primary concern is the latter — actually generating realistic 3D molecular geometries (that fit a shape).
> > > >
> > > > We disagree with the reviewer’s claim that 1K or even 10K ring-containing fragments are insufficient to capture the vast majority of relevant drug-like chemical space. Taylor et al. (2014)’s investigation of ring systems in a database of 1175 **approved** drugs found that there were only 351 unique ‘ring systems’ (their definition of ‘ring system’ matches our definition of a ‘fragment’) [1]. Of these 351 ring systems, only 147 appeared more than once in the dataset. Hence, it would be entirely feasible to (moderately) increase our fragment library size in order to capture the vast majority of ring systems likely to be present in real small-molecule drugs.
> > > >
> > > > [1] Taylor, R., MacCoss, M., and Lawson, D. Rings in Drugs. *J. Med. Chem.*, 2014, 57, 14.
> > > >
> > > > > I'd have liked to see a practical demonstration of the performance of the method, however, because I would have liked to know if such fragmentation perhaps works well on contrived examples and starts to fail in practice, which the authors have the ability to test.
> > > >
> > > > We’re unsure of what specific experiments the reviewer is requesting. Although the time to revise the manuscript during the rebuttal period has long passed, we’d be happy to consider adding additional experiments concerning fragment expressivity for the camera-ready version.
> > > >
> > > >
> > > > We thank the reviewer for prompting these discussion points.

---

### Official Review · Reviewer_bX75 · 2022-10-25

**Confidence:** 5
**Correctness:** 3
**Technical Novelty And Significance:** 3
**Empirical Novelty And Significance:** 2
**Recommendation:** 6

**Clarity, Quality, Novelty And Reproducibility:**

The paper is under very high clarity, quality and reproducibility. However, the novelty to the machine learning community seems to be weak.

**Strength And Weaknesses:**

Strength:
1. The paper proposes a complex generative system for molecular generation with very detailed task-specific model design.
2. In order to build the molecules with 3D conformation, the proposed method introduces a novel scoring rotatable bonds to learn the rolling angles of each fragment.
3. The proposed method resolves a critical problem as scaffold hopping in drug design.
4. The experimental results show the benefits of the proposed method.

Weakness:
1. The contribution to the machine learning community is limited. The whole model is based on a CVAE framework with two embedding networks, VG-DGCNN and GNN and one fragment-based sequential generative decoder. The scoring rotatable bonds is a novel contribution but incremental to the ML community.
2. The rolling bonds of the molecules should contain periodicity. However, the output of scoring rotatable bonds is the output of a sigmoid function which is not a periodic function.
3. Since the base framework follows a CVAE, is the proposed method able to reconstruct the molecules in terms of the point cloud and 2D graph? It is better to report the validity, uniqueness, novelty, and reconstruction rate to indicate whether the model learns the information of molecules properly.
4. I am curious given a fixed shape, what is the diversity of the generated molecules? Are all generated conformers stable at the lowest energy and still fit to the required shape?


**Summary Of The Paper:**

The paper designs a novel framework to generate molecules regarding to the point cloud of molecules. The authors make a lot of effort into building the model consisting of VN-DGCNN, GNN, scoring rotatable bonds and so on.

**Summary Of The Review:**

I tend to reject the paper because I still doubt the contribution to the machine learning community. Although I believe it shows solid contribution to the drug discovery community. This work is more suitable to a journal of computational chemistry.

---

> ### Author Response · Authors · 2022-11-13
> **Authors' Response to Reviewer bX75 (Part 1)**
>
> We thank the reviewer for their comments, and we provide detailed responses to each specific critique below. We will also provide an overall response to all the reviewers in a separate post. We kindly ask the reviewer to increase their recommendation if we address their concerns.
>
> > The contribution to the machine learning community is limited. The whole model is based on a CVAE framework with two embedding networks, VG-DGCNN and GNN and one fragment-based sequential generative decoder. The scoring rotatable bonds is a novel contribution but incremental to the ML community.
>
> It is true that our model is based on a conditional variational autoencoder (CVAE) that uses VN-DGCNNs to encode molecular shape, GNNs to encode chemical identity, and a fragment-based decoder to generate diverse 3D molecules that fit the encoded shape. We do not claim novelty in our use of these popular submodules. However, we emphasize that employing well-established modules—within a completely novel framework designed for a completely novel task—neither precludes technical novelty nor diminishes our contributions to the ML community.
>
> **Specifically, we claim novelty in our unique combination of these existing tools to enable an entirely novel and effective approach in shape-conditioned 3D molecular design.** *No prior work* has attempted to explicitly disentangle the encodings of 3D molecular shape and 2D molecular identity for equivariant shape-conditioned molecular generation in 3D. Our encoder samples and mixes (invariant) chemical features with 3D (equivariant) shape features, enabling the decoder to generate *chemically diverse* 3D molecules in specific (realistic) conformations that natively fit the target 3D shape. Hence, we introduce the first 3D generative model that can explore shape-constrained 3D chemical space in a symmetry-robust way. Moreover, our model design uniquely enables us to separately optimize 2D molecular properties while preserving 3D shape similarity to the target shape, all while actually generating 3D conformations. *To our knowledge, this has never been attempted before using machine learning.* Hence, without retraining, our model can be used for shape-constrained scaffold-hopping, hit expansion, and ligand analogue generation, which are important tasks in drug design.
>
> > The rolling bonds of the molecules should contain periodicity. However, the output of scoring rotatable bonds is the output of a sigmoid function which is not a periodic function.
>
> We agree that a network which directly predicts angles should have periodic symmetry. **However, our rotatable bond scorer does not directly predict dihedral angles.** Given a partially-generated 3D molecule with a particular dihedral angle of the query rotatable bond, our scorer network directly predicts the maximum (downstream) shape similarity that is achievable given the queried 3D conformation. The shape similarity scores (Eq. 1) are bounded by [0,1], which is why we use the sigmoid function. We direct the reviewer to our Appendix 1, which elaborates further on how we train the rotatable bond scorer.
>
> > Since the base framework follows a CVAE, is the proposed method able to reconstruct the molecules in terms of the point cloud and 2D graph? It is better to report the validity, uniqueness, novelty, and reconstruction rate to indicate whether the model learns the information of molecules properly.
>
> We do indeed report the molecular validity, uniqueness, novelty, and (graph) reconstruction rate in Appendix 4, as well as in our main text (Experiments section). By design, our model achieves 100% chemical validity. The model also achieves 99% uniqueness and 95% novelty rates. The graph reconstruction rate (when chemical features are sampled from the posterior) ranges from 15-60% depending on how it is measured. However, reconstruction rate depends on many factors, including the type of decoder. MoLeR, the 2D fragment-based generative model which loosely inspires our 3D graph decoder, does not report their reconstruction rate. From private conversations with the authors of MoLeR, our reconstruction rate is on-par with what they experienced (<20%). Moreover, reconstruction rate is not necessarily an interesting or useful metric in our task, as our end-goal is to generate chemically diverse molecules that fit a target shape – not reconstruct known molecules. We refer the reviewer to our detailed response to Reviewer TAEy for further discussion on the reconstruction rate.
>
> Beyond these 2D metrics, our model also generates realistic 3D geometries of flexible drug-like molecules. Previous 3D generative models for molecules (in free space) are mostly limited to generating tiny molecules with few geometric degrees of freedom. When applied to larger drug-like molecules, prior models often generate unrealistic 2D graphs (e.g., generating complex ring structures) and unrealistic 3D geometries. Our model is a clear improvement in generating realistic molecules in 3D.

---

> > ### Author Response · Authors · 2022-11-13
> > **Authors' Response to Reviewer bX75 (Part 2)**
> >
> > > I am curious given a fixed shape, what is the diversity of the generated molecules?
> >
> > The average internal chemical similarity of the generated molecules for a given shape is $0.26 \pm 0.05$. This was measured by computing the average pairwise chemical (graph) similarity between pairs of generated molecules for a given encoded shape, when sampling chemical features from the prior ($\lambda = 1.0$). This indicates that the generated molecules are quite (chemically) diverse. For reference, the average chemical similarity between any two random molecules in the dataset is approximately $0.22 \pm 0.06$.
> >
> > Note that the molecular diversity, if desired, can be decreased by sampling closer to the posterior (using lower values of $\lambda$). For instance, for $\lambda = 0.3$, the average internal chemical similarity is $0.32 \pm 0.07$. This ability to tune the chemical diversity of the generated molecules, while still fitting the 3D shape, was an explicit design choice in our model.
> >
> > We have included these internal diversity metrics in Appendix 4.
> >
> > > Are all generated conformers stable at the lowest energy and still fit to the required shape?
> >
> > The generated conformers are not necessarily the lowest-energy conformers available to the generated molecules. But, in general, molecules often have many conformations which are scientifically interesting despite not being the lowest-energy. For instance, active binding poses are often not the lowest-energy conformation of the molecule.
> >
> > Nevertheless, we have performed additional experiments (Appendix A.18) where we perform 20-50 steps of MMFF optimization on the generated 3D geometries. Overall, relaxing the geometries of the generated 3D structures only slightly decreases the enrichment in shape similarity to the target shape. Importantly, the shape similarity distributions for the relaxed geometries are still significantly enriched compared to our baselines.
> >
> > Also, note that our model is not explicitly trained to generate ground-state conformations. Even so, because we use domain-informed heuristic bond angles and distances, the generated molecules predominantly have **realistic** 3D geometries. The generated molecules also have limited steric clashes (87% of molecules have no steric clashes under 2 Å), which is the primary contributor of torsional strain.
> >
> > > I tend to reject the paper because I still doubt the contribution to the machine learning community. Although I believe it shows solid contribution to the drug discovery community. This work is more suitable to a journal of computational chemistry.
> >
> > We respectfully request the reviewer to reconsider this viewpoint, especially considering our clarifying comments above and our discussion on our work’s ML contributions.
> >
> > While our paper undoubtedly has relevance for the chemistry community, we are steadfast in our belief that the Machine Learning for Science community (an ICLR-designated area) can immediately benefit from our work – a novel machine learning method for 3D shape-conditioned molecular design. A host of works for molecular generation (in 2D and 3D) are routinely submitted and published at ML venues such as ICLR, particularly in recent years, and there is strong ML community interest in *de novo* molecular design.
> >
> > > Correctness: 3: Some of the paper’s claims have minor issues. A few statements are not well-supported, or require small changes to be made correct.
> >
> > We have clarified the reviewer’s misconceptions about our model’s correctness. Please see our responses above.
> >
> > > Technical Novelty And Significance: 2: The contributions are only marginally significant or novel.
> >
> > Please see our previous discussion about the technical novelty of our model, which explicitly separates the encoding of 3D shape and 2D chemical identity. Significantly, our careful and symmetry-robust model design has enabled our completely novel approaches in 3D shape-conditioned molecular generation and shape-constrained molecular optimization.
> >
> > > Empirical Novelty And Significance: 2: The contributions are only marginally significant or novel.
> >
> > Our task (equivariant, zero-shot generation of chemically diverse 3D molecules that fit a target shape) is the first of its kind, and we demonstrate clear benefits of our method over traditional shape-based virtual screening. Our work has significant impact for shape-constrained 3D drug design tasks such as scaffold hopping, as indicated by the reviewer. *No studies* have attempted to explicitly explore shape-constrained chemical space with 3D generative models in a symmetry-robust manner.
> >
> > Besides the important empirical novelty, significance, and practical utility of this task, we also emphasize that our model generates realistic 3D geometries for very flexible drug-like molecules in an end-to-end fashion. Given notable limitations of prior work in 3D molecular generation, this is not to be understated.

---

> > ### Comment · Reviewer_bX75 · 2022-11-17
> > **Thanks**
> >
> > Thank you for reporting new results in the paper. These results answered my concerns. It is a good application for drug discovery but the novelty of the model itself is limited. I raise my score to 6 now.

---

### Decision · Program_Chairs · 2023-01-20

**Decision:**

Accept: poster

**Justification For Why Not Higher Score:**

All reviewers are moderately supportive of this work, and none of them is willing to champion for this work.

**Justification For Why Not Lower Score:**

All reviewers are moderately supportive of this work, given the novelty of the studied problem of shape-conditional generation, and the proposed methods. The method can potentially be used in drug discovery based on existing drugs.

**Metareview: Summary, Strengths And Weaknesses:**

This paper studies 3D molecule generation conditioned on shapes of existing molecules. All reviewers are moderately supportive of this work, given the novelty of the studied problem of shape-conditional generation, and the proposed methods. The method can potentially be used in drug discovery based on existing drugs. Thus an accept is recommended.

**Note From Pc:**

if the above contains the word "oral" or "spotlight" please see: "oral" presentation means -> notable-top-5% and "spotlight" means -> notable-top-25%. As stated in our emails, we are disassociating presentation type from AC recommendations